# TREE-SLICED SOBOLEV IPM

**Viet-Hoang Tran***
Department of Mathematics
National University of Singapore
`hoang.tranviet@u.nus.edu`

**Thanh Q. Tran***
Department of Computer Science
National University of Singapore
`thanhtran@u.nus.edu`

**Thanh Chu**
Department of Computer Science
National University of Singapore
`thanh.chu@u.nus.edu`

**Duy-Tung Pham**
FPT Software AI Center
Vietnam
`tungpd10@fpt.com`

**Trung-Khang Tran**
Department of Computer Science
National University of Singapore
`TTK0106@u.nus.edu`

**Tam Le**[†]
Department of Advanced Data Science
Institute of Statistical Mathematics
`tam@ism.ac.jp`

**Tan Minh Nguyen**[†]
Department of Mathematics
National University of Singapore
`tanmn@nus.edu.sg`

## ABSTRACT

Recent work shows Tree-Sliced Optimal Transport to be an efficient and more expressive alternative to Sliced Wasserstein (SW), improving downstream performance. Tree-sliced metrics compare probability distributions by projecting measures onto tree metric spaces; a central example is the Tree-Sliced Wasserstein (TSW) distance, which applies the $1$-Wasserstein metric after projection. However, computing tree-based $p$-Wasserstein for general $p$ is costly, largely confining practical use to $p = 1$. This restriction is a significant bottleneck, as higher-order metrics ($p > 1$) are preferred in gradient-based learning for their more favorable optimization landscapes. In this work, we revisit Sobolev integral probability metrics (IPM) on trees to obtain a practical generalization of TSW. Building on the insight that a suitably regularized Sobolev IPM admits a closed-form expression, we introduce TS-Sobolev, a tree-sliced metric that aggregates regularized Sobolev IPMs over random tree systems and remains tractable for all $p \geq 1$; for $p > 1$, TS-Sobolev has the same computational complexity as TSW at $p = 1$. Notably, at $p = 1$ it recovers TSW exactly. Consequently, TS-Sobolev serves as a drop-in replacement for TSW in practical applications, with an additional flexibility in changing $p$. Furthermore, we extend this framework to define a corresponding metric for probability measures on hyperspheres. Experiments on Euclidean and spherical datasets show that TS-Sobolev and its spherical variant improve downstream performance in gradient flows, self-supervised learning, generative modeling, and text topic modeling over recent SW and TSW variants. Our code is available at https://github.com/thanhquangtran/TS-Sobolev.

## 1 INTRODUCTION

Comparing probability measures is a foundational problem in numerous scientific fields where data are often represented as distributions. For example, documents in natural language processing can

---

*Equal contribution
[†]Co–last author. Please correspond to: hoang.tranviet@u.nus.edu and tanmn@nus.edu.sg

be treated as distributions of words or topics (Sparck Jones, 1972; Kusner et al., 2015; Yurochkin et al., 2019), and 3D shapes in computer vision are commonly modeled as point clouds, which are discrete distributions of data points (Achlioptas et al., 2018; Hua et al., 2018; Wu et al., 2019). Optimal Transport (OT) has emerged as a powerful framework for this purpose (Villani, 2008; Peyré et al., 2019), as it defines a metric between distributions that inherently respects their underlying geometry. This key advantage has driven its broad adoption in fields like machine learning (Bunne et al., 2022; Fan et al., 2022; Takezawa et al., 2022), data valuation (Just et al., 2023; Kessler et al., 2025), multimodal data analysis (Park et al., 2024; Luong et al., 2024), statistics (Mena & Niles-Weed, 2019; Wang et al., 2022; Liu et al., 2022; Nietert et al., 2022), and computer vision (Lavenant et al., 2018; Saleh et al., 2022). However, a significant drawback of Optimal Transport (OT) is its computational complexity. For discrete measures supported by $n$ samples, standard algorithms scale as $\mathcal{O}(n^3 \log n)$ (Peyré et al., 2019), rendering OT impractical for large datasets.

**Sliced Optimal Transport.** The substantial computational burden of Optimal Transport (OT) led to the development of the Sliced Wasserstein (SW) distance as a powerful and efficient approximation (Rabin et al., 2011; Bonneel et al., 2015; Nguyen, 2025). At its core, SW simplifies the problem by leveraging the closed-form solution of one-dimensional OT. It projects high-dimensional probability measures onto random one-dimensional subspaces, computes the simple transport cost in each "slice", and averages these costs. This procedure reduces the computational complexity to that of sorting, $\mathcal{O}(n \log n)$ (Peyré et al., 2019), while faithfully preserving important statistical and topological properties of the original metric (Nadjahi et al., 2020; Bayraktar & Guo, 2021; Goldfeld & Greenewald, 2021). The success of this paradigm has inspired a broad family of extensions, including those using structured projections (Kolouri et al., 2019; Deshpande et al., 2019; Nguyen et al., 2020; Ohana et al., 2023; Nguyen et al., 2023) and adaptations for non-Euclidean geometries such as spheres (Bonet et al., 2022; Quellmalz et al., 2023) and hyperbolic space (Bonet et al., 2023b).

**Tree-Sliced Optimal Transport.** A key limitation of classical SW, however, is that one-dimensional projections can be insufficient for capturing complex geometric structures inherent in high-dimensional data. This has spurred research into alternative slicing domains beyond simple lines, with explorations across various metric spaces like Euclidean subspaces (Alvarez-Melis et al., 2018; Paty & Cuturi, 2019; Niles-Weed & Rigollet, 2022), graphs (Le et al., 2022), and non-Euclidean manifolds (Tran et al., 2024c; Bonet et al., 2023a; Lin et al., 2025). Among these, methods based on tree metrics have emerged as a particularly effective approach. The Tree-Sliced Wasserstein (TSW) distance, introduced by Tran et al. (2024d), capitalizes on closed-form OT solutions on tree metric spaces (Le et al., 2019; Indyk & Thaper, 2003; Indyk, 2001). TSW thus achieves a compelling balance: it maintains the computational tractability of SW while better representing the structure of the data. Subsequent works have improved TSW (Tran et al., 2025a;c; 2026b) and extended it to other domains, such as mixed-curvature spaces (Pham et al., 2026) and spheres (Tran et al., 2025d).

**Sobolev Integral Probability Metric.** The computational efficiency of the Tree-Sliced Wasserstein (TSW) distance is a direct result of the closed-form solution for the 1-Wasserstein distance on tree metric spaces (Le et al., 2019). A significant limitation of this framework, however, is that this analytical solution does not extend to orders $p > 1$, which restricts the applicability of TSW. This restriction is problematic because higher-order metrics are often preferred in gradient-based learning tasks. Specifically, $p$-Wasserstein with $p > 1$ offers strict convexity (Santambrogio, 2015; Villani, 2003) and smoother gradients compared to the $p = 1$ case, properties that are known to facilitate more stable and efficient optimization (Peyré et al., 2019). To address this gap, we leverage the framework of Integral Probability Metrics (IPMs), a powerful class of distances for comparing probability measures (Müller, 1997). IPMs function by finding a *critic function* from a predefined class that maximally discriminates between two distributions, a versatile principle with numerous applications in machine learning and statistics (Sriperumbudur et al., 2009; Gretton et al., 2012; Liang, 2019; Uppal et al., 2019; 2020; Nadjahi et al., 2020; Kolouri et al., 2020). A theoretically important instance is the *Sobolev IPM*, which constrains the critic function to a unit ball defined by the Sobolev norm (Adams & Fournier, 2003). This specific metric has been instrumental in theoretical analyses, such as studying convergence rates and error bounds in generative models (Liang, 2017; 2021; Singh et al., 2018). Despite its theoretical appeal, the standard Sobolev IPM lacks a closed-form expression, which has historically limited its practical use. A recent breakthrough by Le et al. (2025) overcomes this challenge by introducing a *regularized* Sobolev IPM for probability

measures supported on a tree. This novel formulation yields a closed-form solution that is both computationally efficient and valid for any order $p \geq 1$.

**Contributions.** Motivated by the expanding Tree-Sliced Wasserstein (TSW) framework and recent advances in Sobolev IPMs on tree metric spaces (Le et al., 2025), this paper introduces a novel family of tree-sliced distances. Our primary contribution is a scalable generalization of TSW for probability measures in Euclidean spaces and on the sphere. By leveraging the closed-form solution of the regularized Sobolev IPM, our proposed distance is efficiently computable for any order $p \geq 1$, overcoming a key limitation of previous TSW variants and leveraging the optimization advantages associated with higher-order metrics. The the paper is organized as follows:

1. In Section 2, we establish the building blocks for our method. We review the theory of tree metric spaces and detail the closed-form solution for the regularized Sobolev IPM on trees. We also revisit the Tree Systems framework and the Radon transform that underpin the Tree-Sliced Wasserstein (TSW) distance.

2. In Section 3, we introduce the Tree-Sliced Sobolev IPM (TS-Sobolev) for measures on Euclidean spaces and the Spherical Tree-Sliced Sobolev IPM (STS-Sobolev) for measures on the sphere. We establish their metric properties, prove they provide a scalable generalization of TSW for any order $p \geq 1$, and analyze their computational complexity.

3. Section 4 presents experiments on Euclidean and spherical data that validate our method's practical effectiveness and efficiency, followed by our conclusion in Section 5.

Supplementary materials, including detailed theoretical background, full proofs, and extended experimental results (setups, tables, and figures), are available in the Appendix.

## 2 BACKGROUND ON SOBOLEV INTEGRAL PROBABILITY METRIC AND TREE-SLICING

This section covers the two foundational concepts behind our method. We first review the *Sobolev Integral Probability Metric (Sobolev IPM)* and its efficient closed-form solution on tree metric spaces. We then describe the *tree-slicing framework* that projects measures from Euclidean space onto these tree metric spaces, thereby enabling the use of the efficient Sobolev IPM.

### 2.1 SOBOLEV INTEGRAL PROBABILITY METRIC

**Tree Metric Spaces.** A *tree metric space* $(\mathcal{T}, d_\mathcal{T})$ is a continuous space built from a tree $\mathcal{T} = (V, E)$ with vertices $V \subset \mathbb{R}^d$ and edges $E$. Each edge $e \in E$ is assigned a non-negative length $w_e$. Crucially, the space $\mathcal{T}$ includes not only the vertices $V$ but also every point along the edges $E$. Its *tree metric*, denoted $d_\mathcal{T}$, is the unique path distance between any two points on the tree (Semple & Steel, 2003b; Le et al., 2019). The unique path between points $x$ and $y$ is denoted $[x, y]$. This structure gives rise to a canonical Borel length measure, $\omega$, where the measure of any path equals its length: $\omega([x, y]) = d_\mathcal{T}(x, y)$. Finally, the subtree rooted at x, denoted $\Lambda(x)$, is the set of all points $y$ whose path from the root $r$ must pass through $x$, i.e., $\Lambda(x) = \{y \in \mathcal{T} : x \in [r, y]\}$.

**Comparing Measures on Trees.** A central task is to define a distance between probability measures on a tree $\mathcal{T}$. Let $\mathcal{P}(\mathcal{T})$ denote the collection of all prob-

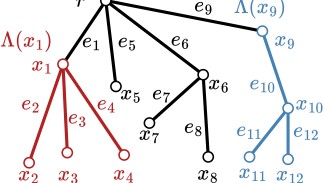

Figure 1: An illustration of a tree metric space. The tree, rooted at $r$, consists of nodes $x_i$ and edges $e_i$ with weights $w_e$. A probability distribution on the tree then assigns mass to nodes. The subtree $\Lambda(x)$ is the collection of all points lying along the edges in the subtree rooted at $x$. For example, $\Lambda(r)$ includes the entire tree, $\Lambda(x_1)$ (red) includes all points on edges $e_2$ $e_3$, and $e_4$, and $\Lambda(x_9)$ (blue) includes all points on edges $e_{10}$, $e_{11}$, and $e_{12}$.

ability measures on $\mathcal{T}$ (i.e., those with a total mass of 1). For any two measures $\mu, \nu \in \mathcal{P}(\mathcal{T})$, we write $\mathcal{P}(\mu, \nu)$ for the set of all valid couplings $\pi$ between them. A popular framework for defining a

distance is the *p-Wasserstein distance*:

$$\mathbf{W}_p(\mu, \nu) = \left( \inf_{\pi \in \mathcal{P}(\mu,\nu)} \int_{\mathcal{T} \times \mathcal{T}} d_{\mathcal{T}}(x,y)^p \, d\pi(x,y) \right)^{\frac{1}{p}}. \tag{1}$$

While powerful, computing this distance is generally expensive. However, for the special case of $p = 1$, the distance on a tree admits a fast, closed-form solution (Le et al., 2019):

$$\mathbf{W}_1(\mu, \nu) = \int_{\mathcal{T}} |\mu(\Lambda(x)) - \nu(\Lambda(x))| \, \omega(dx). \tag{2}$$

Crucially, for $p > 1$, no similar closed-form solution is known to exist. This computational bottleneck for higher-order Wasserstein distances on trees motivates our exploration of the Sobolev IPM, which provides a tractable alternative for all $p \geq 1$.

**Sobolev Integral Probability Metric.** The *Sobolev Integral Probability Metric (IPM)* is a distance between measures on a tree $\mathcal{T}$, defined using concepts from functional analysis. The metric is situated within the *Sobolev space* $W^{1,p}(\mathcal{T}, \omega)$, which consists of functions that have well-defined *tree derivatives*. This space is equipped with a *Sobolev norm*, which in turn defines a *unit ball* of functions, denoted $\mathcal{B}(p')$. For a rigorous treatment of these concepts, we refer to Appendix B. The Sobolev IPM is formally defined by finding the maximum discrepancy between two measures, $\mu$ and $\nu$, as evaluated by a critic function $f$ constrained to this unit ball:

$$\mathcal{S}_p(\mu, \nu) := \sup_{f \in \mathcal{B}(p')} \left| \int_{\mathcal{T}} f(x) \, \mu(dx) - \int_{\mathcal{T}} f(y) \, \nu(dy) \right| \tag{3}$$

where $p'$ is the conjugate of $p$ (where $p' \in [1, \infty]$ satisfies $1/p + 1/p' = 1$; if $p = 1$, then $p' = \infty$).

**Regularized Sobolev Integral Probability Metric.** The computational intractability of the standard form in Equation (3) motivates the use of a *Regularized Sobolev Integral Probability Metric (IPM)*, which admits a direct closed-form solution (Le et al., 2025). For continuous measures over the tree and for any order $1 \leq p < \infty$, this solution is given by:

$$\hat{\mathcal{S}}_p(\mu, \nu)^p = \int_{\mathcal{T}} \hat{w}(x)^{1-p} |\mu(\Lambda(x)) - \nu(\Lambda(x))|^p \, \omega(dx), \tag{4}$$

where $\hat{w}(x) := 1 + \omega(\Lambda(x))$ is a weight function.

For the practical case of discrete measures supported on the tree's nodes $V$, this integral simplifies to an efficiently computable sum over the tree's edges $E$:

$$\hat{\mathcal{S}}_p(\mu, \nu)^p = \sum_{e \in E} \beta_e |\mu(\gamma_e) - \nu(\gamma_e)|^p, \tag{5}$$

where $\gamma_e := \{y \in \mathcal{T} : e \subseteq [r, y]\}$ is the set of points whose path from the root contains edge $e$. The coefficient $\beta_e$ is a pre-computable scalar, making the overall computation efficient:

$$\beta_e = \begin{cases} \log\left(1 + \dfrac{w_e}{1 + \omega(\gamma_e)}\right) & \text{if } p = 2, \\ \dfrac{(1 + \omega(\gamma_e) + w_e)^{2-p} - (1 + \omega(\gamma_e))^{2-p}}{2 - p} & \text{otherwise.} \end{cases} \tag{6}$$

Analyzing Equation (4) shows that $p > 1$ yields gradients that scale with the error magnitude $|\mu(\Lambda(x)) - \nu(\Lambda(x))|$, unlike the constant gradients associated with $p = 1$, thereby facilitating smoother optimization consistent with established properties of $p$-Wasserstein (Santambrogio, 2015; Peyré et al., 2019). Furthermore, the term $\hat{w}(x)^{1-p}$ down-weights global gradients near the root, allowing the optimization to focus on refining fine-grained local structures at the leaves. This weighting constitutes a unique optimization advantage of TS-Sobolev over standard $p$-Wasserstein metrics. We provide a comprehensive analysis of these optimization advantages in Appendix F.8.

## 2.2 THE TREE-SLICING FRAMEWORK

While the Regularized Sobolev IPM defined in Section 2.1 offers a powerful metric for measures supported on trees, data in machine learning typically resides in Euclidean space $\mathbb{R}^d$. To bridge this

gap, we utilize the *Tree-Slicing framework* (Tran et al., 2024d). This framework provides a mechanism to project probability measures from $\mathbb{R}^d$ onto continuous tree metric spaces, thereby allowing us to leverage the closed-form solutions of tree-based metrics for high-dimensional Euclidean data.

**Tree System.** A *line* in $\mathbb{R}^d$ is an element of $\mathbb{R}^d \times \mathbb{S}^{d-1}$, and a *system of $k$ lines* is an element of $(\mathbb{R}^d \times \mathbb{S}^{d-1})^k$. We denote a system of lines by $\mathcal{L} = \{l_i\}_{i=1}^k$, where each line $l_i$ is defined by a source point $x_i \in \mathbb{R}^d$ and a direction vector $\theta_i \in \mathbb{S}^{d-1}$, with parameterization $x_i + t\,\theta_i$ for $t \in \mathbb{R}$.

A *tree system* is a system of lines endowed with an additional tree structure. To highlight this structure, we write $\mathcal{T} = \{l_i\}_{i=1}^k$. The *ground set* of this system, denoted $\bar{\mathcal{T}}$, is the set of all points on all lines in $\mathcal{T}$, formally defined as $\bar{\mathcal{T}} = \{(x, l_i) \in \mathbb{R}^d \times \mathcal{T} \mid x = x_i + t_x\theta_i \text{ for some } t_x \in \mathbb{R}\}$. This ground set forms a continuous tree metric space as defined in Section 2.1. The *space of tree systems* sharing a common tree structure is denoted by $\mathbb{T}_k^d$ (or simply $\mathbb{T}$), equipped with a probability distribution $\sigma$ induced by a random sampling procedure over the lines.

**Radon Transform on Tree Systems.** Let $\mathcal{C}(\mathbb{R}^d \times \mathbb{T}_k^d, \Delta_{k-1})$ be the set of continuous *splitting maps* from $\mathbb{R}^d \times \mathbb{T}_k^d$ to the $(k-1)$-simplex $\Delta_{k-1}$. For $f \in L^1(\mathbb{R}^d)$, define $\mathcal{R}_{\mathcal{T}}^\alpha f \colon \bar{\mathcal{T}} \to \mathbb{R}$ by

$$\mathcal{R}_{\mathcal{T}}^\alpha f(x, l_i) = \int_{\mathbb{R}^d} f(y)\,\alpha(y, \mathcal{T})_i\,\delta\big(t_x - \langle y - x_i,\, \theta_i \rangle\big)\,dy. \tag{7}$$

The operator

$$\mathcal{R}^\alpha \colon f \longmapsto \big(\mathcal{R}_{\mathcal{T}}^\alpha f\big)_{\mathcal{T} \in \mathbb{T}_k^d},$$

is called the *Radon Transform on Tree Systems*. This operator is *injective*.

**Tree Sliced Wasserstein distance.** The Tree-Sliced Wasserstein (TSW) distance is a metric between probability measures, defined as the expected 1-Wasserstein distance between the measures after projection onto a tree system. Recent variants include the *Tree-Sliced Wasserstein Distance on Systems of Lines* (TSW-SL) (Tran et al., 2024d) and its generalization, the *Distance-based Tree-Sliced Wasserstein Distance* (Db-TSW) (Tran et al., 2025a), which we refer to collectively as TSW in this paper. For two probability measures $\mu, \nu \in \mathcal{P}(\mathbb{R}^d)$ with corresponding densities $f_\mu, f_\nu$, the TSW distance is formally defined as:

$$\text{TSW}(\mu, \nu) \coloneqq \int_{\mathbb{T}} \text{W}_1\left(\mathcal{R}_{\mathcal{T}}^\alpha f_\mu, \mathcal{R}_{\mathcal{T}}^\alpha f_\nu\right)\,d\sigma(\mathcal{T}). \tag{8}$$

TSW is a valid metric on $\mathcal{P}(\mathbb{R}^d)$ and can be efficiently approximated via Monte Carlo sampling, thanks to the closed-form solution for the 1-Wasserstein distance on trees Equation (2).

It is crucial to distinguish the variants from earlier work that also uses the TSW name (Le et al., 2019; Sato et al., 2020; Yamada et al., 2022; Takezawa et al., 2022). Those methods were primarily designed for *static-support* measures where the data points are fixed. In contrast, the TSW-SL and Db-TSW formulations are specifically built for *dynamic-support* measures, where the points change during optimization. Our goal is to leverage this powerful tree-slicing framework to develop a new Sobolev IPM-based metric suitable for these dynamic tasks that mitigates the computational bottleneck of the p-Wasserstein distance on trees.

## 3 TREE-SLICED SOBOLEV IPM

In this section, we propose the Tree-Sliced Sobolev IPM (TS-Sobolev) framework for probability distributions on Euclidean spaces and the sphere. We establish its theoretical guarantees, clarify its connections to existing metrics, and characterize its computational complexity.

### 3.1 TREE-SLICED SOBOLEV IPM

For probability measures $\mu, \nu \in \mathcal{P}(\mathbb{R}^d)$ with respective densities $f_\mu$ and $f_\nu$, the *Tree-Sliced Sobolev IPM* is defined as the average regularized Sobolev IPM between $\mu$ and $\nu$ induced by tree-metric projections. Given a tree system $\mathcal{T} \in \mathbb{T}_k^d$ and a splitting map $\alpha \in \mathcal{C}(\mathbb{R}^d \times \mathbb{T}_k^d, \Delta_{k-1})$, the Radon transform $\mathcal{R}^\alpha$ maps $f_\mu, f_\nu$ to densities $\mathcal{R}_{\mathcal{T}}^\alpha f_\mu, \mathcal{R}_{\mathcal{T}}^\alpha f_\nu$ on $\mathcal{T}$, inducing measures $\mu_{\mathcal{T}}, \nu_{\mathcal{T}} \in \mathcal{P}(\mathcal{T})$. We then evaluate the regularized Sobolev IPM $\hat{\mathcal{S}}_p(\mu_{\mathcal{T}}, \nu_{\mathcal{T}})$ as in Equation (5) and define the resulting distance as the expectation of this quantity over $\mathbb{T}$ with respect to the sampling distribution $\sigma$.

**Definition 3.1** (Tree-Sliced Sobolev IPM). The *Tree-Sliced Sobolev IPM* of order $p \in [1, \infty)$, denoted as TS-Sobolev$_p$, between $\mu, \nu \in \mathcal{P}(\mathbb{R}^d)$ is defined by:

$$\text{TS-Sobolev}_p(\mu, \nu) := \left( \int_{\mathbb{T}} \hat{\mathcal{S}}_p(\mu_{\mathcal{T}}, \nu_{\mathcal{T}})^p \, d\sigma(\mathcal{T}) \right)^{\frac{1}{p}}. \tag{9}$$

## 3.2 Properties of Tree-Sliced Sobolev IPM

**Metricity of TS-Sobolev$_p$.** We recall the Euclidean group $\mathrm{E}(d)$ and state that $\mathrm{E}(d)$–invariance of the splitting map ensures that the Tree–Sliced Sobolev IPM defines a metric. Let $\mathbb{R}^d$ be equipped with the Euclidean norm $\| \cdot \|_2$. The *Euclidean group* $\mathrm{E}(d)$ is the group of all distance–preserving transformations of $\mathbb{R}^d$; it is the semidirect product $\mathrm{T}(d) \rtimes \mathrm{O}(d)$ of the translation group $\mathrm{T}(d) = \{x \mapsto x + v : v \in \mathbb{R}^d\}$ and the orthogonal group $\mathrm{O}(d) = \{Q \in \mathbb{R}^{d \times d} : Q^\top Q = I_d\}$. Every $g \in \mathrm{E}(d)$ can be written as $g = (Q, v)$ with $Q \in \mathrm{O}(d)$ and $v \in \mathbb{R}^d$, acting on $y \in \mathbb{R}^d$ by $gy = Qy + v$. This action extends to the space of tree systems $\mathbb{T}_k^d$ by $g\mathcal{T} = \{gl_i\}_{i=1}^k$; for lines represented as $(x_i, \theta_i)$, we set $gl_i := (Qx_i + v, Q\theta_i)$, which preserves the underlying tree structure. A splitting map $\alpha \in \mathcal{C}(\mathbb{R}^d \times \mathbb{T}_k^d, \Delta_{k-1})$ is $\mathrm{E}(d)$–*invariant* if

$$\alpha(gx, g\mathcal{T}) = \alpha(x, \mathcal{T}) \quad \text{for all } x \in \mathbb{R}^d, \ \mathcal{T} \in \mathbb{T}_k^d, \text{ and } g \in \mathrm{E}(d). \tag{10}$$

Invariance under the Euclidean group, $\mathrm{E}(d)$, is a desirable property for distances between probability measures on $\mathbb{R}^d$. Standard metrics like the 2-Wasserstein and Sliced $p$-Wasserstein distances possess this $\mathrm{E}(d)$-invariance. However, for the Tree-Sliced Sobolev IPM, this property is even more fundamental: it guarantees that TS-Sobolev is a valid metric on $\mathcal{P}(\mathbb{R}^d)$.

**Theorem 3.2.** TS-Sobolev *is an* $\mathrm{E}(d)$-*invariant metric on* $\mathcal{P}(\mathbb{R}^d)$.

The proof for Theorem 3.2 is presented in Appendix D.1.

**Connections to TSW.** The Tree-Sliced Sobolev IPM serves as a natural generalization of the Tree-Sliced Wasserstein (TSW) distance. Notably, for the order $p = 1$, the TS-Sobolev IPM *recovers the TSW distance exactly*. For any order $p \in [1, \infty)$, it is *upper-bounded by the TSW distance*.

**Theorem 3.3.** *For any* $\mu, \nu \in \mathcal{P}(\mathbb{R}^d)$ *and* $p \in [1, \infty)$: TS-Sobolev$_p(\mu, \nu)^p \leq \text{TSW}(\mu, \nu)$, *where equality holds for* $p = 1$, *i.e.,* TS-Sobolev$_1(\mu, \nu) = \text{TSW}(\mu, \nu)$.

The proof for Theorem 3.3 appear in Appendix D.2.

**Remark 3.4.** We note that the values of TS-Sobolev$_p$ and TSW depends on the choice of tree system ($\mathbb{T}$), sampling distribution ($\sigma$), and splitting map ($\alpha$). In this paper, we utilize the specific choices established in the Distance-based TSW (Db-TSW) framework (Tran et al., 2025a). Therefore, the TSW we analyze is precisely Db-TSW. To maintain readability, these dependencies are suppressed in the notation for TS-Sobolev$_p$, with a detailed description of the framework available in Appendix C.

**Computation of Tree-Sliced Sobolev IPM.** The intractable integral in Equation (9) is approximated using a Monte Carlo estimate:

$$\widehat{\text{TS-Sobolev}}_p(\mu, \nu) = \left( \frac{1}{L} \sum_{i=1}^L \hat{\mathcal{S}}_p(\mu_{\mathcal{T}_i}, \nu_{\mathcal{T}_i})^p \right)^{\frac{1}{p}}, \tag{11}$$

where we sample $L$ tree systems $\{\mathcal{T}_i\}_{i=1}^L$ from a distribution $\sigma$. Let $\mu$ and $\nu$ be discrete measures with $n$ and $m$ support points, respectively, and assume $n \gg m$. The overall computational complexity is $\mathcal{O}(Lkn \log n + Lkdn)$, where $k$ is the number of lines per tree and $d$ is the data dimension. This is identical to the complexity of TSW variants like Db-TSW (Tran et al., 2025a), as the extra step of computing the coefficients $\beta_e$ per Equation (6) adds a negligible $\mathcal{O}(Lkn)$ cost.

A key advantage is that this complexity holds for any order $p \in [1, \infty)$, resolving the computational intractability of higher-order TSW. Empirically, the runtime of TS-Sobolev is nearly identical to that of the first-order Db-TSW, confirming its efficiency. A runtime analysis is provided in Appendix F.1.

The practical application of TS-Sobolev depends on its hyperparameters: the number of trees $L$, lines per tree $k$, and the order $p$. Prior work (Tran et al., 2025a) shows that using multiple lines

$(k > 1)$ is crucial for capturing complex data topology. This creates a natural trade-off between increasing $k$ for expressiveness and increasing $L$ to improve the precision of the Monte Carlo estimate by reducing its variance. The convergence rate of this estimate with respect to $L$ is formalized in Theorem 3.5. A detailed sensitivity analysis for the tree parameters $L$ and $k$ is presented in Appendix F.6, while the influence of the order $p$ is analyzed in Appendix F.7.

**Theorem 3.5.** *The approximation error of TS-Sobolev decreases at a rate of $\mathcal{O}(L^{-1/2})$.*

We defer the proof for Theorem 3.5 to Appendix D.3.

### 3.3 EXTENSION TO THE SPHERICAL SETTING

The TS-Sobolev$_p$ framework extends to measures on the hypersphere, $\mu, \nu \in \mathcal{P}(\mathbb{S}^d)$, by using spherical tree systems (Tran et al., 2025d). We provide a brief derivation of the resulting metric below, deferring a complete treatment to Appendix E. The core idea is to use the *spherical Radon transform on spherical tree systems* to map the densities of $\mu$ and $\nu$ onto a given tree $\mathcal{T}$, which induces the projected measures $\mu_\mathcal{T}$ and $\nu_\mathcal{T}$.

**Definition 3.6** (Spherical Tree-Sliced Sobolev IPM)**.** The *Spherical Tree-Sliced Sobolev IPM* of order $p \in [1, \infty)$, denoted as STS-Sobolev$_p$, between $\mu, \nu \in \mathcal{P}(\mathbb{S}^d)$ is defined by

$$\text{STS-Sobolev}_p(\mu, \nu) := \left( \int_\mathbb{T} \hat{\mathcal{S}}_p(\mu_\mathcal{T}, \nu_\mathcal{T})^p \, d\sigma(\mathcal{T}) \right)^{\frac{1}{p}}. \tag{12}$$

A detailed derivation of STS-Sobolev$_p$, along with a full analysis of its properties, is provided in Appendices E.2 and E.3, respectively. Notably, for the $p = 1$ case, STS-Sobolev$_1$ recovers the Spherical Tree-Sliced Wasserstein (STSW) distance exactly, as our implementation adopts the specific splitting map and tree sampling methodology from Tran et al. (2025d).

## 4 EXPERIMENTAL RESULTS

In this section, we empirically evaluate our proposed methods across a diverse range of applications to demonstrate their effectiveness in both Euclidean and spherical settings. For the *Euclidean setting*, we conduct experiments on *gradient flows* and generative modeling with *diffusion models*. For the *spherical setting*, our evaluation focuses on a *self-supervised learning* benchmark. Additionally, we assess our methods on a *topic modeling* task, for which we provide results in both domains.

### 4.1 EVALUATION ON EUCLIDEAN DATA

**Gradient Flow on $\mathbb{R}^d$.** In this experiment, we apply our methods to a gradient flow task, which seeks to find a path of distributions $\mu_t$ that minimizes a distance $\mathcal{D}$ between an initial source $\mu_0 = \mathcal{N}(0, I)$ and a fixed target $\nu$. The evolution of this path is governed by the update rule $\partial_t \mu_t = -\nabla \mathcal{D}(\mu_t, \nu)$, where $\mu_t$ is the distribution at time $t$ and $\nabla \mathcal{D}(\mu_t, \nu)$ is the corresponding distance gradient. Our evaluation is conducted on the *8 Gaussians* and *Gaussian 30d* datasets, where we benchmark our *TS-Sobolev* variants ($p \in \{1.2, 1.5, 2\}$) against a comprehensive suite of baselines. These include Sliced-Wasserstein (SW) methods—such as vanilla SW (Bonneel et al., 2015), MaxSW (Deshpande et al., 2019), LCVSW (Nguyen & Ho, 2023), and SWGG (Mahey et al., 2023)—as well as recent Tree-Sliced (TSW) distances like TSW-SL (Tran et al., 2024d), Db-TSW, and Db-TSW$^\perp$ (Tran et al., 2025a). We assess performance by measuring the Wasserstein distance to the target at intervals up to 2500 iterations, with detailed results available in Table 1.

The results demonstrate that our TS-Sobolev$_p$ methods ($p \in \{1.2, 1.5, 2\}$) achieve better convergence compared to baselines. While some SW variants perform well initially, our methods consistently improve and ultimately outperform their Wasserstein-based TSW counterparts. For example, on the *8 Gaussians* dataset, TS-Sobolev$_{1.2}$ achieves a final distance of $8.88 \times 10^{-7}$, outperforming the strongest baseline, Db-TSW ($2.50 \times 10^{-6}$). Similarly, on the *Gaussian 30d* dataset, TS-Sobolev$_{1.2}$ has the best final distance of $1.40$, surpassing both TSW-SL ($1.93$) and Db-TSW ($1.78$).

**Diffusion Models.** This experiment applies our proposed TS-Sobolev distance to the task of training Denoising Diffusion Generative Adversarial Networks (DDGANs) (Xiao et al., 2021) for unconditional image synthesis. Following the approach of Nguyen et al. (2024), we integrate our

Table 1: Average Wasserstein distance (multiplied by $10^{-1}$ for Gaussian 30d) between source and target distributions of 10 runs. All methods use 100 projecting directions.

| Methods | 8 Gaussians | | | | | Gaussian 30d | | | | |
|---|---|---|---|---|---|---|---|---|---|---|
| | Iteration | | | | | Iteration | | | | |
| | 500 | 1000 | 1500 | 2000 | 2500 | 500 | 1000 | 1500 | 2000 | 2500 |
| SW | 3.97e-2 | 6.48e-3 | 1.08e-3 | 1.09e-3 | 1.08e-3 | 2.93 | 2.87 | 2.80 | 2.72 | 2.64 |
| MaxSW | 4.66e-2 | 3.53e-2 | 2.74e-2 | 2.33e-2 | 2.08e-2 | **2.24** | 2.53 | 2.68 | 2.68 | 2.64 |
| SWGG | 7.57e-3 | 7.00e-5 | 5.80e-5 | 5.68e-5 | 5.71e-5 | 2.72 | 2.74 | 2.74 | 2.74 | 2.74 |
| LCVSW | 7.50e-4 | 5.42e-4 | 5.53e-4 | 5.58e-4 | 5.43e-4 | 2.85 | 2.71 | 2.58 | 2.45 | 2.33 |
| TSW-SL | 1.92e-2 | 7.42e-4 | 1.34e-6 | 1.33e-6 | 1.17e-6 | 2.48 | 2.31 | 2.16 | 2.04 | 1.93 |
| Db-TSW | **8.18e-5** | 2.51e-6 | 2.26e-6 | 2.24e-6 | 2.50e-6 | 2.44 | 2.24 | 2.07 | 1.90 | 1.78 |
| TS-Sobolev$_{1.2}$ | 2.12e-2 | **1.95e-6** | **1.25e-6** | **1.08e-6** | **8.88e-7** | 2.38 | **2.10** | **1.85** | **1.62** | **1.40** |
| TS-Sobolev$_{1.5}$ | 2.93e-2 | 1.17e-3 | 3.28e-6 | 2.27e-6 | 2.03e-6 | 2.44 | 2.25 | 2.02 | 1.77 | 1.51 |
| TS-Sobolev$_2$ | 3.05e-2 | 8.69e-3 | 1.43e-4 | 4.54e-6 | 3.50e-6 | 3.21 | 3.49 | 3.58 | 3.61 | 3.68 |

Table 2: A comparison of DDGAN models on the CIFAR-10 unconditional generation benchmark, showing Fréchet Inception Distance (FID) scores and per-epoch training times averaged over 10 runs.

| Model | FID ↓ | Time/Epoch (s) ↓ |
|---|---|---|
| DDGAN (Xiao et al., 2021) | 3.64 | 72 |
| SW-DD (Nguyen et al., 2024) | 2.90 | 74 |
| DSW-DD (Nguyen et al., 2024) | 2.88 | 498 |
| EBSW-DD (Nguyen et al., 2024) | 2.87 | 76 |
| RPSW-DD (Nguyen et al., 2024) | 2.82 | 76 |
| IWRPSW-DD (Nguyen et al., 2024) | 2.70 | 77 |
| TSW-SL-DD (Tran et al., 2024d) | 2.83 | 80 |
| Db-TSW-DD (Tran et al., 2025a) | 2.60 | 84 |
| Db-TSW-DD$^{\perp}$ (Tran et al., 2025a) | 2.53 | 85 |
| TS-Sobolev$_{1.5}$-DD (ours) | $2.302 \pm 0.004$ | 84 |
| TS-Sobolev$_2$-DD (ours) | **$2.277 \pm 0.003$** | 84 |

Table 3: Accuracy of the linear classifier on encoded (E) features and projected (P) features on $\mathbb{S}^9$.

| Method | Acc. E(%) ↑ | Acc. P(%) ↑ |
|---|---|---|
| Hypersphere | 79.76 | 74.57 |
| SimCLR | 79.69 | 72.78 |
| SSW | 70.46 | 64.52 |
| S3W | 78.54 | 73.84 |
| RI-S3W (5) | 79.97 | 74.27 |
| ARI-S3W (5) | 79.92 | 75.07 |
| STSW | 80.53 | 76.78 |
| STS-Sobolev$_{1.5}$ | 79.88 | 76.07 |
| STS-Sobolev$_2$ | **80.6** | **77.65** |

distance into the Augmented Generalized Mini-batch Energy (AGME) loss function. We benchmark TS-Sobolev against several Sliced and Tree-Sliced Wasserstein-based DDGAN variants, with full experimental details available in Appendix F.3.

The results, summarized in Table 2, show that our methods yield notable improvements in sample quality. Notably, both TS-Sobolev$_{1.5}$-DD and TS-Sobolev$_2$-DD surpass the strongest baseline, Db-TSW-DD$^{\perp}$ (Tran et al., 2025a), reducing the Fréchet Inception Distance (FID) by 0.228 and 0.253, respectively. This gain in sample quality is achieved without a trade-off in efficiency, as our methods have comparable training times to other Tree-Sliced variants.

We attribute TS-Sobolev's performance gains to its ability to preserve fine-grained image details, as theoretically and empirically demonstrated in Appendix F.8.

## 4.2 EVALUATION ON SPHERICAL DATA

**Self-Supervised Learning (SSL).** Previous work by (Wang & Isola, 2020) demonstrated that the contrastive objective can be separated into two key components: an alignment loss, which ensures that embeddings of similar inputs remain close, and a uniformity loss, which prevents collapse by encouraging the representations to distribute more evenly. Building on the idea of (Bonet et al., 2022), we substitute the Gaussian kernel used in the uniformity term with our proposed method.

$$\mathcal{L} = \underbrace{\frac{1}{n} \sum_{i=1}^{n} \left\| z_i^A - z_i^B \right\|_2^2}_{\text{Alignment loss}} + \underbrace{\frac{\lambda}{2} \left( \text{STS-Sobolev}_p(z^A, \nu) + \text{STS-Sobolev}_p(z^B, \nu) \right)}_{\text{Uniformity loss}}$$

Table 4: Log of the Wasserstein distance between source and target distributions over 10 runs on a mixture of 12 vMFs.

| Methods | Epoch | | | | |
|---|---|---|---|---|---|
| | 50 | 100 | 150 | 200 | 250 |
| SSW | $-2.439 \pm 0.053$ | $-2.787 \pm 0.040$ | $-2.909 \pm 0.041$ | $-2.979 \pm 0.037$ | $-3.014 \pm 0.034$ |
| S3W | $-2.022 \pm 0.036$ | $-2.211 \pm 0.045$ | $-2.284 \pm 0.056$ | $-2.290 \pm 0.054$ | $-2.289 \pm 0.064$ |
| RI-S3W (1) | $-2.094 \pm 0.028$ | $-2.488 \pm 0.028$ | $-2.693 \pm 0.025$ | $-2.814 \pm 0.029$ | $-2.900 \pm 0.026$ |
| RI-S3W (5) | $-2.433 \pm 0.029$ | $-2.790 \pm 0.023$ | $-2.939 \pm 0.019$ | $-3.032 \pm 0.026$ | $-3.093 \pm 0.021$ |
| ARI-S3W (30) | $-2.612 \pm 0.043$ | $-2.942 \pm 0.029$ | $-3.090 \pm 0.035$ | $-3.189 \pm 0.039$ | $-3.270 \pm 0.047$ |
| LSSOT | $-2.078 \pm 0.030$ | $-2.444 \pm 0.023$ | $-2.546 \pm 0.023$ | $-2.582 \pm 0.023$ | $-2.598 \pm 0.021$ |
| STSW | $-2.693 \pm 0.030$ | $-3.171 \pm 0.041$ | $-3.376 \pm 0.031$ | $-3.488 \pm 0.049$ | $\underline{-3.549 \pm 0.072}$ |
| STS-Sobolev$_{1.5}$ | $\mathbf{-3.099 \pm 0.032}$ | $-3.324 \pm 0.050$ | $-3.427 \pm 0.055$ | $-3.499 \pm 0.064$ | $-3.540 \pm 0.078$ |
| STS-Sobolev$_2$ | $\underline{-3.081 \pm 0.026}$ | $\mathbf{-3.376 \pm 0.058}$ | $\mathbf{-3.513 \pm 0.094}$ | $\mathbf{-3.578 \pm 0.108}$ | $\mathbf{-3.616 \pm 0.123}$ |

where $\nu = \mathcal{U}(\mathbb{S}^d)$ represents the uniform distribution on the unit sphere $\mathbb{S}^d$, $z^A, z^B \in \mathbb{R}^{n \times (d+1)}$ denote the embeddings of two augmented views of the same sample and $\lambda > 0$ serves as a weight to balance the alignment and uniformity terms. Following the approach in Bonet et al. (2022); Tran et al. (2024c; 2025d), we apply this objective to pretrain a ResNet18 He et al. (2016) encoder on CIFAR-10 Krizhevsky et al. (2009) for 200 epochs. After pretraining, a linear classifier is trained on top of the frozen encoder to evaluate learned features.

As shown in Table 3, our proposed STS-Sobolev variants demonstrate superior performance compared to both tree-sliced and standard sliced baselines. STS-Sobolev$_2$ achieves the best overall accuracy (80.6% Encoded / 77.65% Projected), outperforming its direct tree-based counterpart, STSW (Tran et al., 2025d). Furthermore, our method significantly improves upon standard spherical slicing approaches, such as SSW (Bonet et al., 2022) and S3W variants (Tran et al., 2024c).

**Gradient Flow on the sphere.** In this task, our objective is to learn the target distribution $\nu$ from a source distribution $\mu$ by minimizing $d(\nu, \mu)$ where $d$ is the distance metric such as SSW (Bonet et al., 2022), S3W (Tran et al., 2024c), LSSOT (Liu et al., 2025) and STSW (Tran et al., 2025d). Consistent with prior works Bonet et al. (2022); Tran et al. (2024c; 2025d), we use a mixture of 12 von Mises-Fisher distributions (vMFs) with 2400 samples as the target distribution. Optimization is carried out using projected gradient descent Bonet et al. (2022) on the sphere with full-batch training. We report in Table 4 the log 2-Wasserstein distance at epochs 50, 100, 150, 200, and 250, averaged over 10 runs. Across all epochs, our proposed STS-Sobolev consistently outperforms the baselines. In particular, while STS-Sobolev$_{1.5}$ achieves the best result at epoch 50, STS-Sobolev$_2$ demonstrates the strongest overall performance at other epochs. At the final epoch, STS-Sobolev$_2$ achieves a distance of $-3.616 \pm 0.123$, surpassing its tree-sliced counterpart STSW ($-3.549 \pm 0.072$) as well as the sliced baselines LSSOT ($-2.598 \pm 0.021$) and ARI-S3W ($-3.270 \pm 0.047$).

**Topic Modeling.** Topic modeling task (Blei et al., 2003) seeks to automatically extract distinct themes from collections of text documents, revealing the underlying structure of a corpus. Recent neural approaches typically employ a variational autoencoder (VAE) setup, in which the optimization balances accurate document reconstruction with a regularization that encourages the inferred topic distributions to resemble a chosen prior (Srivastava & Sutton, 2017). Inspired by Nan et al. (2019); Adhya & Sanyal (2025), we propose replacing the conventional KL-divergence regularizer with a Wasserstein-based alternative. This leads to the following objective:

$$\inf_{\varphi, \psi} \mathbb{E}_{p(\mathbf{x})} \mathbb{E}_{q_\varphi(\theta | \mathbf{x})} \left[ \text{CE}(\mathbf{x}, \hat{\mathbf{x}}) \right] + \lambda \, \text{STS-Sobolev}_p(q_\varphi(\theta), p(\theta)),$$

where CE represents the cross-entropy between the input document $\mathbf{x}$ (in bag-of-words representation) and its reconstruction $\hat{\mathbf{x}}$. The variational posterior $q_\varphi(\theta | \mathbf{x})$ is generated by encoder $\varphi$, and the decoder $\psi$ maps topic mixtures $\theta$ back to word distributions to form $\hat{\mathbf{x}}$. The encoder $\varphi$ can be adapted to output $\theta$ in either $\mathbb{R}^d$ or $\mathbb{S}^d$, which allows for a direct evaluation of our methods in both the Euclidean and spherical settings.

We evaluate our proposed TS-Sobolev$_2$ and STS-Sobolev$_2$ topic models against a comprehensive set of baselines, with all results summarized in Table 5. In the *Euclidean setting*, our model is benchmarked against several modern Sliced Wasserstein (SW) and Tree-Sliced Wasserstein (TSW)

Table 5: Average topic coherence CV($\uparrow$) on BBC and M10 over 10 runs.

| Method | BBC | M10 |
|---|---|---|
| LDA (Blei et al., 2003) | $0.457_{\pm 0.054}$ | $0.341_{\pm 0.018}$ |
| ProdLDA (Srivastava & Sutton, 2017) | $0.688_{\pm 0.018}$ | $0.491_{\pm 0.017}$ |
| WTM (Nan et al., 2019) | $0.741_{\pm 0.034}$ | $0.403_{\pm 0.047}$ |
| *Euclidean setting* | | |
| SW (Bonneel et al., 2015) | $0.760_{\pm 0.048}$ | $0.484_{\pm 0.043}$ |
| RPSW (Nguyen et al., 2024) | $0.775_{\pm 0.026}$ | $0.472_{\pm 0.032}$ |
| EBRPSW (Nguyen et al., 2024) | $0.777_{\pm 0.019}$ | $0.490_{\pm 0.054}$ |
| TSW-SL (Tran et al., 2024d) | $0.796_{\pm 0.038}$ | $0.456_{\pm 0.040}$ |
| Db-TSW (Tran et al., 2025a) | $0.787_{\pm 0.041}$ | $0.458_{\pm 0.081}$ |
| TS-Sobolev$_2$ (Ours) | $\mathbf{0.805}_{\pm 0.029}$ | $\mathbf{0.497}_{\pm 0.051}$ |
| *Spherical setting* | | |
| SSW (Adhya & Sanyal, 2025) | $0.755_{\pm 0.050}$ | $0.408_{\pm 0.018}$ |
| S3W (Tran et al., 2024c) | $0.700_{\pm 0.051}$ | $0.402_{\pm 0.056}$ |
| LSSOT (Liu et al., 2025) | $0.743_{\pm 0.051}$ | $0.395_{\pm 0.049}$ |
| STSW (Tran et al., 2025d) | $0.752_{\pm 0.031}$ | $0.373_{\pm 0.043}$ |
| STS-Sobolev$_2$ (Ours) | $\mathbf{0.776}_{\pm 0.032}$ | $\mathbf{0.423}_{\pm 0.059}$ |

variants from recent works (Bonneel et al., 2015; Nguyen et al., 2024; Tran et al., 2024d; 2025a). In the *spherical setting*, we compare against recent spherical slicing methods, specifically SSW (Adhya & Sanyal, 2025), STSW (Tran et al., 2025d), S3W (Tran et al., 2024c), and LSSOT (Liu et al., 2025). To provide a broader context, we also include results from three foundational topic models: LDA (Blei et al., 2003), ProdLDA (Srivastava & Sutton, 2017), and WTM (Nan et al., 2019).

To assess the quality of the discovered topics, we use the standard coherence metric $C_V$ (Röder et al., 2015), where higher values indicate better topic quality. The results, summarized in Table 5, show that our proposed methods consistently achieve the highest topic coherence scores in all settings. In the *Euclidean setting*, our TS-Sobolev$_2$ model obtains a top score of $0.805$ on the BBC dataset, surpassing the best baseline score of $0.796$. On the M10 dataset, it also leads with a score of $0.497$ compared to the baseline best of $0.490$. This strong performance extends to the *spherical setting*, where our STS-Sobolev$_2$ model attains scores of $0.776$ on BBC and $0.423$ on M10, outperforming the strongest respective baselines ($0.755$ and $0.408$). Full details on the experimental setup are provided in Appendix F.5.

## 5 CONCLUSION

This paper introduced the *Tree-Sliced Sobolev Integral Probability Metric (TS-Sobolev)* and its spherical variant, *STS-Sobolev*, as novel approaches for comparing probability measures. Our work generalizes the Tree-Sliced Wasserstein (TSW) framework by leveraging a regularized Sobolev IPM, enabling the efficient computation of tree-sliced distances for any order $p \geq 1$. We presented a formal derivation of these metrics and provided comprehensive theoretical guarantees, including proofs of metricity and the formal connection to the original TSW distance. Experimental evaluations show that TS-Sobolev and STS-Sobolev consistently outperform state-of-the-art Sliced Wasserstein and Tree-Sliced Wasserstein methods across various tasks, including topic modeling and training diffusion models. Crucially, these performance gains are achieved with no additional computational overhead, as our methods maintain a runtime comparable to existing TSW techniques. A limitation of our current work is that TS-Sobolev is not designed to compare unbalanced measures, where input measures may have different total masses. Therefore, a promising future direction is to design an extension of our method for the unbalanced optimal transport setting.

ACKNOWLEDGMENTS

We thank the area chairs and anonymous reviewers for their comments. TL gratefully acknowledges the support of the JST-BOOST program (FY2025), JSPS KAKENHI Grant number 23K11243, and Mitsui Knowledge Industry Co., Ltd. grant. TT acknowledges support from the Application Driven Mathematics Program funded and organized by the Vingroup Innovation Fund and VinBigData.

This research / project is supported by the National Research Foundation Singapore under the AI Singapore Programme (AISG Award No: AISG2-TC-2023-012-SGIL). This research / project is supported by the Ministry of Education, Singapore, under the Academic Research Fund Tier 1 (FY2023) (A-8002040-00-00, A-8002039-00-00). This research / project is also supported by the NUS Presidential Young Professorship Award (A-0009807-01-00) and the NUS Artificial Intelligence Institute–Seed Funding (A-8003062-00-00).

**Ethics Statement.** Given the nature of the work, we do not foresee any negative societal and ethical impacts of our work.

**Reproducibility Statement.** Source codes for our experiments are provided in the supplementary materials of the paper. The details of our experimental settings and computational infrastructure are given in Appendix F. All datasets that we used in the paper are published, and they are easy to access in the Internet.

**LLM Usage Declaration.** We use large language models (LLMs) for grammar checking and correction.

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

TABLE OF NOTATION

| | |
|---|---|
| $\mathbb{R}^d$ | $d$-dimensional Euclidean space |
| $\|\cdot\|_2$ | Euclidean norm |
| $\mathbb{S}^{d-1}$ | $(d-1)$-dimensional hypersphere |
| $\theta, \psi$ | unit vector |
| $\sqcup$ | disjoint union |
| $L^1(X)$ | space of Lebesgue integrable functions on $X$ |
| $\mathcal{P}(X)$ | space of probability measures on $X$ |
| $\mathcal{M}(X)$ | space of measures on $X$ |
| $\mu, \nu$ | measures |
| $\delta(\cdot)$ | 1-dimensional Dirac delta function |
| $\mathcal{U}(\mathbb{S}^{d-1})$ | uniform distribution on $\mathbb{S}^{d-1}$ |
| $\mathcal{C}(X, Y)$ | space of continuous maps from $X$ to $Y$ |
| $d(\cdot, \cdot)$ | metric in metric space |
| $\mathrm{T}(d)$ | translation group of order $d$ |
| $\mathrm{O}(d)$ | orthogonal group of order $d$ |
| $d_{\mathcal{T}}(\cdot, \cdot)$ | tree metric |
| $\mathrm{E}(d)$ | Euclidean group of order $d$ |
| $\mathrm{W}_p$ | $p$-Wasserstein distance |
| $\mathrm{SW}_p$ | Sliced $p$-Wasserstein distance |
| $\Lambda$ | (rooted) subtree |
| $\mathcal{S}_p$ | $p$-Sobolev Integral Probability Metric |
| $\hat{\mathcal{S}}_p$ | $p$-Regularized Sobolev Integral Probability Metric |
| $\mathcal{T}$ | tree system or spherical tree system |
| $r_y^x$ | spherical ray |
| $L$ | number of Monte Carlo samples |
| $k$ | number of lines in a system of lines or a tree system |
| $\mathcal{R}^\alpha$ | Radon Transform on Systems of Lines, or Radon Transform on Spherical Trees |
| $\Delta_{k-1}$ | $(k-1)$-dimensional standard simplex |
| $\alpha$ | splitting map |
| $\xi, \zeta, c$ | tuning parameter |
| $\mathbb{T}$ | space of tree systems |
| $\sigma$ | distributions on (components of) space of tree systems |
| $\mathcal{N}$ | normal (Gaussian) distribution |
| $\mathcal{U}$ | uniform distribution |
| $\delta$ | Dirac delta distribution |

# Appendix of "Tree-Sliced Sobolev IPM"

## Table of Contents

## A    BACKGROUND ON DISTANCES ON METRIC SPACES WITH TREE METRICS

We denote a tree as $\mathcal{T} = (V, E)$, where $V$ and $E$ represent the sets of vertices and edges, respectively, and let $r \in V$ be the designated root. Each edge $e \in E$ is associated with a non-negative weight $w_e$ that denotes its length. Following Semple & Steel (2003a), we now provide a formal definition of the corresponding tree metric.

**Definition A.1** (Tree metric). Let $\Omega$ be a set and let $d : \Omega \times \Omega \to [0, \infty)$ be a metric. We say that $d$ is a *tree metric* on $\Omega$ if there exists a weighted tree $\mathcal{T}$ such that $\Omega \subseteq V(\mathcal{T})$, and for any $x, y \in \Omega$, the distance $d(x, y)$ is equal to the total weight of the unique path in $\mathcal{T}$ connecting $x$ and $y$.

Suppose $V$ is a subset of a vector space, and let $d_{\mathcal{T}}(\cdot, \cdot)$ denote the tree metric associated with $\mathcal{T}$. For any two points $x, y \in \mathcal{T}$, let $[x, y]$ represent the unique shortest path in $\mathcal{T}$ connecting them. Consider the unique Borel (length) measure $\omega$ on $\mathcal{T}$ such that

$$\omega([x, y]) = d_{\mathcal{T}}(x, y), \quad \forall\, x, y \in \mathcal{T}. \tag{13}$$

Additionally, given a root $r \in \mathcal{T}$, we define the subtree rooted at $x \in \mathcal{T}$ as

$$\Lambda(x) := \{\, y \in \mathcal{T} : x \in [r, y] \,\}. \tag{14}$$

We denote by $\mathcal{P}(\mathcal{T})$ the set of all probability measures on $\mathcal{T}$, that is, the collection of Borel measures on $\mathcal{T}$ with total mass equal to one. We now define the Wasserstein distanc on the space metric following the work of Le et al. (2019).

**Theorem A.2** (Wasserstein on Tree Metric Spaces). *Let $\mu, \nu \in \mathcal{P}(\mathcal{T})$. Then the 1-Wasserstein distance with respect to the tree metric $d_{\mathcal{T}}$ can be expressed as*

$$\mathbf{W}_{1, d_{\mathcal{T}}}(\mu, \nu) = \int_{\mathcal{T}} \big| \mu(\Lambda(x)) - \nu(\Lambda(x)) \big|\, \omega(dx), \tag{15}$$

*where $\Lambda(x)$ denotes the subtree of $\mathcal{T}$ rooted at $x$, and $\omega$ is the associated length measure on $\mathcal{T}$.*

While the 1-Wasserstein distance ($W_1$) on a tree has a convenient closed-form solution, this is generally not true for higher orders ($p > 1$). This computational challenge motivated the development of alternatives like Sobolev Transport (ST), which provides a scalable and valid metric for comparing probability measures on tree and graph structures (Le et al., 2022).

The key idea behind ST is to modify the constraints on the "critic" function used to differentiate between two measures. Instead of the standard Lipschitz condition, it constrains the critic function within a graph-based Sobolev space, primarily by limiting the $L^p$-norm of the function's gradient (Le et al., 2022). This approach has proven versatile, with extensions for measures of different total masses (Le et al., 2023) and for more general geometric structures beyond the standard $L^p$ framework (Le et al., 2024).

A closely related concept is the Sobolev Integral Probability Metric (Sobolev IPM), which is a type of IPM where the critic function is constrained to a unit ball defined by the full Sobolev norm—a measure that considers both the function's values and its gradient (Adams & Fournier, 2003). The crucial innovation is a regularized variant of this metric. By relaxing the constraint to focus only on the gradient of the critic function, the regularized Sobolev IPM successfully admits a closed-form solution for any order $p > 1$, making it a powerful and computationally efficient tool for comparing measures on trees (Le et al., 2025).

## B    BACKGROUND ON REGULARIZED SOBOLEV IPM FOR MEASURES ON TREE

In this section, we introduce the framework of regularized Sobolev integral probability metrics (IPMs) for probability measures supported on tree structures. Specifically, we begin by reviewing the tree setting for probability measures, including the relevant notational conventions and the functional spaces. We then formulate the Sobolev IPM problem for measures supported on trees. Finally, we introduce the regularized variant of the Sobolev IPM, which admits a closed-form solution. Proofs in Section closely follow that of Le et al. (2025).

## B.1 SETTING AND NOTATIONS

Let $\mathcal{T} = (V, E)$ be a finite rooted tree, where $V$ denotes the set of vertices and $E$ the set of edges. For each edge $e \in E$, we associate a positive weight $w_e > 0$, which represents the length of $e$. The tree $\mathcal{T}$ is naturally equipped with a metric $d_{\mathcal{T}} : V \times V \to \mathbb{R}_+$, such that for $x, y \in V$, $d_{\mathcal{T}}(x, y) = \sum_{e \in [x,y]} w_e$ is the length of the unique path connecting $x$ and $y$. Furthermore, given two vertices $x, z \in V$, we let $[x, z]$ denote the path connecting $x$ and $z$. Moreover, for each edge $e \in E$, we denote by $v_e$ the endpoint of $e$ that is farther from the root of $\mathcal{T}$, and by $\gamma_e$ the subtree of $\mathcal{T}$ rooted at $v_e$.

**Measures and functions.** Let $\mathcal{P}(\mathcal{T})$ denote the set of all nonnegative Borel measures on a graph $\mathcal{T}$, and let $\mathcal{P}(\mathcal{T} \times \mathcal{T})$ denote the corresponding set of measures on the product space $\mathcal{T} \times \mathcal{T}$ with finite mass. A function $f : \mathcal{T} \to \mathbb{R}$ is said to be continuous if it is continuous with respect to the topology on $\mathcal{T}$ induced by the Euclidean distance. We write $C(\mathcal{T})$ for the space of all continuous functions on $\mathcal{T}$, and analogously $C(\mathcal{T} \times \mathcal{T})$ for continuous functions on $\mathcal{T} \times \mathcal{T}$. Given a nonnegative Borel measure $\omega$ on $\mathcal{T}$ and an exponent $1 \le p < \infty$, we define the space $L^p(\mathcal{T}, \omega)$ as

$$L^p(\mathcal{T}, \omega) := \left\{ f : \mathcal{T} \to \mathbb{R} \, \Big| \, \int_{\mathcal{T}} |f(x)|^p \, \omega(dx) < \infty \right\}. \tag{16}$$

This is a normed space equipped with the norm $\|f\|_{L^p} = \left( \int_{\mathcal{T}} |f(x)|^p \, \omega(dx) \right)^{1/p}$.

In addition, let $\hat{w} : \mathcal{T} \to \mathbb{R}_+$ be a strictly positive weight function, i.e., $\hat{w}(x) > 0$ for every $x \in G$. The associated weighted $L^p$ space, denoted $L^p_{\hat{w}}(G, \omega)$, is given by

$$L^p_{\hat{w}}(\mathcal{T}, \omega) := \left\{ f : \mathcal{T} \to \mathbb{R} \, \Big| \, \int_{\mathcal{T}} \hat{w}(x) \, |f(x)|^p \, \omega(dx) < \infty \right\}. \tag{17}$$

## B.2 SOBOLEV IPM FOR PROBABILITY MEASURE ON TREE

Following the definition of graph-based Sobolev spaces Le et al. (2022), we define the tree-based Sobolev space as follows.

**Definition B.1** (Tree-based Sobolev). Let $\omega$ be a nonnegative Borel measure on the tree $\mathcal{T}$, and let $1 \le p \le \infty$. A continuous function $f : \mathcal{T} \to \mathbb{R}$ is said to belong to the Sobolev space $W^{1,p}(\mathcal{T}, \omega)$ if there exists a function $h \in L^p(\mathcal{T}, \omega)$ such that

$$f(x) - f(z_0) = \int_{[z_0, x]} h(y) \, \omega(dy), \quad \forall x \in \mathcal{T}, \tag{18}$$

where $z_0$ is a fixed reference vertex in $\mathcal{T}$. The function $h$ is uniquely determined in $L^p(\mathcal{T}, \omega)$ and is referred to as the tree derivative of $f$ with respect to the measure $\omega$. We denote the tree derivative of $f \in W^{1,p}(\mathcal{T}, \omega)$ by $f' \in L^p(\mathcal{T}, \omega)$.

The Sobolev space $W^{1,p}(\mathcal{T}, \omega)$ is endowed with the norm

$$\|f\|_{W^{1,p}} = \left( \|f\|^p_{L^p(\mathcal{T}, \omega)} + \|f'\|^p_{L^p(\mathcal{T}, \omega)} \right)^{1/p}, \tag{19}$$

which is referred to as the Sobolev norm (Adams & Fournier, 2003) turning $W^{1,p}(\mathcal{T}, \omega)$ into a normed space. We further denote the subspace $W^{1,p}_0(\mathcal{T}, \omega) = \{f \in W^{1,p}(\mathcal{T}, \omega) \, : \, f(z_0) = 0\}$, where $z_0$ is the root of $\mathcal{T}$. The unit ball in this space is then denoted by

$$\mathcal{B}(p) := \{f \in W^{1,p}_0(\mathcal{T}, \omega) \, : \, \|f\|_{W^{1,p}} \le 1\}. \tag{20}$$

**Definition B.2** (Tree-based Sobolev IPM). Let $\omega$ be a nonnegative Borel measure on $\mathcal{T}$, and let $1 \le p \le \infty$ with conjugate exponent $p'$ defined by $\frac{1}{p} + \frac{1}{p'} = 1$ (with the convention $p' = \infty$ when $p = 1$). For two probability measures $\mu, \nu \in \mathcal{P}(\mathcal{T})$, the Sobolev IPM is defined as

$$\mathcal{S}_p(\mu, \nu) = \sup_{f \in \mathcal{B}(p')} \left| \int_{\mathcal{T}} f(x) \, \mu(dx) - \int_{\mathcal{T}} f(y) \, \nu(dy) \right|, \tag{21}$$

where $\mathcal{B}(p')$ denotes the unit ball in $W^{1,p'}_0(\mathcal{T}, \omega)$.

Overall, the Sobolev IPM for probability measures on a graph can be viewed as a particular case of the IPM, where the witness critic functions are restricted to the graph-based Sobolev space and further constrained to lie within its unit ball. Furthermore, Notice that the quantity inside the absolute signs is unchanged if $f$ is replaced by $f - f(z_0)$. Thus, we can assume without loss of generality that $f(z_0) = 0$. This is the motivation for $W_0^{1,p}(\mathcal{T}, \omega)$. Next, we introduce a weight function. Let $\omega$ be measure $\omega$ on a set. We have

$$\hat{w}(x) := 1 + \omega(\Lambda(x)), \quad \forall x \in \mathcal{T}, \tag{22}$$

An example of $\omega$ is when it is chosen as the length measure (Le et al., 2022), in which case $\omega(\Lambda(x))$ corresponds to the total length of the subtree $\Lambda(x)$. We now present a key theorem establishing that, for any critic function $f \in W_0^{1,p}(\mathcal{T}, \omega)$, the Sobolev norm is equivalent to the weighted $L^p$-norm of its derivative $f'$.

**Theorem B.3.** *Let $\omega$ be a nonnegative Borel measure on $\mathcal{T}$ and let $1 \leq p < \infty$. Define the constants*

$$a_1 := \left( \frac{\min\{1, \omega(\mathcal{T})^{p-1}\}}{1 + \omega(\mathcal{T})^p} \right)^{\frac{1}{p}}, \qquad a_2 := \left( \max\{1, \omega(\mathcal{T})^{p-1}\} \right)^{\frac{1}{p}}. \tag{23}$$

*Then, for every $f \in W_0^{1,p}(\mathcal{T}, \omega)$, the following norm equivalence holds:*

$$a_1 \|f'\|_{L^p_{\hat{w}}} \leq \|f\|_{W^{1,p}} \leq a_2 \|f'\|_{L^p_{\hat{w}}}. \tag{24}$$

Proof of Theorem B.3 is defered to Section D.4.

### B.3 REGULARIZED SOBOLEV IPM FOR PROBABILITY MEASURE ON TREE

Having established the equivalence relation in Theorem B.3, we now introduce the regularized Sobolev IPM. Specifically, rather than constraining the critic $f$ to lie in the unit ball $\mathcal{B}(p')$ of the Sobolev space, we instead restrict $f$ to the unit ball $\mathcal{B}(p', \hat{w})$, defined with respect to the weighted $L^{p'}$-norm of its derivative $f'$ under the weight function $\hat{w}$. Hereafter, we define $\mathcal{B}(p', \hat{w})$ as

$$\mathcal{B}(p', \hat{w}) := \left\{ f \in W_0^{1,p'}(\mathcal{T}, \omega) : \|f'\|_{L^{p'}_{\hat{w}}} \leq 1 \right\}. \tag{25}$$

We now formally define the regularized Sobolev IPM between two probability distributions on tree $\mathcal{T}$

**Definition B.4** (Tree-base Regularized Sobolev IPM)**.** Let $\omega$ be a nonnegative Borel measure on $\mathcal{T}$ and let $1 \leq p \leq \infty$. For any probability measures $\mu, \nu \in \mathcal{P}(\mathcal{T})$, the regularized Sobolev IPM is defined by

$$\hat{\mathcal{S}}_p(\mu, \nu) := \sup_{f \in \mathcal{B}(p', \hat{w})} \left| \int_{\mathcal{T}} f(x)\mu(dx) - \int_{\mathcal{T}} f(y)\nu(dy) \right|, \tag{26}$$

where $\mathcal{B}(p', \hat{w})$ denotes the unit ball in the weighted Sobolev space induced by the norm $|f'|_{L^{p'}\hat{w}}$.

Next, we show that the tree-based Sobolev IPM has a closed-form solution that is as follow

**Theorem B.5** (Closed-form Expression)**.** *Let $\omega$ be a nonnegative Borel measure on $\mathcal{T}$, and let $1 \leq p < \infty$. Then, for any probability measures $\mu, \nu \in \mathcal{P}(\mathcal{T})$, the regularized Sobolev IPM admits the closed-form expression*

$$\hat{\mathcal{S}}_p(\mu, \nu)^p = \int_{\mathcal{T}} \hat{w}(x)^{1-p} \left| \mu(\Lambda(x)) - \nu(\Lambda(x)) \right|^p \omega(dx), \tag{27}$$

*where $\Lambda(x)$ denotes the subtree rooted at $x$.*

Proof of Theorem B.5 is defered to Section D.5. Additionally, when the input probability measures are supported on nodes V of $\mathcal{T}$ and we choose the length measure on tree $\mathcal{T}$ for the nonnegative Borel measure $\omega$, we can derive an explicit formula for Equation (26) as follow

**Theorem B.6** (Explicit formula for Discrete Case)**.** *Let $\omega$ denote the length measure on $\mathcal{T}$, and let $1 \leq p < \infty$. Suppose that $\mu, \nu \in \mathcal{P}(\mathcal{T})$ are supported on the vertex set $V$ of the tree $\mathcal{T}$. Then the regularized Sobolev IPM admits the closed-form expression*

$$\hat{\mathcal{S}}_p(\mu, \nu) = \left( \sum_{e \in E} \beta_e \left| \mu(\gamma_e) - \nu(\gamma_e) \right|^p \right)^{1/p}, \tag{28}$$

*where for each edge $e \in E$, the scalar weight $\beta_e$ is given by*

$$\beta_e := \begin{cases} \log\left(1 + \dfrac{w_e}{1 + \omega(\gamma_e)}\right), & \text{if } p = 2, \\ \dfrac{(1 + \omega(\gamma_e) + w_e)^{2-p} - (1 + \omega(\gamma_e))^{2-p}}{2 - p}, & \text{otherwise}, \end{cases} \tag{29}$$

*with $w_e$ denoting the length of edge $e$ and $\gamma_e$ the subtree rooted at the endpoint $v_e$, which is the endpoint of edge $e$ farther from the root.*

Proof of Theorem B.6 is defered to Section D.6.

**Implementation of Explicit Form for Discrete Case.** The regularized Sobolev IPM $\hat{\mathcal{S}}_p$ depends only on the graph structure $(V, E)$ and edge weights $\{w_e\}_{e \in E}$, and can therefore be applied beyond physical graphs. For efficient computation, the sets $\gamma_e$ and coefficients $\beta_e$ (cf. Equations (10)–(11)) can be precomputed once from the root $z_0$ to all vertices.

## C BACKGROUND ON TREE-SLICED WASSERSTEIN DISTANCE ON EUCLIDEAN SPACES

This section revisits the fundamental components of the Tree-Sliced Wasserstein (TSW) distance, formulated over tree systems embedded in Euclidean spaces. For completeness, we summarize key definitions and core mathematical formulations. The reader is referred to (Tran et al., 2024d; 2025a) for detailed proofs and a detailed explanation.

### C.1 TREE SYSTEM

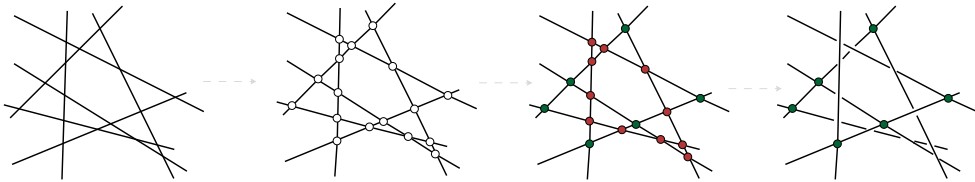

Figure 2: The construction of the tree system is illustrated in the two-dimensional plane $\mathbb{R}^2$, though the approach naturally extends to higher dimensions. The procedure begins with a set of infinite lines placed without any predefined arrangement. All pairwise intersections of these lines are determined, though some may lie outside the visible region of the figure due to their unbounded extent. Among these intersections, a subset is marked in red to indicate those that will be discarded. The remaining intersections in green are preserved in order to impose a tree structure on the system—ensuring that any two points along the lines are linked by a unique path passing only through the retained intersections. These preserved points serve as the fundamental nodes defining the tree topology. Once the red intersections are removed, the resulting network forms the desired tree system.

**Components of Tree Systems.** A line in $\mathbb{R}^d$ is an element $l = (x, \theta) \in \mathbb{R}^d \times \mathbb{S}^{d-1}$, where $x$ is the *source* and $\theta$ is the *direction*. It is parameterized by $x + t\,\theta$ for $t \in \mathbb{R}$.

Given an integer $k \geq 1$, a *system of $k$ lines in $\mathbb{R}^d$* refers to a collection of $k$ such lines. The notation $(\mathbb{R}^d \times \mathbb{S}^{d-1})^k$ is abbreviated as $\mathbb{L}_k^d$, representing the *space of systems of $k$ lines in $\mathbb{R}^d$*. An element in this space, commonly denoted by $\mathcal{L}$, corresponds to a specific system of lines, written as $\mathcal{L} = \{l_i\}_{i=1}^k$ where each $l_i = (x_i, \theta_i) \in \mathbb{R}^d \times \mathbb{S}^{d-1}$ and $i$ indexes the lines.

A line system $\mathcal{L}$ is connected if the union of points lying on its individual lines is a connected subset of $\mathbb{R}^d$. A *tree structure* can be imposed by removing selected intersection points so that any two points in the resulting configuration are joined by a unique simple path. The resulting object is a *tree system*, denoted $\mathcal{T} = \{l_i\}_{i=1}^k$. We use the term *tree system* to emphasize this unique-path property, in direct analogy with trees in graph theory. Using remaining intersections, we build a topological tree system by coherently gluing segments of $\mathbb{R}$ via disjoint union and quotient topology (Hatcher, 2005), resulting in a space endowed with a valid tree metric. An illustration of the construction appears in Figure 2.

**Sampling Tree Systems.** Tree system spaces admit diverse structures, but Tran et al. (2024d) highlight *chain-structured* variants. A generative model for such systems is as follows: sample an initial point $x_1 \sim \mu_1$ and a direction $\theta_1 \sim \nu_1$; then for each $i > 1$, sample an offset $t_i \sim \mu_i$ and a direction $\theta_i \sim \nu_i$, and set $x_i = x_{i-1} + t_i\theta_{i-1}$. Each $\mu_i$ and $\nu_i$ is an independent distribution. In practice, we take $\mu_1 = \mathcal{U}([-1,1]^d)$, $\mu_i = \mathcal{U}([-1,1])$ for $i > 1$, and $\nu_i = \mathcal{U}(\mathbb{S}^{d-1})$ for all $i$.

In Tran et al. (2025a), a *concurrent-line* tree structure is introduced, where all lines $\{l_i\}_{i=1}^k$ share the same source point $x$. The corresponding generative model is simpler: first sample the common root $x \sim \mu$, then independently sample $\theta_i \sim \nu$ for each $i = 1, \ldots, k$. Here $\mu = \mathcal{U}([-1,1]^d)$ specifies the distribution of the root, and $\nu = \mathcal{U}(\mathbb{S}^{d-1})$ is the common distribution over directions.

A visualization of the two tree structures is provided in Figure 3.

**Remark C.1.** Recent advances in Tree-Sliced Wasserstein distance (Tran et al., 2025a;d) employ the *concurrent-line* tree structure. For TS-Sobolev, we likewise adopt this structure for its simplicity.

**Remark C.2.** The chain-structured and concurrent sampling schemes each induce a probability measure $\sigma$ over the space $\mathbb{T}$ of tree systems.

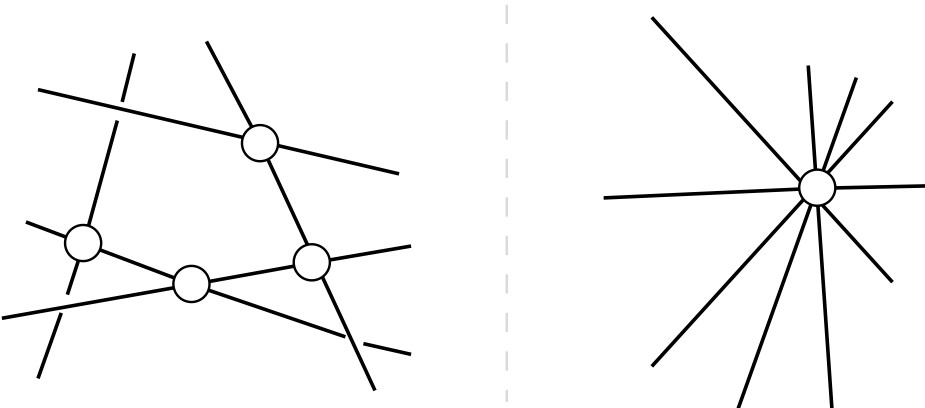

Figure 3: Visualizations of two popular tree structures: a chain structure (left) and a concurrent-lines structure (right).

## C.2 RADON TRANSFORM ON TREE SYSTEMS

Let $L^1(\mathbb{R}^d)$ be the space of Lebesgue–integrable functions on $\mathbb{R}^d$, equipped with the norm $\|\cdot\|_1$. Fix a tree system $\mathcal{T} = \{l_i\}_{i=1}^k \in \mathbb{T}_k^d$ with lines $l_i = (x_i, \theta_i)$, and let $\bar{\mathcal{T}}$ denote the union of all points on these lines. A function $f : \bar{\mathcal{T}} \to \mathbb{R}$ is *integrable over $\mathcal{T}$* if

$$\|f\|_{\mathcal{T}} := \sum_{i=1}^k \int_{\mathbb{R}} |f(t_x, l_i)| \, dt_x < \infty, \tag{30}$$

and the collection of such functions is denoted $L^1(\mathcal{T})$, the space of Lebesgue integrable functions over the tree system $\mathcal{T}$.

The standard $(k-1)$–simplex is

$$\Delta_{k-1} = \left\{ a = (a_1, \ldots, a_k) \in \mathbb{R}^k \;\middle|\; a_i \geq 0, \; \sum_{i=1}^k a_i = 1 \right\}. \tag{31}$$

Write $\mathcal{C}(\mathbb{R}^d \times \mathbb{T}_k^d, \Delta_{k-1})$ for the set of continuous *splitting maps* $\alpha : \mathbb{R}^d \times \mathbb{T}_k^d \to \Delta_{k-1}$.

Given $f \in L^1(\mathbb{R}^d)$ and $\alpha \in \mathcal{C}(\mathbb{R}^d \times \mathbb{T}_k^d, \Delta_{k-1})$, we define the projection operator that maps $f \in L^1(\mathbb{R}^d)$ to a function on $\bar{\mathcal{T}}$:

$$\mathcal{R}_{\mathcal{T}}^\alpha f(x, l_i) = \int_{\mathbb{R}^d} f(y) \, \alpha(y, \mathcal{T})_i \, \delta\big(t_x - \langle y - x_i, \theta_i \rangle\big) \, dy, \tag{32}$$

where $(x_i, \theta_i)$ specifies line $l_i$ and $\delta$ is the Dirac delta. According to Tran et al. (2025a), $\mathcal{R}_{\mathcal{T}}^\alpha f \in L^1(\mathcal{T})$ and $\|\mathcal{R}_{\mathcal{T}}^\alpha f\|_{\mathcal{T}} \le \|f\|_1$, hence $\mathcal{R}_{\mathcal{T}}^\alpha : L^1(\mathbb{R}^d) \to L^1(\mathcal{T})$ is a well-defined operator.

Aggregating over all tree systems $\mathcal{T} \in \mathbb{T}_k^d$, we define the *Radon transform on tree systems* by

$$\mathcal{R}^\alpha : L^1(\mathbb{R}^d) \longrightarrow \prod_{\mathcal{T} \in \mathbb{T}_k^d} L^1(\mathcal{T}), \tag{33}$$

$$f \longmapsto \left(\mathcal{R}_{\mathcal{T}}^\alpha f\right)_{\mathcal{T} \in \mathbb{T}_k^d}. \tag{34}$$

If $\alpha$ is invariant under the Euclidean group $\mathrm{E}(d)$, then $\mathcal{R}^\alpha$ is injective.

### C.3 TREE-SLICED WASSERSTEIN DISTANCE FOR PROBABILITY MEASURES ON EUCLIDEAN SPACES

Consider probability measures $\mu, \nu \in \mathcal{P}(\mathbb{R}^d)$. Given a tree-structured collection of lines $\mathcal{T} \in \mathbb{T}$ and an $\mathrm{E}(d)$-invariant splitting map $\alpha \in \mathcal{C}(\mathbb{R}^d \times \mathbb{L}_k^d, \Delta_{k-1})$, the operator $\mathcal{R}_{\mathcal{T}}^\alpha$ transports $\mu$ and $\nu$ onto corresponding measures $\mathcal{R}_{\mathcal{T}}^\alpha \mu$ and $\mathcal{R}_{\mathcal{T}}^\alpha \nu$ supported on $\mathcal{T}$. Since each tree system $\mathcal{T}$ is endowed with a tree metric $d_{\mathcal{T}}$, one can evaluate the 1-Wasserstein distance $\mathrm{W}_{d_{\mathcal{T}},1}$ between these transformed measures. This motivates the definition of the *Distance-based Tree-Sliced Wasserstein* (Db-TSW) distance Tran et al. (2025a):

$$\text{Db-TSW}(\mu, \nu) := \int_{\mathbb{T}} \mathrm{W}_1(\mathcal{R}_{\mathcal{T}}^\alpha \mu, \mathcal{R}_{\mathcal{T}}^\alpha \nu) \, d\sigma(\mathcal{T}), \tag{35}$$

where $\sigma$ denotes a probability distribution over the space of tree systems $\mathbb{T}$. The value of Db-TSW depends on the choice of tree space $\mathbb{T}$, the sampling process that induces $\sigma$, and the $\mathrm{E}(d)$-invariant map $\alpha$, although this dependence is suppressed in the notation for simplicity. The resulting Db-TSW provides an $\mathrm{E}(d)$-invariant metric on $\mathcal{P}(\mathbb{R}^d)$.

**Remark C.3.** When the tree systems consist of a single line, the Db-TSW distance reduces to the standard Sliced Wasserstein distance.

**Splitting Maps.** Since $\mathcal{R}^\alpha$ is injective whenever the splitting map $\alpha$ is $\mathrm{E}(d)$-invariant, we seek constructions that satisfy this property. For $x \in \mathbb{R}^d$ and a tree system $\mathcal{T} = \{l_i\}_{i=1}^k \in \mathbb{T}_k^d$, define the Euclidean distance from $x$ to the line $l_i \in \mathcal{T}$ by

$$d(x, \mathcal{T})_i = \inf_{y \in l_i} \|x - y\|_2, \tag{36}$$

which is invariant under the Euclidean group $\mathrm{E}(d)$. Any splitting rule that depends only on the collection $\{d(x, \mathcal{T})_i\}_{i=1}^k$ therefore inherits $\mathrm{E}(d)$-invariance. A practical and widely used choice is the softmax:

$$\alpha(x, \mathcal{T})_i = \text{softmax}\left(\{-\xi \cdot d(x, \mathcal{T})_j\}_{j=1}^k\right)_i, \tag{37}$$

where the parameter $\xi > 0$ controls the sharpness of the distribution. This map assigns weights to the lines in $\mathcal{T}$ according to their proximity to $x$, while preserving Euclidean symmetries. Empirically, softmax-based splitting maps have been found to perform well in applications (Tran et al., 2025a).

**Remark C.4.** Equivariant neural networks (Cohen & Welling, 2016) explicitly encode symmetry priors of a task directly into the architecture, thereby promoting better out-of-sample generalization and higher sample efficiency. They have achieved substantial empirical success across diverse areas, including trajectory forecasting (Walters et al., 2020), robotics (Simeonov et al., 2022), graph representation learning (Satorras et al., 2021; Tran et al., 2024b), Optimal Transport–driven frameworks (Tran et al., 2024e; 2025e), equivariant metanetwork design (Tran et al., 2026a; Vo et al., 2025; Tran et al., 2024a; 2025b), and analyses conducted in weight space (Tran et al., 2025f).

## D THEORETICAL PROOFS

In this section, we prove all results for TS-Sobolev stated in the main text and establish the properties of Sobolev IPM introduced in Appendix B.

### D.1 PROOF FOR THEOREM 3.2

*Proof.* We show that TS-Sobolev$_p$ is an $\mathrm{E}(d)$-invariant metric on the space of probability measures $\mathcal{P}(\mathbb{R}^d)$. The definition is given by

$$\text{TS-Sobolev}_p(\mu, \nu) = \left( \int_{\mathbb{T}} \hat{\mathcal{S}}_p(\mu_{\mathcal{T}}, \nu_{\mathcal{T}})^p \, d\sigma(\mathcal{T}) \right)^{\frac{1}{p}}, \tag{38}$$

where $\hat{\mathcal{S}}_p$ is the regularized Sobolev IPM, which is a metric on the space of measures on a tree $\mathcal{T}$. The proof relies on the injectivity of the Radon Transform $\mathcal{R}^\alpha$, which holds because the splitting map $\alpha$ is chosen to be $\mathrm{E}(d)$-invariant (see Tran et al. (2025a)).

We now verify the three metric axioms.

**Positive definiteness.** It is clear that $\text{TS-Sobolev}_p(\mu, \mu) = 0$ and $\text{TS-Sobolev}_p(\mu, \nu) \geq 0$. If $\text{TS-Sobolev}_p(\mu, \nu) = 0$, this implies $\int_{\mathbb{T}} \hat{\mathcal{S}}_p(\mu_{\mathcal{T}}, \nu_{\mathcal{T}})^p \, d\sigma(\mathcal{T}) = 0$. Since the integrand is non-negative, this means $\hat{\mathcal{S}}_p(\mu_{\mathcal{T}}, \nu_{\mathcal{T}}) = 0$ for almost all $\mathcal{T} \in \mathbb{T}$. As $\hat{\mathcal{S}}_p$ is a metric, it follows that $\mu_{\mathcal{T}} = \nu_{\mathcal{T}}$ for almost all $\mathcal{T}$. By the injectivity of $\mathcal{R}^\alpha$, we conclude that the densities are equal, $f_\mu = f_\nu$, and thus $\mu = \nu$.

**Symmetry.** The symmetry of $\hat{\mathcal{S}}_p$ on each tree implies $\hat{\mathcal{S}}_p(\mu_{\mathcal{T}}, \nu_{\mathcal{T}})^p = \hat{\mathcal{S}}_p(\nu_{\mathcal{T}}, \mu_{\mathcal{T}})^p$. Therefore,

$$\begin{aligned}
\text{TS-Sobolev}_p(\mu, \nu) &= \left( \int_{\mathbb{T}} \hat{\mathcal{S}}_p(\mu_{\mathcal{T}}, \nu_{\mathcal{T}})^p \, d\sigma(\mathcal{T}) \right)^{\frac{1}{p}} \\
&= \left( \int_{\mathbb{T}} \hat{\mathcal{S}}_p(\nu_{\mathcal{T}}, \mu_{\mathcal{T}})^p \, d\sigma(\mathcal{T}) \right)^{\frac{1}{p}} = \text{TS-Sobolev}_p(\nu, \mu). \tag{39}
\end{aligned}$$

**Triangle inequality.** For any $\mu_1, \mu_2, \mu_3 \in \mathcal{P}(\mathbb{R}^d)$, we use the triangle inequality of $\hat{\mathcal{S}}_p$ on each tree, which states $\hat{\mathcal{S}}_p(\mu_{1,\mathcal{T}}, \mu_{3,\mathcal{T}}) \leq \hat{\mathcal{S}}_p(\mu_{1,\mathcal{T}}, \mu_{2,\mathcal{T}}) + \hat{\mathcal{S}}_p(\mu_{2,\mathcal{T}}, \mu_{3,\mathcal{T}})$. We then apply Minkowski's integral inequality:

$$\begin{aligned}
\text{TS-Sobolev}_p(\mu_1, \mu_3) &= \left( \int_{\mathbb{T}} \hat{\mathcal{S}}_p(\mu_{1,\mathcal{T}}, \mu_{3,\mathcal{T}})^p \, d\sigma(\mathcal{T}) \right)^{\frac{1}{p}} \\
&\leq \left( \int_{\mathbb{T}} \left( \hat{\mathcal{S}}_p(\mu_{1,\mathcal{T}}, \mu_{2,\mathcal{T}}) + \hat{\mathcal{S}}_p(\mu_{2,\mathcal{T}}, \mu_{3,\mathcal{T}}) \right)^p \, d\sigma(\mathcal{T}) \right)^{\frac{1}{p}} \\
&\leq \left( \int_{\mathbb{T}} \hat{\mathcal{S}}_p(\mu_{1,\mathcal{T}}, \mu_{2,\mathcal{T}})^p \, d\sigma(\mathcal{T}) \right)^{\frac{1}{p}} + \left( \int_{\mathbb{T}} \hat{\mathcal{S}}_p(\mu_{2,\mathcal{T}}, \mu_{3,\mathcal{T}})^p \, d\sigma(\mathcal{T}) \right)^{\frac{1}{p}} \\
&= \text{TS-Sobolev}_p(\mu_1, \mu_2) + \text{TS-Sobolev}_p(\mu_2, \mu_3). \tag{40}
\end{aligned}$$

Thus, TS-Sobolev$_p$ is a metric on $\mathcal{P}(\mathbb{R}^d)$.

$\mathrm{E}(d)$**-invariance.** We aim to show that for any $g \in \mathrm{E}(d)$, $\text{TS-Sobolev}_p(\mu, \nu) = \text{TS-Sobolev}_p(g\sharp\mu, g\sharp\nu)$. Let $\mathcal{T} = \{l_i = (x_i, \theta_i)\}_{i=1}^k$ be a tree system. Under the action of $g = (Q, v)$, we have $g\mathcal{T} = \{gl_i = (Qx_i + v, Q\theta_i)\}_{i=1}^k$. Since $|\det(Q)| = 1$ and $\alpha$ is $\mathrm{E}(d)$-invariant, we compute for a line $l_i \in \mathcal{T}$:

$$\begin{aligned}
&\mathcal{R}^\alpha_{g\mathcal{T}}(g\sharp f_\mu)(gx, gl_i) \\
&= \int_{\mathbb{R}^d} (g\sharp f_\mu)(y) \, \alpha(y, g\mathcal{T})_i \, \delta\big(t_{gx} - \langle y - x_{gl_i}, \theta_{gl_i} \rangle\big) \, dy \\
&= \int_{\mathbb{R}^d} f_\mu(g^{-1}y) \, \alpha(y, g\mathcal{T})_i \, \delta\big(t_x - \langle y - x_{gl_i}, \theta_{gl_i} \rangle\big) \, dy \\
&= \int_{\mathbb{R}^d} f_\mu(g^{-1}gy) \, \alpha(gy, g\mathcal{T})_i \, \delta\big(t_x - \langle gy - x_{gl_i}, \theta_{gl_i} \rangle\big) \, d(gy) \\
&= \int_{\mathbb{R}^d} f_\mu(y) \, \alpha(y, \mathcal{T})_i \, \delta\big(t_x - \langle gy - x_{gl_i}, \theta_{gl_i} \rangle\big) \, dy
\end{aligned}$$

$$
\begin{aligned}
&= \int_{\mathbb{R}^d} f_\mu(y)\, \alpha(y,\mathcal{T})_i\, \delta\big(t_x - \langle Qy + v - (Qx_i + v),\, Q\theta_i\rangle\big)\, dy \\
&= \int_{\mathbb{R}^d} f_\mu(y)\, \alpha(y,\mathcal{T})_i\, \delta\big(t_x - \langle Q(y - x_i),\, Q\theta_i\rangle\big)\, dy \\
&= \int_{\mathbb{R}^d} f_\mu(y)\, \alpha(y,\mathcal{T})_i\, \delta\big(t_x - \langle y - x_i,\, \theta_i\rangle\big)\, dy \\
&= \mathcal{R}^\alpha_\mathcal{T} f_\mu(x, l_i).
\end{aligned}
\tag{41}
$$

This implies that the action of $g$ is an isometry, so $\hat{\mathcal{S}}_p(\mu_\mathcal{T}, \nu_\mathcal{T}) = \hat{\mathcal{S}}_p((g\sharp\mu)_{g\mathcal{T}}, (g\sharp\nu)_{g\mathcal{T}})$. Using this, we compute:

$$
\begin{aligned}
\text{TS-Sobolev}_p(g\sharp\mu, g\sharp\nu)^p &= \int_{\mathbb{T}} \hat{\mathcal{S}}_p((g\sharp\mu)_\mathcal{T}, (g\sharp\nu)_\mathcal{T})^p\, d\sigma(\mathcal{T}) \\
&= \int_{\mathbb{T}} \hat{\mathcal{S}}_p((g\sharp\mu)_{g\mathcal{T}}, (g\sharp\nu)_{g\mathcal{T}})^p\, d\sigma(g\mathcal{T}) \\
&= \int_{\mathbb{T}} \hat{\mathcal{S}}_p(\mu_\mathcal{T}, \nu_\mathcal{T})^p\, d\sigma(\mathcal{T}) = \text{TS-Sobolev}_p(\mu,\nu)^p.
\end{aligned}
\tag{42}
$$

Taking the $p$-th root of both sides, we conclude that TS-Sobolev$_p$ is E$(d)$-invariant. $\qquad\square$

**Remark D.1.** For clarity, we omit the almost-sure conditions in the proof. Verifying these conditions is straightforward, and their inclusion would make the core argument harder to follow.

### D.2 Proof for Theorem 3.3

*Proof.* We prove the theorem in two parts. First, we establish the equality for the case $p = 1$. Second, we prove the general inequality for any $p \in [1, \infty)$.

**Part 1.** We first proof equality for $p = 1$. By definition, the Tree-Sliced Sobolev IPM is given by:

$$
\text{TS-Sobolev}_p(\mu, \nu) = \left( \int_{\mathbb{T}} \hat{\mathcal{S}}_p(\mu_\mathcal{T}, \nu_\mathcal{T})^p\, d\sigma(\mathcal{T}) \right)^{\frac{1}{p}}.
\tag{43}
$$

For the case $p = 1$, this definition simplifies to the expectation of the base metric:

$$
\text{TS-Sobolev}_1(\mu, \nu) = \int_{\mathbb{T}} \hat{\mathcal{S}}_1(\mu_\mathcal{T}, \nu_\mathcal{T})\, d\sigma(\mathcal{T}).
\tag{44}
$$

The Tree-Sliced Wasserstein distance is defined as $\text{TSW}(\mu, \nu) = \int_{\mathbb{T}} \text{W}_1(\mu_\mathcal{T}, \nu_\mathcal{T})\, d\sigma(\mathcal{T})$. To prove the equality, it is sufficient to show the integrands are equal, i.e., $\hat{\mathcal{S}}_1(\mu_\mathcal{T}, \nu_\mathcal{T}) = \text{W}_1(\mu_\mathcal{T}, \nu_\mathcal{T})$.

We analyze the discrete form of the Regularized Sobolev IPM from Equation (5) for $p = 1$:

$$
\hat{\mathcal{S}}_1(\mu_\mathcal{T}, \nu_\mathcal{T}) = \sum_{e \in E} \beta_e\, |\mu(\gamma_e) - \nu(\gamma_e)|.
\tag{45}
$$

From Equation (6), the coefficient $\beta_e$ simplifies to $\beta_e = w_e$ for $p = 1$. Substituting this result gives:

$$
\hat{\mathcal{S}}_1(\mu_\mathcal{T}, \nu_\mathcal{T}) = \sum_{e \in E} w_e\, |\mu(\gamma_e) - \nu(\gamma_e)|.
\tag{46}
$$

This expression is the known closed-form solution for the 1-Wasserstein distance on a tree, $\text{W}_1(\mu_\mathcal{T}, \nu_\mathcal{T})$ (Le et al., 2019). Since the integrands are equal, their expectations are equal, which proves that TS-Sobolev$_1(\mu, \nu) = \text{TSW}(\mu, \nu)$.

**Part 2.** We now prove the general inequality TS-Sobolev$_p(\mu, \nu)^p \leq \text{TSW}(\mu, \nu)$. It is sufficient to show that the integrand of the first expression is bounded by the integrand of the second on any given tree $\mathcal{T}$. That is, we will prove $\hat{\mathcal{S}}_p(\mu_\mathcal{T}, \nu_\mathcal{T})^p \leq \text{W}_1(\mu_\mathcal{T}, \nu_\mathcal{T})$.

We first establish that the Sobolev coefficient $\beta_e \leq w_e$ for all $p \geq 1$ by a case analysis.

**Case 1:** $p = 2$. The coefficient is $\beta_e = \log\left(1 + \frac{w_e}{1+\omega(\gamma_e)}\right)$. Using the inequality $\log(1+x) \leq x$ for $x \geq 0$, we have:

$$\beta_e = \log\left(1 + \frac{w_e}{1+\omega(\gamma_e)}\right) \leq \frac{w_e}{1+\omega(\gamma_e)}. \tag{47}$$

Since $\omega(\gamma_e) \geq 0$, the denominator is at least 1, which implies $\frac{w_e}{1+\omega(\gamma_e)} \leq w_e$. Thus, $\beta_e \leq w_e$.

**Case 2:** $p \geq 1$ **and** $p \neq 2$. We apply the Mean Value Theorem. Let the function be $f(x) = x^{2-p}$ and consider the interval $[a, b]$ where $a = 1 + \omega(\gamma_e)$ and $b = 1 + \omega(\gamma_e) + w_e$. The theorem states there is a value $c \in (a, b)$ such that $f(b) - f(a) = f'(c)(b-a)$. The derivative is $f'(c) = (2-p)c^{1-p}$. Substituting this into the definition of $\beta_e$:

$$\beta_e = \frac{(1+\omega(\gamma_e)+w_e)^{2-p} - (1+\omega(\gamma_e))^{2-p}}{2-p}$$

$$= \frac{f(b) - f(a)}{2-p} = \frac{f'(c)(b-a)}{2-p} = \frac{(2-p)c^{1-p} \cdot w_e}{2-p} = w_e \cdot c^{1-p}. \tag{48}$$

Since $a = 1 + \omega(\gamma_e) \geq 1$, the intermediate value $c$ must be greater than 1. For any $p \geq 1$, the exponent $1 - p$ is non-positive ($\leq 0$), which ensures $c^{1-p} \leq 1$ and therefore $\beta_e \leq w_e$.

Now, using the universal bound $\beta_e \leq w_e$ and the fact that $|\mu(\gamma_e) - \nu(\gamma_e)| \in [0, 1]$, which implies $|\dots|^p \leq |\dots|$ for $p \geq 1$, we can bound the $p$-th power of the Sobolev IPM:

$$\begin{aligned}
\hat{\mathcal{S}}_p(\mu_{\mathcal{T}}, \nu_{\mathcal{T}})^p &= \sum_{e \in E} \beta_e \, |\mu(\gamma_e) - \nu(\gamma_e)|^p \\
&\leq \sum_{e \in E} w_e \, |\mu(\gamma_e) - \nu(\gamma_e)|^p \qquad \text{(since } \beta_e \leq w_e\text{)} \\
&\leq \sum_{e \in E} w_e \, |\mu(\gamma_e) - \nu(\gamma_e)| \qquad \text{(since } |\dots|^p \leq |\dots|\text{)} \\
&= W_1(\mu_{\mathcal{T}}, \nu_{\mathcal{T}}). \qquad \text{(by Part 1 of this proof)} \tag{49}
\end{aligned}$$

This establishes the key inequality on a single tree: $\hat{\mathcal{S}}_p(\mu_{\mathcal{T}}, \nu_{\mathcal{T}})^p \leq W_1(\mu_{\mathcal{T}}, \nu_{\mathcal{T}})$. Integrating this inequality over all trees $\mathcal{T} \in \mathbb{T}$ directly yields the theorem, as the left-hand side becomes TS-Sobolev$_p(\mu, \nu)^p$ and the right-hand side becomes TSW$(\mu, \nu)$. This completes the proof. $\square$

### D.3    PROOF FOR THEOREM 3.5

*Proof.* We analyze the convergence of the Monte Carlo estimator for the TS-Sobolev, which is defined as:

$$\widehat{\text{TS-Sobolev}}_p(\mu, \nu) = \left(\frac{1}{L} \sum_{i=1}^{L} \hat{\mathcal{S}}_p(\mu_{\mathcal{T}_i}, \nu_{\mathcal{T}_i})^p\right)^{\frac{1}{p}}. \tag{50}$$

Let us define the random variable $X_i = \hat{\mathcal{S}}_p(\mu_{\mathcal{T}_i}, \nu_{\mathcal{T}_i})^p$, where each $X_i$ is an independent sample drawn by sampling a tree $\mathcal{T}_i \sim \sigma$. The estimator can then be written as a function of the sample mean $\bar{X} = \frac{1}{L} \sum_{i=1}^{L} X_i$.

The expected value of $X_i$ is the quantity we are trying to estimate (raised to the $p$-th power):

$$\mu_X = \mathbb{E}[X_i] = \int_{\mathbb{T}} \hat{\mathcal{S}}_p(\mu_{\mathcal{T}}, \nu_{\mathcal{T}})^p \, d\sigma(\mathcal{T}) = \text{TS-Sobolev}_p(\mu, \nu)^p. \tag{51}$$

Let the variance of $X_i$ be finite, denoted by $\sigma_X^2 = \mathbb{V}[X_i]$. By the Central Limit Theorem, the sample mean $\bar{X}$ is asymptotically normal, and its variance is $\mathbb{V}[\bar{X}] = \sigma_X^2/L$.

Our estimator is a function of this sample mean, specifically $g(\bar{X})$, where $g(y) = y^{1/p}$. To find the variance of our estimator, we apply the Delta Method. The variance of $g(\bar{X})$ can be approximated by:

$$\mathbb{V}[g(\bar{X})] \approx (g'(\mu_X))^2 \, \mathbb{V}[\bar{X}], \tag{52}$$

where $g'(\mu_X)$ is the derivative of $g$ evaluated at the true mean $\mu_X$. The derivative is $g'(y) = \frac{1}{p}y^{\frac{1}{p}-1}$. Substituting this into the variance approximation, we get:

$$\mathbb{V}\big[\widehat{\text{TS-Sobolev}}_p(\mu,\nu)\big] \approx \left(\frac{1}{p}\mu_X^{\frac{1}{p}-1}\right)^2 \frac{\sigma_X^2}{L}. \tag{53}$$

The Root Mean Squared Error (RMSE) of the estimator is the square root of the variance.

$$\text{RMSE} = \sqrt{\mathbb{V}\big[\widehat{\text{TS-Sobolev}}_p(\mu,\nu)\big]} \approx \frac{1}{\sqrt{L}}\left|\frac{1}{p}\mu_X^{\frac{1}{p}-1}\right|\sigma_X. \tag{54}$$

Since the terms $\mu_X$ and $\sigma_X$ are finite constants that do not depend on $L$, the Monte Carlo approximation error decays at the standard rate of $\mathcal{O}(L^{-1/2})$. $\qquad\square$

### D.4 PROOF FOR THEOREM B.3

To ensure a rigorous and self-contained presentation, we now derive the result in full, adopting the framework proposed by Le et al. (2025).

*Proof.* Let $f \in W_0^{1,p}(\mathcal{T},\omega)$. We first derive an upper bound for $\|f\|_{L^p}^p$ in terms of $\|f'\|_{L_{\hat{w}}^p}^p$. Since $f(z_0) = 0$, it follows that

$$\begin{aligned}
\|f\|_{L^p}^p &= \int_{\mathcal{T}} |f(x)|^p\,\omega(dx) \\
&= \int_{\mathcal{T}} \left|\int_{[z_0,x]} f'(y)\,\omega(dy)\right|^p \omega(dx) \\
&= \int_{\mathcal{T}} \left|\int_{\mathcal{T}} 1_{[z_0,x]}(y)\,f'(y)\,\omega(dy)\right|^p \omega(dx).
\end{aligned}$$

Applying Jensen's inequality, we obtain

$$\|f\|_{L^p}^p \le \omega(\mathcal{T})^{p-1} \int_{\mathcal{T}}\int_{\mathcal{T}} 1_{[z_0,x]}(y)\,|f'(y)|^p\,\omega(dy)\,\omega(dx).$$

By Fubini's theorem, we may interchange the order of integration, which yields

$$\begin{aligned}
\|f\|_{L^p}^p &\le \omega(\mathcal{T})^{p-1}\int_{\mathcal{T}} |f'(y)|^p \left(\int_{\mathcal{T}} 1_{[z_0,x]}(y)\,\omega(dx)\right)\omega(dy) \\
&= \omega(\mathcal{T})^{p-1}\int_{\mathcal{T}} |f'(y)|^p\,\omega(\Gamma(y))\,\omega(dy). \tag{55}
\end{aligned}$$

where we recall that $\Gamma(y) := \{x \in \mathcal{T} : y \in [z_0,x]\}$. Using the estimate from Equation (55), we obtain

$$\begin{aligned}
\|f\|_{W^{1,p}} &= (\|f\|_{L^p}^p + \|f'\|_{L^p}^p)^{\frac{1}{p}} \\
&\le \left(\omega(\mathcal{T})^{p-1}\int_{\mathcal{T}} [1 + \omega(\Lambda(x))]\,|f'(x)|^p\,\omega(dx) + \int_{\mathcal{T}} |f'(x)|^p\,\omega(dx)\right)^{\frac{1}{p}} \\
&= \left(\int_{\mathcal{T}} \left(1 + \omega(\mathcal{T})^{p-1}\omega(\Lambda(x))\right)|f'(x)|^p\,\omega(dx)\right)^{\frac{1}{p}} \\
&\le \left(\max\{1,\omega(\mathcal{T})^{p-1}\}\int_{\mathcal{T}} [1 + \omega(\Lambda(x))]\,|f'(x)|^p\,\omega(dx)\right)^{\frac{1}{p}} \\
&= a_2\,\|f'\|_{L_{\hat{w}}^p}, \tag{56}
\end{aligned}$$

where $\hat{w}(x) = 1 + \omega(\Lambda(x))$ and $a_2 = \left(\max\{1,\omega(\mathcal{T})^{p-1}\}\right)^{1/p}$.

Next, we derive a corresponding lower bound for $\|f\|_{W^{1,p}}$. Since $\|f\|_{L^p} \geq 0$, it follows that

$$
\begin{aligned}
\|f\|_{W^{1,p}} &= (\|f\|_{L^p}^p + \|f'\|_{L^p}^p)^{\frac{1}{p}} \\
&\geq \|f'\|_{L^p} \\
&= \left( \int_{\mathcal{T}} |f'(x)|^p \, \omega(dx) \right)^{\frac{1}{p}} \\
&= \left( \int_{\mathcal{T}} \frac{1}{1 + \omega(\mathcal{T})^p} \left[ 1 + \omega(\mathcal{T})^p \right] |f'(x)|^p \, \omega(dx) \right)^{\frac{1}{p}} \\
&\geq \left( \frac{\min\{1, \omega(\mathcal{T})^{p-1}\}}{1 + \omega(\mathcal{T})^p} \int_{\mathcal{T}} [1 + \omega(\Lambda(x))] \, |f'(x)|^p \, \omega(dx) \right)^{\frac{1}{p}} \\
&= a_1 \, \|f'\|_{L_{\hat{w}}^p},
\end{aligned}
\tag{57}
$$

where

$$
a_1 = \left( \frac{\min\{1, \omega(\mathcal{T})^{p-1}\}}{1 + \omega(\mathcal{T})^p} \right)^{1/p}.
\tag{58}
$$

Combining Equation (56) and Equation (57), we conclude that

$$
a_1 \, \|f'\|_{L_{\hat{w}}^p} \;\leq\; \|f\|_{W^{1,p}} \;\leq\; a_2 \, \|f'\|_{L_{\hat{w}}^p},
\tag{59}
$$

which completes the proof.

$\square$

### D.5 PROOF FOR THEOREM B.5

To ensure a rigorous and self-contained presentation, we now derive the result in full, adopting the framework proposed by Le et al. (2025).

*Proof.* Let $f \in W_0^{1,p'}(\mathcal{T}, \omega)$. By Definition B.1, we have

$$
f(x) = f(z_0) + \int_{[z_0,x]} f'(y) \, \omega(dy), \qquad \forall \, x \in \mathcal{T}.
\tag{60}
$$

Using Equation (60) together with the indicator function of the path $[z_0, x]$, and noting that $\mu(\mathcal{T}) = 1$, we obtain

$$
\begin{aligned}
\int_{\mathcal{T}} f(x) \, \mu(dx) &= \int_{\mathcal{T}} f(z_0) \, \mu(dx) + \int_{\mathcal{T}} \int_{[z_0,x]} f'(y) \, \omega(dy) \, \mu(dx) \\
&= f(z_0) + \int_{\mathcal{T}} \int_{\mathcal{T}} \mathbf{1}_{[z_0,x]}(y) \, f'(y) \, \omega(dy) \, \mu(dx).
\end{aligned}
$$

Applying Fubini's theorem to interchange the order of integration yields

$$
\int_{\mathcal{T}} f(x) \, \mu(dx) = f(z_0) + \int_{\mathcal{T}} \left( \int_{\mathcal{T}} \mathbf{1}_{[z_0,x]}(y) \, \mu(dx) \right) f'(y) \, \omega(dy).
$$

By the definition of $\Gamma(y)$, this becomes

$$
\int_{\mathcal{T}} f(x) \, \mu(dx) = f(z_0) + \int_{\mathcal{T}} f'(y) \, \mu(\Gamma(y)) \, \omega(dy).
\tag{61}
$$

An analogous computation gives

$$
\int_{\mathcal{T}} f(x) \, \nu(dx) = f(z_0) + \int_{\mathcal{T}} f'(y) \, \nu(\Gamma(y)) \, \omega(dy).
\tag{62}
$$

Hence, the regularized Sobolev IPM Equation (26) can be written as

$$\hat{\mathcal{S}}_p(\mu, \nu) = \sup_{f \in \mathcal{B}(p', \hat{w})} \left| \int_{\mathcal{T}} f'(x) \left( \mu(\Lambda(x)) - \nu(\Lambda(x)) \right) \omega(dx) \right|, \tag{63}$$

where

$$\mathcal{B}(p', \hat{w}) := \{ f \in W_0^{1,p'}(\mathcal{T}, \omega) : \|f'\|_{L_{\hat{w}}^{p'}} \leq 1 \}. \tag{64}$$

Observe that

$$\{ f' : f \in \mathcal{B}(p', \hat{w}) \} = \{ g \in L^{p'}(\mathcal{T}, \omega) : \|g\|_{L_{\hat{w}}^{p'}} \leq 1 \}. \tag{65}$$

Indeed, the inclusion "$\subseteq$" is immediate, while the reverse direction follows by constructing $f(x) := \int_{[z_0, x]} g(y) \, \omega(dy)$ for any $g \in L^{p'}(\mathcal{T}, \omega)$.

Now define

$$\hat{f}(x) := \frac{\mu(\Lambda(x)) - \nu(\Lambda(x))}{\hat{w}(x)}, \qquad x \in \mathcal{T}. \tag{66}$$

Substituting into Equation (63), we obtain

$$\hat{\mathcal{S}}_p(\mu, \nu) = \sup_{g \in L^{p'}(\mathcal{T}, \omega) : \|g\|_{L_{\hat{w}}^{p'}} \leq 1} \left| \int_{\mathcal{T}} \hat{w}(x) \, \hat{f}(x) \, g(x) \, \omega(dx) \right| \tag{67}$$

$$= \left( \int_{\mathcal{T}} \hat{w}(x) \, |\hat{f}(x)|^p \, \omega(dx) \right)^{1/p} \tag{68}$$

$$= \left( \int_{\mathcal{T}} \hat{w}(x)^{1-p} \, |\mu(\Lambda(x)) - \nu(\Lambda(x))|^p \, \omega(dx) \right)^{1/p},$$

where Equation (68) follows from the dual norm characterization of weighted $L^{p'}$ spaces.

Therefore,

$$\hat{\mathcal{S}}_p(\mu, \nu)^p = \int_{\mathcal{T}} \hat{w}(x)^{1-p} \left| \mu(\Lambda(x)) - \nu(\Lambda(x)) \right|^p \omega(dx), \tag{69}$$

which proves the claim. $\qquad \square$

### D.6 PROOF FOR THEOREM B.6

To ensure a rigorous and self-contained presentation, we now derive the result in full, adopting the framework proposed by Le et al. (2025).

*Proof.* We work with the length measure $\omega$ on the tree $\mathcal{T}$, so that $\omega(\{x\}) = 0$ for all $x \in \mathcal{T}$. From Theorem B.5, it follows that

$$\hat{\mathcal{S}}_p(\mu, \nu)^p = \sum_{e = \langle u, v \rangle \in E} \int_{(u,v)} \hat{w}(x)^{1-p} \left| \mu(\Lambda(x)) - \nu(\Lambda(x)) \right|^p \omega(dx). \tag{70}$$

Since $\mu, \nu$ are supported on vertices $V$, for any $x \in (u, v)$ we have

$$\mu(\Lambda(x)) - \nu(\Lambda(x)) = \mu(\Lambda(x) \setminus (u, v)) - \nu(\Lambda(x) \setminus (u, v)). \tag{71}$$

Substituting this into Equation (70) yields

$$\hat{\mathcal{S}}_p(\mu, \nu)^p = \sum_{e = \langle u, v \rangle \in E} \int_{(u,v)} \hat{w}(x)^{1-p} \left| \mu(\Lambda(x) \setminus (u, v)) - \nu(\Lambda(x) \setminus (u, v)) \right|^p \omega(dx). \tag{72}$$

For any edge $e = \langle u, v \rangle$, it follows that $\Lambda(x) \setminus (u, v) = \gamma_e$ for all $x \in (u, v)$. Hence, Equation (72) simplifies to

$$\hat{\mathcal{S}}_p(\mu, \nu)^p = \sum_{e = \langle u, v \rangle \in E} \left| \mu(\gamma_e) - \nu(\gamma_e) \right|^p \int_{(u,v)} \hat{w}(x)^{1-p} \, \omega(dx). \tag{73}$$

We now compute the integral term. Recall that $\hat{w}(x) = 1 + \omega(\Lambda(x))$. Without loss of generality, assume $d_{\mathcal{T}}(z_0, u) \leq d_{\mathcal{T}}(z_0, v)$, i.e., $v$ is farther from the root $z_0$. For $x \in (u, v)$, write $x = v + t(u - v)$ with $t \in (0, 1)$. Then

$$\omega(\Lambda(x)) = \omega(\gamma_e) + w_e t, \tag{74}$$

and therefore

$$\int_{(u,v)} \hat{w}(x)^{1-p} \, \omega(dx) = \int_0^1 \left[ 1 + \omega(\gamma_e) + w_e t \right]^{1-p} w_e \, dt. \tag{75}$$

This integral evaluates explicitly as

$$\int_{(u,v)} \hat{w}(x)^{1-p} \, \omega(dx) = \begin{cases} \log\left( 1 + \dfrac{w_e}{1 + \omega(\gamma_e)} \right), & p = 2, \\ \dfrac{(1 + \omega(\gamma_e) + w_e)^{2-p} - (1 + \omega(\gamma_e))^{2-p}}{2 - p}, & p \neq 2. \end{cases} \tag{76}$$

Thus, $\int_{(u,v)} \hat{w}(x)^{1-p} \, \omega(dx) = \beta_e$ (see Equation (29)). Substituting into Equation (73), we obtain

$$\hat{\mathcal{S}}_p(\mu, \nu) = \left( \sum_{e \in E} \beta_e \left| \mu(\gamma_e) - \nu(\gamma_e) \right|^p \right)^{1/p}. \tag{77}$$

This proves the result. $\qquad\square$

# E   SPHERICAL TREE-SLICED SOBOLEV IPM

## E.1   BACKGROUND ON SPHERICAL TREE-SLICED WASSERSTEIN DISTANCE

In this section, we review the concepts of Spherical Tree Systems, the Spherical Radon Transform, and the Spherical Tree-Sliced Wasserstein distance, as proposed by Tran et al. (2025d). These are the spherical analogs to the Euclidean framework of Tree Systems and their corresponding Radon Transforms (Tran et al., 2024d; 2025a). We will follow the construction of the Euclidean background, explaining the spherical components in the same order.

**Hypersphere.** The underlying space for these spherical constructions is the $d$-dimensional unit hypersphere, denoted $\mathbb{S}^d$. This is the set of all points in $(d+1)$-dimensional Euclidean space $\mathbb{R}^{d+1}$ that are at a distance of 1 from the origin. Formally:

$$\mathbb{S}^d := \left\{ x \in \mathbb{R}^{d+1} : \|x\|_2 = 1 \right\} \tag{78}$$

This space is a metric space equipped with the geodesic distance, which is the shortest distance between two points along the surface of the sphere. For any two points $a, b \in \mathbb{S}^d$, this distance is the angle between them, calculated as:

$$d_{\mathbb{S}^d}(a, b) = \arccos\left( \langle a, b \rangle_{\mathbb{R}^{d+1}} \right) \tag{79}$$

A key tool for relating the curved geometry of the sphere to flat Euclidean geometry is the stereographic projection $\varphi_x$. For a point $x \in \mathbb{S}^d$ (the "pole" of the projection), the map $\varphi_x$ projects point $y$ from the sphere onto the hyperplane $H_x$ that is tangent to the sphere at the antipode $-x$.

$$\varphi_x \colon \mathbb{S}^d \setminus \{x\} \longrightarrow H_x$$
$$y \longmapsto \frac{-\langle x, y \rangle}{1 - \langle x, y \rangle} \cdot x + \frac{1}{1 - \langle x, y \rangle} \cdot y. \tag{80}$$

By convention, the map is extended to the entire sphere by defining $\varphi_x(x) = \infty$, which completes the mapping $\varphi_x \colon \mathbb{S}^d \to H_x \cup \{\infty\}$.

**Components of Spherical Tree Systems.** The fundamental building block for a spherical tree is the *spherical ray*, the analog of a straight line in Euclidean space. It is formally constructed using the inverse stereographic projection. The intuitive idea is to first draw a straight ray on the flat hyperplane $H_x$ and then use the inverse map $\varphi_x^{-1}$ to trace this path back onto the curved surface of the sphere.

Mathematically, a spherical ray with root $x$ and direction $y$, denoted $r_y^x$, is defined as:

$$r_y^x = \varphi_x^{-1}\big(\{t \cdot y : t > 0\} \cup \{\infty\}\big) \tag{81}$$

Here, the set $\{t \cdot y : t > 0\}$ represents a straight ray on the hyperplane $H_x$, and $\varphi_x^{-1}$ maps this line back to the sphere. Each resulting spherical ray is isomorphic to the interval $[0, \pi]$ via the geodesic distance from its root, $d_{\mathbb{S}^d}(x, \cdot)$, which allows any point on the ray to be uniquely parameterized by its distance from $x$.

A *spherical tree system* $\mathcal{T}$ is formed by gluing a set of $k$ spherical rays $\{r_{y_i}^x\}_{i=1}^k$ at their common root $x$. This construction ensures that any two points in the resulting configuration are joined by a unique simple path, endowing the system with a valid tree metric, $d_{\mathcal{T}}$. The space of all such spherical trees with $k$ rays (or edges) in $\mathbb{S}^d$ is denoted by $\mathbb{T}_k^d$. This space is equipped with a probability distribution $\sigma$ that governs the tree sampling process.

**Sampling Spherical Tree Systems.** To sample a spherical tree system, we employ the generative process described in Tran et al. (2025d). The process begins by sampling a common root $x$ from the uniform distribution on the $d$-sphere, $\mu = \mathcal{U}(\mathbb{S}^d)$. To generate the $k$ orthogonal direction vectors, a second step is performed: for each direction, an initial vector is also sampled from $\mathcal{U}(\mathbb{S}^d)$. This vector is then projected onto the hyperplane $H_x$ (which is orthogonal to the root $x$) and re-normalized.

**Spherical Radon Transform on Spherical Tree Systems** Let $L^1(\mathbb{S}^d)$ be the space of Lebesgue-integrable functions on the hypersphere $\mathbb{S}^d$. For a given spherical tree system $\mathcal{T} \in \mathbb{T}_k^d$, we define $L^1(\mathcal{T})$ as the space of integrable functions $f : \mathcal{T} \to \mathbb{R}$ such that their norm, $\|f\|_{\mathcal{T}} = \sum_{i=1}^k \int_0^\pi |f(t, r_{y_i}^x)| \, dt$, is finite.

A *splitting map* is a continuous function $\alpha \in \mathcal{C}(\mathbb{S}^d \times \mathbb{T}_k^d, \Delta_{k-1})$ that assigns a weight distribution to the rays of a tree for any given point on the sphere. Given such a map, the *Spherical Radon Transform on Spherical Trees* projects a function $f \in L^1(\mathbb{S}^d)$ onto a function $\mathcal{R}_{\mathcal{T}}^\alpha f$ in $L^1(\mathcal{T})$. For a point $(t, r_{y_i}^x)$ on the $i$-th ray of the tree, the transform is defined as:

$$\mathcal{R}_{\mathcal{T}}^\alpha f(t, r_{y_i}^x) = \int_{\mathbb{S}^d} f(y) \cdot \alpha(y, \mathcal{T})_i \cdot \delta(t - \arccos \langle x, y \rangle) \, dy, \tag{82}$$

where $\delta$ is the Dirac delta. This operator is well-defined from $L^1(\mathbb{S}^d)$ to $L^1(\mathcal{T})$. Aggregating over all trees, the transform $\mathcal{R}^\alpha$ maps a function on the sphere to a collection of functions on all possible trees. If the splitting map $\alpha$ is invariant under the orthogonal group $O(d+1)$, this transform is injective (Tran et al., 2025d).

**Spherical Tree-Sliced Wasserstein Distance** For probability measures $\mu, \nu \in \mathcal{P}(\mathbb{S}^d)$, the operator $\mathcal{R}_{\mathcal{T}}^\alpha$ transports them to corresponding measures on the tree, $\mathcal{R}_{\mathcal{T}}^\alpha \mu$ and $\mathcal{R}_{\mathcal{T}}^\alpha \nu$. Since $\mathcal{T}$ has a tree metric $d_{\mathcal{T}}$, we can compute the 1-Wasserstein distance between these projected measures. The *Spherical Tree-Sliced Wasserstein (STSW) distance* is then defined as the expectation over all trees:

$$\text{STSW}(\mu, \nu) := \int_{\mathbb{T}_k^d} W_1(\mathcal{R}_{\mathcal{T}}^\alpha \mu, \mathcal{R}_{\mathcal{T}}^\alpha \nu) \, d\sigma(\mathcal{T}). \tag{83}$$

When the splitting map $\alpha$ is chosen to be $O(d+1)$-invariant, the STSW distance is an $O(d+1)$-invariant metric on $\mathcal{P}(\mathbb{S}^d)$.

**Splitting Maps.** The invariance of the metric relies on an $O(d+1)$-invariant splitting map. The group $O(d+1)$ consists of transformations on $\mathbb{R}^{d+1}$ that preserve the Euclidean norm and thus leave the sphere $\mathbb{S}^d$ invariant. A splitting map $\alpha$ is $O(d+1)$-invariant if $\alpha(gy, g\mathcal{T}) = \alpha(y, \mathcal{T})$ for all $g \in O(d+1)$.

A practical way to construct such a map is to base it on an invariant quantity. For a point $y \in \mathbb{S}^d$ and a spherical tree $\mathcal{T}$, one can define an invariant "distance" $\beta(y, \mathcal{T})_i$ from $y$ to each ray $i$ of the tree. A continuous and $O(d+1)$-invariant choice for this map, as presented in Tran et al. (2025d), is given by $\beta \colon \mathbb{S}^d \times \mathbb{T}_k^d \to \mathbb{R}^k$:

$$\beta(y, \mathcal{T}_{y_1,\ldots,y_k}^x)_i = \begin{cases} 0, & \text{if } y = \pm x, \\ \arccos\left(\frac{\langle y, y_i \rangle}{\sqrt{1 - \langle x, y \rangle^2}}\right) \cdot \sqrt{1 - \langle x, y \rangle^2}, & \text{if } y \neq \pm x. \end{cases} \tag{84}$$

A valid splitting map $\alpha$ can then be constructed by applying the softmax function to these values:

$$\alpha(y, \mathcal{T})_i = \text{softmax}\left(\{-\xi \cdot \beta(y, \mathcal{T})_j\}_{j=1}^k\right)_i. \tag{85}$$

This map assigns higher weights to the rays that are "closer" to the point $y$, while preserving the necessary rotational symmetries.

## E.2 SPHERICAL TREE-SLICED SOBOLEV IPM

The Tree-Sliced Sobolev IPM framework can be extended from Euclidean spaces to measures defined on the $d$-dimensional hypersphere, $\mu, \nu \in \mathcal{P}(\mathbb{S}^d)$. This is achieved by replacing the Euclidean components with their spherical analogs. Instead of projecting onto tree systems formed from straight lines, we project onto spherical tree systems built from spherical rays.

The projection is performed by the Spherical Radon Transform ($\mathcal{R}^\alpha$), which maps the spherical measures $\mu, \nu \in \mathcal{P}(\mathbb{S}^d)$ to corresponding measures $\mu_\mathcal{T} := \mathcal{R}_\mathcal{T}^\alpha(\mu)$ and $\nu_\mathcal{T} := \mathcal{R}_\mathcal{T}^\alpha(\nu)$ on a given spherical tree $\mathcal{T}$. We then compute the regularized Sobolev IPM, $\hat{\mathcal{S}}_p(\mu_\mathcal{T}, \nu_\mathcal{T})$, between these projected measures. The final distance is the expected value of this quantity, taken over the space of random spherical trees $\mathbb{T}$ with respect to a probability distribution $\sigma$.

**Definition E.1** (Spherical Tree-Sliced Sobolev IPM). The *Spherical Tree-Sliced Sobolev IPM* of order $p \in [1, \infty)$, denoted as STS-Sobolev$_p$, between $\mu, \nu \in \mathcal{P}(\mathbb{S}^d)$ is defined by

$$\text{STS-Sobolev}_p(\mu, \nu) := \left(\int_{\mathbb{T}} \hat{\mathcal{S}}_p(\mu_\mathcal{T}, \nu_\mathcal{T})^p \, d\sigma(\mathcal{T})\right)^{\frac{1}{p}}. \tag{86}$$

## E.3 PROPERTIES OF SPHERICAL TREE-SLICED SOBOLEV IPM

**Metricity of STS-Sobolev$_p$.** The metric properties of STS-Sobolev are guaranteed by the invariance of its components under the relevant symmetry group for the sphere: the *orthogonal group* $O(d+1)$. This is the group of distance-preserving linear transformations (rotations and reflections) in $\mathbb{R}^{d+1}$, which leave the sphere $\mathbb{S}^d$ invariant. An $O(d+1)$-invariant splitting map ensures that the Spherical Radon Transform is injective, which is crucial for the metric properties. Just as in the Euclidean case, this invariance guarantees that STS-Sobolev is not only invariant but also a valid metric.

**Theorem E.2.** *The* STS-Sobolev *is an* $O(d+1)$-*invariant metric on* $\mathcal{P}(\mathbb{S}^d)$.

The proof is analogous to that of Theorem 3.2.

**Connections to STSW.** The STS-Sobolev IPM is a natural generalization of the Spherical Tree-Sliced Wasserstein (STSW) distance (Tran et al., 2025d). It recovers STSW exactly for the case $p = 1$ and is bounded by it for all other orders.

**Theorem E.3.** *For any* $\mu, \nu \in \mathcal{P}(\mathbb{S}^d)$ *and* $p \geq 1$: STS-Sobolev$_p(\mu, \nu)^p \leq$ STSW$(\mu, \nu)$, *with equality if* $p = 1$, *i.e.,* STS-Sobolev$_1(\mu, \nu) =$ STSW$(\mu, \nu)$.

The proof is analogous to its Euclidean counterpart in Theorem 3.3.

**Computation of Spherical Tree-Sliced Sobolev IPM.** The integral in Equation (86) is intractable and is approximated using a Monte Carlo estimate by sampling $L$ spherical trees:

$$\widehat{\text{STS-Sobolev}}_p(\mu, \nu) = \left(\frac{1}{L} \sum_{i=1}^{L} \hat{\mathcal{S}}_p(\mu_{\mathcal{T}_i}, \nu_{\mathcal{T}_i})^p\right)^{\frac{1}{p}}. \tag{87}$$

The computational complexity of STS-Sobolev matches its first-order counterpart, STSW, at $\mathcal{O}(Ln \log n + Lkdn)$. The additional step of computing the coefficients $\beta_e$ per Equation (6) introduces a negligible overhead of only $\mathcal{O}(Lkn)$. Crucially, key advantages are preserved: this complexity holds for any order $p \in [1, \infty)$, and the empirical runtime remains nearly identical to that of STSW.

**Theorem E.4.** *The approximation error of STS-Sobolev decreases at a rate of $\mathcal{O}(L^{-1/2})$.*

The proof is analogous to that of Theorem 3.5.

### E.4 THEORETICAL PROOFS FOR SPHERICAL TREE-SLICED SOBOLEV IPM

In this section, we provide the proofs for the results stated in Appendix E.3.

**Proof for Theorem E.2**

*Proof.* We show that STS-Sobolev$_p$ is an $\mathrm{O}(d+1)$-invariant metric on the space of probability measures $\mathcal{P}(\mathbb{S}^d)$. The definition is given by

$$\text{STS-Sobolev}_p(\mu, \nu) = \left( \int_{\mathbb{T}} \hat{\mathcal{S}}_p(\mu_{\mathcal{T}}, \nu_{\mathcal{T}})^p \, d\sigma(\mathcal{T}) \right)^{\frac{1}{p}}, \tag{88}$$

where $\mu_{\mathcal{T}}$ and $\nu_{\mathcal{T}}$ are the projections of $\mu$ and $\nu$ via the Spherical Radon Transform, and $\hat{\mathcal{S}}_p$ is a metric on the space of measures on a spherical tree $\mathcal{T}$. The proof relies on the injectivity of the Spherical Radon Transform $\mathcal{R}^\alpha$, which holds because the splitting map $\alpha$ is chosen to be $\mathrm{O}(d+1)$-invariant (Tran et al., 2025d).

We now verify the three metric axioms. First, for positive definiteness, it is clear that STS-Sobolev$_p(\mu, \mu) = 0$ and STS-Sobolev$_p(\mu, \nu) \geq 0$. If STS-Sobolev$_p(\mu, \nu) = 0$, this implies $\int_{\mathbb{T}} \hat{\mathcal{S}}_p(\mu_{\mathcal{T}}, \nu_{\mathcal{T}})^p \, d\sigma(\mathcal{T}) = 0$. Since the integrand is non-negative, this means $\hat{\mathcal{S}}_p(\mu_{\mathcal{T}}, \nu_{\mathcal{T}}) = 0$ for almost all $\mathcal{T} \in \mathbb{T}$. As $\hat{\mathcal{S}}_p$ is a metric, it follows that $\mu_{\mathcal{T}} = \nu_{\mathcal{T}}$ for almost all $\mathcal{T}$. By the injectivity of the Spherical Radon Transform $\mathcal{R}^\alpha$, we conclude that the measures are equal, $\mu = \nu$.

Second, for *symmetry*, the property on each tree implies $\hat{\mathcal{S}}_p(\mu_{\mathcal{T}}, \nu_{\mathcal{T}})^p = \hat{\mathcal{S}}_p(\nu_{\mathcal{T}}, \mu_{\mathcal{T}})^p$. Therefore,

$$\begin{aligned}
\text{STS-Sobolev}_p(\mu, \nu) &= \left( \int_{\mathbb{T}} \hat{\mathcal{S}}_p(\mu_{\mathcal{T}}, \nu_{\mathcal{T}})^p \, d\sigma(\mathcal{T}) \right)^{\frac{1}{p}} \\
&= \left( \int_{\mathbb{T}} \hat{\mathcal{S}}_p(\nu_{\mathcal{T}}, \mu_{\mathcal{T}})^p \, d\sigma(\mathcal{T}) \right)^{\frac{1}{p}} = \text{STS-Sobolev}_p(\nu, \mu). 
\end{aligned} \tag{89}$$

Third, for the *triangle inequality*, we use the triangle inequality of $\hat{\mathcal{S}}_p$ for any $\mu_1, \mu_2, \mu_3 \in \mathcal{P}(\mathbb{S}^d)$ and then apply Minkowski's integral inequality:

$$\begin{aligned}
\text{STS-Sobolev}_p(\mu_1, \mu_3) &= \left( \int_{\mathbb{T}} \hat{\mathcal{S}}_p(\mu_{1,\mathcal{T}}, \mu_{3,\mathcal{T}})^p \, d\sigma(\mathcal{T}) \right)^{\frac{1}{p}} \\
&\leq \left( \int_{\mathbb{T}} \left( \hat{\mathcal{S}}_p(\mu_{1,\mathcal{T}}, \mu_{2,\mathcal{T}}) + \hat{\mathcal{S}}_p(\mu_{2,\mathcal{T}}, \mu_{3,\mathcal{T}}) \right)^p \, d\sigma(\mathcal{T}) \right)^{\frac{1}{p}} \\
&\leq \left( \int_{\mathbb{T}} \hat{\mathcal{S}}_p(\mu_{1,\mathcal{T}}, \mu_{2,\mathcal{T}})^p \, d\sigma(\mathcal{T}) \right)^{\frac{1}{p}} + \left( \int_{\mathbb{T}} \hat{\mathcal{S}}_p(\mu_{2,\mathcal{T}}, \mu_{3,\mathcal{T}})^p \, d\sigma(\mathcal{T}) \right)^{\frac{1}{p}} \\
&= \text{STS-Sobolev}_p(\mu_1, \mu_2) + \text{STS-Sobolev}_p(\mu_2, \mu_3).
\end{aligned} \tag{90}$$

Thus, STS-Sobolev$_p$ is a metric on $\mathcal{P}(\mathbb{S}^d)$. For $\mathrm{O}(d+1)$-*invariance*, we aim to show that for any $g \in \mathrm{O}(d+1)$, STS-Sobolev$_p(\mu, \nu) = \text{STS-Sobolev}_p(g\sharp\mu, g\sharp\nu)$. Since $\alpha$ is $\mathrm{O}(d+1)$-invariant, the action of $g$ is an isometry, i.e., $\hat{\mathcal{S}}_p(\mu_{\mathcal{T}}, \nu_{\mathcal{T}}) = \hat{\mathcal{S}}_p((g\sharp\mu)_{g\mathcal{T}}, (g\sharp\nu)_{g\mathcal{T}})$ (Tran et al., 2025d). Using this

and a change of variables, we compute:

$$\text{STS-Sobolev}_p(g\sharp\mu, g\sharp\nu)^p = \int_{\mathbb{T}} \hat{\mathcal{S}}_p((g\sharp\mu)_{\mathcal{T}}, (g\sharp\nu)_{\mathcal{T}})^p \, d\sigma(\mathcal{T})$$

$$= \int_{\mathbb{T}} \hat{\mathcal{S}}_p((g\sharp\mu)_{g\mathcal{T}}, (g\sharp\nu)_{g\mathcal{T}})^p \, d\sigma(g\mathcal{T})$$

$$= \int_{\mathbb{T}} \hat{\mathcal{S}}_p(\mu_{\mathcal{T}}, \nu_{\mathcal{T}})^p \, d\sigma(\mathcal{T}) = \text{STS-Sobolev}_p(\mu, \nu)^p. \tag{91}$$

Taking the $p$-th root of both sides, we conclude that STS-Sobolev$_p$ is $\mathrm{O}(d+1)$-invariant. $\qquad\square$

**Proof for Theorem E.3**

*Proof.* We prove the theorem in two parts. First, we establish the equality for the case $p = 1$, and second, we prove the general inequality for any $p \in [1, \infty)$.

**Part 1.** We first prove equality for $p = 1$. By definition, for the case $p = 1$, the Spherical Tree-Sliced Sobolev IPM is:

$$\text{STS-Sobolev}_1(\mu, \nu) = \int_{\mathbb{T}} \hat{\mathcal{S}}_1(\mu_{\mathcal{T}}, \nu_{\mathcal{T}}) \, d\sigma(\mathcal{T}). \tag{92}$$

The Spherical Tree-Sliced Wasserstein distance is defined as $\text{STSW}(\mu, \nu) = \int_{\mathbb{T}} \mathrm{W}_1(\mu_{\mathcal{T}}, \nu_{\mathcal{T}}) \, d\sigma(\mathcal{T})$. To prove the theorem, it is sufficient to show the integrands are equal. We analyze the discrete form of the Sobolev IPM for $p = 1$:

$$\hat{\mathcal{S}}_1(\mu_{\mathcal{T}}, \nu_{\mathcal{T}}) = \sum_{e \in E} \beta_e \left| \mu(\gamma_e) - \nu(\gamma_e) \right|. \tag{93}$$

For $p = 1$, the coefficient $\beta_e$ simplifies to $\beta_e = w_e$. Substituting this result gives:

$$\hat{\mathcal{S}}_1(\mu_{\mathcal{T}}, \nu_{\mathcal{T}}) = \sum_{e \in E} w_e \left| \mu(\gamma_e) - \nu(\gamma_e) \right|, \tag{94}$$

which is the known closed-form solution for the 1-Wasserstein distance on a tree, $\mathrm{W}_1(\mu_{\mathcal{T}}, \nu_{\mathcal{T}})$. Since the integrands are equal, their expectations are equal, proving that $\text{STS-Sobolev}_1(\mu, \nu) = \text{STSW}(\mu, \nu)$.

**Part 2.** Next, we prove the general inequality by showing that on any given tree $\mathcal{T}$, the integrand is bounded as $\hat{\mathcal{S}}_p(\mu_{\mathcal{T}}, \nu_{\mathcal{T}})^p \leq \mathrm{W}_1(\mu_{\mathcal{T}}, \nu_{\mathcal{T}})$.

This relies on two facts established in Appendix D.2: (1) the Sobolev coefficient $\beta_e \leq w_e$ for all $p \geq 1$, and (2) for probability measures, $|\mu(\gamma_e) - \nu(\gamma_e)| \in [0, 1]$, which implies $|\ldots|^p \leq |\ldots|$ for $p \geq 1$. Using these facts, we can bound the $p$-th power of the Sobolev IPM on a tree:

$$\hat{\mathcal{S}}_p(\mu_{\mathcal{T}}, \nu_{\mathcal{T}})^p = \sum_{e \in E} \beta_e \left| \mu(\gamma_e) - \nu(\gamma_e) \right|^p$$

$$\leq \sum_{e \in E} w_e \left| \mu(\gamma_e) - \nu(\gamma_e) \right|^p \qquad \text{(since } \beta_e \leq w_e)$$

$$\leq \sum_{e \in E} w_e \left| \mu(\gamma_e) - \nu(\gamma_e) \right| \qquad \text{(since } |\ldots|^p \leq |\ldots|)$$

$$= \mathrm{W}_1(\mu_{\mathcal{T}}, \nu_{\mathcal{T}}). \tag{95}$$

Integrating the inequality $\hat{\mathcal{S}}_p(\mu_{\mathcal{T}}, \nu_{\mathcal{T}})^p \leq \mathrm{W}_1(\mu_{\mathcal{T}}, \nu_{\mathcal{T}})$ over all trees $\mathcal{T} \in \mathbb{T}$ directly yields the theorem and completes the proof. $\qquad\square$

**Proof for Theorem E.4**

*Proof.* The Monte Carlo estimator for STS-Sobolev is $\widehat{\text{STS-Sobolev}}_p(\mu, \nu) = \left( \frac{1}{L} \sum_{i=1}^{L} \hat{\mathcal{S}}_p(\mu_{\mathcal{T}_i}, \nu_{\mathcal{T}_i})^p \right)^{\frac{1}{p}}$. Let the random variable $X_i = \hat{\mathcal{S}}_p(\mu_{\mathcal{T}_i}, \nu_{\mathcal{T}_i})^p$, where each $\mathcal{T}_i \sim \sigma$. The

expected value of $X_i$ is $\mu_X = \mathbb{E}[X_i] = \text{STS-Sobolev}_p(\mu, \nu)^p$. Let the variance of $X_i$ be finite, $\sigma_X^2 = \mathbb{V}[X_i]$. By the Central Limit Theorem, the sample mean $\bar{X} = \frac{1}{L}\sum_{i=1}^{L} X_i$ has variance $\mathbb{V}[\bar{X}] = \sigma_X^2/L$.

Our estimator is the function $g(\bar{X}) = \bar{X}^{1/p}$. Applying the Delta Method, the variance of the estimator can be approximated by $\mathbb{V}[g(\bar{X})] \approx (g'(\mu_X))^2 \mathbb{V}[\bar{X}]$, where the derivative is $g'(y) = \frac{1}{p}y^{\frac{1}{p}-1}$. The Root Mean Squared Error (RMSE) is the square root of the variance:

$$\text{RMSE} = \sqrt{\mathbb{V}[\widehat{\text{STS-Sobolev}}_p(\mu, \nu)]} \approx \frac{1}{\sqrt{L}}\left|\frac{1}{p}\mu_X^{\frac{1}{p}-1}\right|\sigma_X. \tag{96}$$

Since $\mu_X$ and $\sigma_X$ are finite constants independent of the number of samples $L$, the Monte Carlo approximation error decays at the standard rate of $\mathcal{O}(L^{-1/2})$. □

## F EXPERIMENTAL DETAILS

### F.1 RUNTIME AND MEMORY ANALYSIS

Table 6: Complexity Analysis of TS-Sobolev and STS-Sobolev.

| Distance | Operation | Description | Computation | Memory |
|---|---|---|---|---|
| TS-Sobolev | Sampling | Random sampling concurrent-line trees | $O(Lkd)$ | $O(Lkd)$ |
| | Projection | Matrix multiplication of points and lines | $O(Lknd)$ | $O(Lkd + nd)$ |
| | Distance-based weight splitting | Distance calculation and softmax | $O(Lknd)$ | $O(Lkn + Lkd + nd)$ |
| | Sorting | Sorting projected coordinates | $O(Lkn \log n)$ | $O(Lkn)$ |
| | Coefficient computation | Computing coefficients ($\beta_e$) | $O(Lkn)$ | $O(Lkn)$ |
| | **Total** | | $O(Lknd + Lkn \log n + Tnd)$ | $O(Lkn + Lkd + nd + Tnd)$ |
| STS-Sobolev | Sampling | Random sampling spherical trees | $O(Lkd)$ | $O(Lkd)$ |
| | Projection | Matrix multiplication of points and source | $O(Lnd)$ | $O(Ld + nd)$ |
| | Distance-based weight splitting | Distance calculation and softmax | $O(Lknd)$ | $O(Lkn + Lkd + nd)$ |
| | Sorting | Sorting projected coordinates | $O(Ln \log n)$ | $O(Ln)$ |
| | Coefficient computation | Computing coefficients ($\beta_e$) | $O(Lkn)$ | $O(Lkn)$ |
| | **Total** | | $O(Lknd + Ln \log n)$ | $O(Lkn + Lkd + nd)$ |

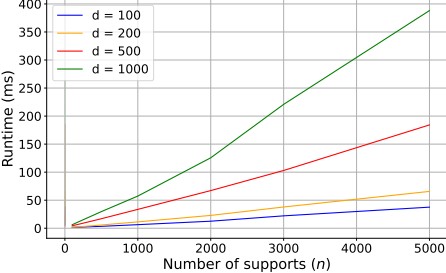 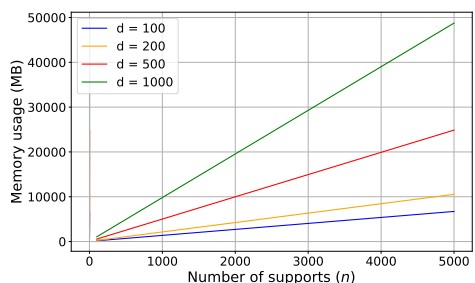

Figure 4: Execution time and memory usage of TS-Sobolev$_{1.5}$.

**Computational and Memory Complexity.** We summarize the complexity of our proposed distance measures in Table 6. The overall computational and memory costs for TS-Sobolev and STS-Sobolev are identical to their respective counterparts, Db-TSW (Tran et al., 2025a) and STSW (Tran et al., 2025d). This is because the additional step of computing the coefficients, as defined in Equation Equation (6), has a low complexity of $\mathcal{O}(Lkn)$, which is subsumed by the dominant terms of the projection and sorting operations.

**Empirical Scaling Analysis.** To verify our theoretical complexity, we benchmark the runtime and memory scaling of TS-Sobolev$_{1.5}$ and TS-Sobolev$_2$ with respect to the number of support points ($n$) and the data dimension ($d$). For these experiments, we fix the hyperparameters at $L = 2500$ trees and $k = 4$ lines per tree, and run all tests on a single NVIDIA H100 GPU. Results are averaged over 10 runs.

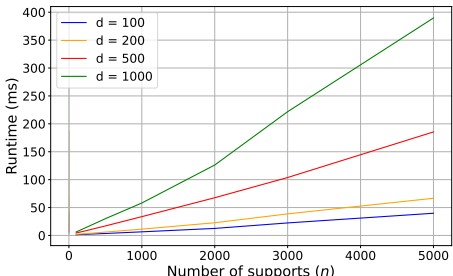 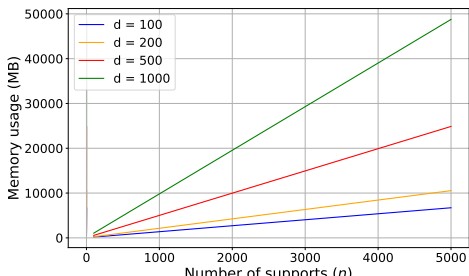

Figure 5: Execution time and memory usage of TS-Sobolev$_2$.

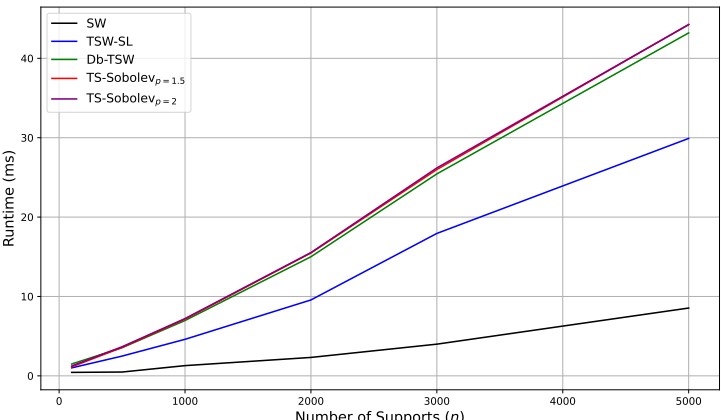

Figure 6: Runtime Comparison of TS-Sobolev and other methods.

As detailed in Figures 4 and 5, the results confirm our analysis. The left panel of each figure shows that runtime exhibits a clear linear scaling with both $n$ and $d$. Similarly, the right panel of each figure shows that memory usage also scales linearly. These empirical findings are fully consistent with the theoretical complexities presented earlier.

**Runtime Comparison with Other Methods.** We empirically compare the runtime of our method against its counterparts in Figure 6. For the tree-sliced methods (Db-TSW and TS-Sobolev), we use $L = 2500$ trees and $k = 4$ lines per tree, while for Sliced Wasserstein (SW), we use $10,000$ projections. All results are averaged over 10 independent runs. The experiment confirms that TS-Sobolev's runtime is nearly identical to that of Db-TSW, as the additional coefficient computation step introduces negligible overhead.

**Runtime Across Order** $p$**.** As also shown in Figure 6, the runtime of TS-Sobolev remains consistent across different orders of $p$. This is a crucial advantage of our framework, demonstrating that it provides the flexibility to use higher-order metrics without incurring any performance penalty compared to the standard $p = 1$ TSW.

### F.2 Gradient Flow on Euclidean space and on the sphere

**Euclidean Datasets.** Table 1 reports the performance of our proposed methods compared with several baselines on the 8 Gaussians and Gaussian 30d datasets. We use $L = 25$ trees and $k = 4$ lines for tree-sliced methods and $L = 100$ projections for other sliced methods. We train for 2500 steps using the Adam optimizer with a global learning rate of $0.005$. For the TSW variant, we use a learning rate of $0.005$ for the 25 Gaussians dataset and $0.05$ for the Gaussian 30d dataset, as in Tran et al. (2025a). For TSW-SL and Db-TSW, we set p=1. Each distribution has 500 samples.

Table 7: Average 1-Wasserstein distance ($W_1$) on Gaussian 30d (Euclidean)

| Method | Iter 500 | Iter 1000 | Iter 1500 | Iter 2000 | Iter 2500 |
|---|---|---|---|---|---|
| SW | 23.5 | 23.2 | 22.9 | 22.6 | 22.3 |
| SWGG | 22.6 | 22.7 | 22.7 | 22.7 | 22.7 |
| LCVSW | 23.1 | 22.5 | 22.0 | 21.4 | 20.8 |
| TSW-SL | 21.4 | 20.6 | 20.0 | 19.4 | 18.8 |
| Db-TSW | 21.1 | 19.8 | 18.6 | 17.3 | 16.1 |
| TS-Sobolev$_{1.2}$ | 21.0 | 19.5 | 18.0 | 16.4 | 14.9 |
| TS-Sobolev$_{1.5}$ | 20.4 | 18.1 | 14.7 | 10.5 | **6.69** |
| TS-Sobolev$_2$ | 22.1 | 21.2 | 20.1 | 19.2 | 18.6 |

Table 8: Average Log 1-Wasserstein distance ($W_1$) on Mixture of vMFs (Spherical)

| Method | Epoch 50 | Epoch 100 | Epoch 150 | Epoch 200 | Epoch 250 |
|---|---|---|---|---|---|
| SSW | -1.477 | -1.740 | -1.870 | -1.951 | -2.008 |
| S3W | -1.213 | -1.373 | -1.451 | -1.486 | -1.509 |
| RI-S3W (1) | -1.225 | -1.479 | -1.602 | -1.672 | -1.722 |
| RI-S3W (5) | -1.423 | -1.662 | -1.782 | -1.863 | -1.913 |
| ARI-S3W | -1.527 | -1.792 | -1.950 | -2.057 | -2.136 |
| STSW | -1.530 | -1.812 | -1.969 | -2.046 | -2.082 |
| STS-Sobolev$_{1.5}$ | -1.842 | -2.051 | -2.097 | -2.151 | -2.139 |
| STS-Sobolev$_2$ | -1.814 | -2.054 | -2.120 | -2.150 | **-2.158** |

**Spherical Datasets.** The probability density of the von Mises-Fisher distribution with mean direction $\mu \in \mathbb{S}^d$ is expressed as:

$$f(x; \mu, \kappa) = C_d(\kappa) \exp(\kappa \mu^T x)$$

where $\kappa > 0$ controls concentration, and $C_d(\kappa) = \dfrac{\kappa^{d/2-1}}{(2\pi)^{p/2} I_{p/2-1}(\kappa)}$ is the normalization factor.

Following Bonet et al. (2022); Tran et al. (2024c; 2025d), we consider a target distribution of 12 vMFs with 2400 samples (200 per vFM) where $\kappa = 50$ and

$$\begin{aligned}
\mu_1 &= (-1, \phi, 0), & \mu_2 &= (1, \phi, 0), & \mu_3 &= (-1, -\phi, 0), & \mu_4 &= (1, -\phi, 0) \\
\mu_5 &= (0, -1, \phi), & \mu_6 &= (0, 1, \phi), & \mu_6 &= (0, -1, -\phi), & \mu_8 &= (0, 1, -\phi) \\
\mu_9 &= (\phi, 0, -1), & \mu_{10} &= (\phi, 0, 1), & \mu_{11} &= (-\phi, 0, -1), & \mu_{12} &= (-\phi, 0, 1)
\end{aligned}$$

where $\phi = \dfrac{1 + \sqrt{5}}{2}$.

We fix $L = 200$ trees and $k = 5$ lines for tree-sliced distance while using $L = 1000$ projections for the rest. ARI-S3W (30) has 30 rotations with a pool size of 1000. RI-S3W (1) and RI-S3W (5) have 1 and 5 rotations, respectively. All methods are trained using Adam (Kinga et al., 2015) optimizer with $lr = 0.01$ over 250 epochs. For STS_Sobolev, we use a learning rate of 0.05.

**Evaluation using $W_1$.** To verify our performance gains, we re-evaluated the gradient flow experiments using the **1-Wasserstein ($W_1$) distance**. As detailed in Table 7, the proposed method demonstrates significant improvements in the Euclidean setting under $W_1$ evaluation; notably, on the high-dimensional Gaussian 30d dataset, TS-Sobolev ($p = 1.5$) achieves a final $W_1$ distance of **6.69**, substantially outperforming the strongest baseline, Db-TSW (16.1). Furthermore, in the spherical setting (Table 8), STS-Sobolev ($p = 2$) achieves the lowest final log $W_1$ distance of **-2.158**, surpassing both STSW (-2.082) and the strongest sliced baseline, ARI-S3W (-2.136).

**Ablating Tree-Projection Settings.** We conduct an ablation study to verify that the performance gains of the TS-Sobolev framework translate to other tree projection settings. Theoretically, the core advantage of our proposed method is derived from the metric formulation itself and should therefore persist regardless of the specific tree structure or splitting map employed. To empirically validate

this, we conducted additional Gradient Flow experiments on the Gaussian 30d dataset, evaluating the method across four distinct configurations: combinations of **Chain** versus **Concurrent** tree sampling strategies, and **Uniform** versus **Distance-based** splitting maps.

As presented in Table 9, TS-Sobolev$_{1.2}$ consistently outperforms the standard TSW$_1$ baseline across all four settings. This consistency confirms that the performance gains are intrinsic to the regularized Sobolev metric formulation rather than being specific to the default projection settings.

Table 9: Ablation study on the Gaussian 30d dataset. We report the average Wasserstein distance (multiplied by $10^{-1}$) between source and target distributions at iteration 2500 across different tree structures and splitting maps. Results are averaged over 10 runs.

| Tree Sampling | Splitting Map | TSW$_1$ | TS-Sobolev$_{1.2}$ | TS-Sobolev$_{1.5}$ | TS-Sobolev$_2$ |
|---|---|---|---|---|---|
| Chain | Uniform | 2.01 | **1.89** | 4.54 | 12.30 |
| Chain | Distance | 1.56 | **1.49** | 1.88 | 5.30 |
| Concurrent | Uniform | 1.93 | **1.83** | 3.37 | 11.10 |
| Concurrent | Distance | 1.78 | **1.40** | 1.51 | 3.68 |

### F.3 DIFFUSION MODELS

**Diffusion Models.** Diffusion models (Sohl-Dickstein et al., 2015; Ho et al., 2020) are a class of generative models renowned for producing high-quality samples. Their methodology is based on a dual-process framework. The first is a fixed *forward process*, where data $x_0$ is progressively corrupted over $T$ timesteps by adding Gaussian noise according to a predefined variance schedule, $\beta_t$. This noising cascade is defined by the transition kernel:

$$q(x_t|x_{t-1}) = \mathcal{N}(x_t; \sqrt{1-\beta_t}x_{t-1}, \beta_t I).$$

The second is a learned *reverse process*, where a neural network, parameterized by $\theta$, is trained to reverse the corruption. At each timestep $t$, the model learns to predict the denoised sample $x_{t-1}$ from the noisy input $x_t$. This learned denoising step is also modeled as a Gaussian distribution:

$$p_\theta(x_{t-1}|x_t) = \mathcal{N}(x_{t-1}; \mu_\theta(x_t, t), \sigma_t^2 I).$$

The model is trained by optimizing the Evidence Lower Bound (ELBO), which is equivalent to minimizing the Kullback-Leibler (KL) divergence between the model's predicted distribution $p_\theta(x_{t-1}|x_t)$ and the true posterior $q(x_{t-1}|x_t)$.

**Denoising Diffusion GANs.** A significant drawback of traditional diffusion models is their slow sampling speed, which stems from the large number of sequential steps ($T$) required. *Denoising Diffusion GANs (DDGANs)* (Xiao et al., 2021) address this inefficiency by reformulating the reverse process. Instead of a simple denoising network, DDGANs employ a conditional Generative Adversarial Network (GAN) for each reverse step. This approach allows for much larger and more expressive denoising transitions, drastically reducing the number of sampling steps needed—sometimes to as few as four—and enabling over 2000x speedups without substantial loss in sample quality.

The training objective for DDGANs has also evolved. While the original work relied on a standard adversarial loss, Nguyen et al. (2024) successfully replaced it with the *Augmented Generalized Mini-batch Energy (AGME)* distance. The AGME is a sophisticated metric derived from the Generalized Mini-batch Energy (GME) distance (Salimans et al., 2018), which quantifies the difference between two distributions by comparing the distances between mini-batches of their samples. The GME distance is defined as:

$$\text{GME}_b^2(\mu, \nu) = 2\mathbb{E}[D(P_X, P_Y)] - \mathbb{E}[D(P_X, P_X')] - \mathbb{E}[D(P_Y, P_Y')],$$

where $P_X, P_Y$ are empirical measures from mini-batches and $D$ is a chosen base metric. The effectiveness of this training scheme is highly dependent on the choice of $D$. In our work, we explore the performance of Sliced Wasserstein (SW) and our proposed Tree-Sliced Wasserstein (TSW) variants as the base metric $D$ within this framework.

**Implementation Details.** Our experimental configuration closely follows the setup of Nguyen et al. (2024) and Tran et al. (2025a) for model architecture and core hyperparameters. All models are trained for 1800 epochs.

For the tree-sliced methods, we set the number of sampled trees to $L = 2500$ and lines per tree to $k = 4$, with a sampling standard deviation of 0.1, per (Tran et al., 2025a). In contrast, for Sliced Wasserstein (SW) methods, we use $L = 10000$ projections, consistent with (Nguyen et al., 2024). We adopt the learning rates from the same work, setting them to $lr_d = 1.25 \times 10^{-4}$ for the discriminator and $lr_g = 1.6 \times 10^{-4}$ for the generator. All runtime evaluations are conducted with a batch size of 128 on two NVIDIA H100 GPUs. Our results for TS-Sobolev are averaged over 10 runs while other results are obtained from previous results.

### F.4 SELF-SUPERVISED LEARNING

**Encoder.** In line with (Bonet et al., 2022; Tran et al., 2024c; 2025d), we train a ResNet18 (He et al., 2016) on CIFAR-10 data for 200 epochs with a batch size of 512. Training uses SGD with $lr = 0.05$, a momentum 0.9, and a weight decay of $10^{-3}$. Data augmentations for creating positive pairs are aligned with earlier studies (Wang & Isola, 2020; Bonet et al., 2022; Tran et al., 2024c), including resizing, cropping, horizontal flipping, color jittering, and random grayscale conversion.

For tree-sliced methods, we use $L = 200$ trees, $k = 20$ lines for STSW, and $\lambda = 10$. We set $L = 200$ projections for all other sliced distances. We report result in Table 3 where $d = 9$.

**Linear Classifier.** We then train a linear classifier on the frozen representations produced by the encoder. Similar to prior works Bonet et al. (2022), training runs for 100 epochs using the Adam (Kinga et al., 2015) optimizer using a learning rate of $10^{-3}$, a weight decay of 0.2 at epochs 60 and 80.

### F.5 TOPIC MODELING.

Topic modeling task (Blei et al., 2003) seeks to automatically extract distinct themes from collections of text documents, revealing the underlying structure of a corpus. Recent approaches typically employ a variational autoencoder (VAE) setup, in which the optimization balances accurate document reconstruction with a regularization that encourages the inferred topic distributions to resemble a chosen prior (Srivastava & Sutton, 2017). Inspired by Nan et al. (2019); Adhya & Sanyal (2025), we propose replacing the conventional KL-divergence regularizer with a Wasserstein-based alternative. This leads to the following objective:

$$\inf_{\varphi, \psi} \mathbb{E}_{p(\mathbf{x})} \mathbb{E}_{q_\varphi(\theta|\mathbf{x})} \left[ \mathrm{CE}(\mathbf{x}, \hat{\mathbf{x}}) \right] + \lambda \, \mathrm{TS\_Sobolev}(q_\varphi(\theta), p(\theta)),$$

where CE represents the cross-entropy between the input document $\mathbf{x}$ (in bag-of-words representation) and its reconstruction $\hat{\mathbf{x}}$. The variational posterior $q_\varphi(\theta|\mathbf{x})$ is generated by encoder $\varphi$, and the decoder $\psi$ maps topic mixtures $\theta$ back to word distributions to form $\hat{\mathbf{x}}$.

**Datasets.** We evaluate our proposed methods on three well-known benchmark corpora used extensively for topic modeling research:

- **BBC** (Greene & Cunningham, 2006): Comprising more than 2,000 news articles published by the BBC, grouped into 5 topical classes.
- **M10** (Pan et al., 2016): Extracted from the CiteSeer$^X$ digital library, containing over 8,000 academic papers spanning 10 distinct research fields.

Preprocessing include lowercasing, punctuation removal, lemmatization, filtering out words shorter than three characters, and exclusion of documents with fewer than three words. Comprehensive statistics on these datasets after preprocessing are summarized in Table 10.

**Evaluation Metrics.** To quantitatively measure model effectiveness, we consider topic coherence and diversity. Topic coherence is measured using the $C_V$ (CV) metric (Röder et al., 2015), which correlates well with human interpretability, while topic diversity is assessed via the IRBO metric (Terragni et al., 2021), which evaluates how distinct the topics are. Topic coherence reflects the extent to which high-probability words within topics co-occur in documents, whereas diversity reflects how thematically different the topics are from one another.

Table 10: Statistics and hyperparameters for datasets used.

| Dataset | Dataset statistics | | | Hyperparameters | | | |
|---------|-------|---------|--------|--------------|------------|--------------|------------------|
| | #Docs | #Labels | #Words | #Projections | Batch size | Dropout rate | Spherical prior |
| M10 | 8,355 | 10 | 1,696 | 2,000 | 64 | 0.5 | Uniform |
| BBC | 2,225 | 5 | 2,949 | 8,000 | 256 | 0.05 | vMF |

Table 11: Topic diversity scores as measured by IRBO ($\uparrow$) on the BBC and M10 datasets

| Method | BBC | M10 |
|--------|-----|-----|
| LDA (Blei et al., 2003) | $0.917 \pm 0.016$ | $0.915 \pm 0.012$ |
| ProdLDA (Srivastava & Sutton, 2017) | $1.000 \pm 0.000$ | $0.997 \pm 0.002$ |
| WTM (Nan et al., 2019) | $0.997 \pm 0.003$ | $0.812 \pm 0.060$ |
| *Euclidean setting* | | |
| SW-TM (Bonneel et al., 2015) | $0.998 \pm 0.004$ | $0.971 \pm 0.008$ |
| RPSW-TM (Nguyen et al., 2024) | $0.994 \pm 0.010$ | $0.969 \pm 0.010$ |
| EBRPSW-TM (Nguyen et al., 2024) | $0.999 \pm 0.003$ | $0.963 \pm 0.013$ |
| TSW-SL-TM (Tran et al., 2024d) | $0.999 \pm 0.003$ | $0.981 \pm 0.005$ |
| Db-TSW-TM (Tran et al., 2025a) | $1.000 \pm 0.000$ | $0.805 \pm 0.044$ |
| TS-Sobolev$_2$-TM (ours) | $1.000 \pm 0.000$ | $0.984 \pm 0.006$ |
| *Spherical setting* | | |
| S2WTM (Adhya & Sanyal, 2025; Bonet et al., 2022) | $0.995 \pm 0.008$ | $0.944 \pm 0.006$ |
| S3W-TM (Tran et al., 2024c) | $0.987 \pm 0.014$ | $0.763 \pm 0.079$ |
| LSSOT-TM (Liu et al., 2025) | $0.994 \pm 0.004$ | $0.922 \pm 0.026$ |
| STSW-TM (Tran et al., 2025d) | $0.995 \pm 0.005$ | $0.854 \pm 0.028$ |
| STS-Sobolev$_2$-TM (ours) | $0.996 \pm 0.005$ | $0.895 \pm 0.027$ |

**Training Protocol.** The experiments are conducted using the OCTIS framework (Terragni et al., 2021), adhering to the setup described in (Adhya & Sanyal, 2025). Each model is trained for 100 epochs, employing a Dirichlet prior when operating in the Euclidean latent space, while parameters for the spherical latent space prior are specified in Table 10. The regularization weight $\lambda$ is systematically varied between 0.5 and 10 in steps of 0.5. For approaches involving tree-based objectives, the number of trees is fixed at 100. Additional training configurations can be found in Table 10.

**Topic Diversity.** The topic diversity results measured by IRBO scores $\uparrow$ on the DBLP, M10, and BBC datasets are displayed in Table 11. Notably, our proposed methods achieved comparable topic diversity and superior topic coherence relative to the baselines, underscoring their practical advantages.

F.6 EFFECTS OF THE NUMBER OF TREES ($L$) AND LINES PER TREE ($k$)

The computational complexity of the framework scales linearly with both the number of sampled trees $L$ and the number of lines per tree $k$, making the selection of these hyperparameters a key practical consideration.

**Number of Trees ($L$).** The parameter $L$ directly governs the precision of the Monte Carlo estimate. Since the approximation error decays at a rate of $\mathcal{O}(L^{-1/2})$, a sufficiently large $L$ is necessary to ensure accuracy. Our experiments confirm this, showing that higher values of $L$ consistently lead to improved performance on downstream tasks.

**Lines per Tree ($k$).** The parameter $k$ controls the geometric expressiveness of each individual tree structure. We observe empirically that using multiple lines ($k > 1$) yields significantly better results than the $k = 1$ case (which simplifies to a standard sliced distance). This confirms that the richer geometry of the tree structure is vital for performance.

**The Trade-off between $L$ and $k$.** While a larger $k$ can intuitively capture more intricate data geometries, it also dramatically expands the space of possible tree structures. This, in turn, may require

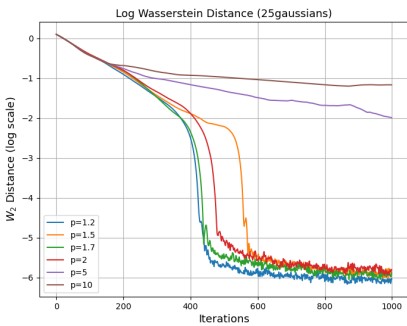 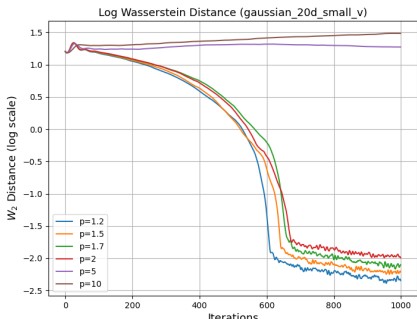

Figure 7: Log Wasserstein Distance between two distributions for $p \in \{1.2, 1.5, 1.6, 2, 5, 10\}$

a much larger number of samples $L$ to ensure the space of trees is adequately represented. Our experiments indicate that naively increasing $k$ to very high values does not always improve results, likely due to this sampling challenge. Developing methods that can leverage the expressiveness of high-$k$ trees without requiring a prohibitively large $L$ remains an important direction for future research.

### F.7 EFFECTS OF THE ORDER ($p$)

We evaluate the sensitivity of our method to the hyperparameter $p$ on a gradient flow task with the 25 Gaussians and Gaussian 20d datasets. For these experiments, we use $L = 25$ trees and $k = 4$ lines, with learning rates set to $0.005$ and $0.05$, respectively. Figure 7 plots the log $W_2$ distance between the source and target distributions over 1000 steps for several choices of $p$.

The results reveal a clear trade-off. Very high values of $p$ can lead to unstable training and divergence, as the associated gradients can become excessively large. Conversely, lower values of $p$ generally lead to more stable training dynamics and consistently strong performance. However, a moderately high $p$ can also be beneficial, as the increased variance in the training dynamics can help the model escape local optima. This is supported by our main experiments in Section 4, where a larger $p$ sometimes yields superior results. Based on this analysis, we recommend selecting $p$ within the range $[1.0, 2.0]$ as a robust starting point for tuning.

**Theoretical Analysis.** Values of $p > 2$ introduce significant optimization challenges due to the extreme scaling of the structural weights. The weighting term $\hat{w}(x)^{1-p}$ contains a negative exponent that grows in magnitude as $p$ increases. This creates a loss landscape where weights can shift drastically based on the subtree volume: weights for edges near the root can vanish entirely (causing a loss of global structural guidance), while weights for edges with small volumes can grow disproportionately large (if the effective weight base is small), leading to exploding gradients. This numerical instability makes the optimization difficult to tune and prone to divergence.

The specific success of $p = 2$ likely stems from its unique geometric and computational balance. Geometrically, gradients are linear with respect to the error ($\Delta^{2-1} = \Delta$), providing the most consistent and stable signal across different scales of error. Computationally, $p = 2$ is a special case where the edge coefficient $\beta_e$ takes a logarithmic form $\log(1 + \frac{w_e}{1+\omega(\gamma_e)})$, in contrast to the polynomial form for other values. This logarithmic scaling naturally compresses the dynamic range of the spatial weights. It prevents the bias against global structure from becoming too extreme, avoiding the vanishing weight problem, and allows $p = 2$ to effectively prioritize fine-grained local details (leaves) while retaining sufficient sensitivity to global alignment (root).

In this paper, we prioritize gradient-based optimization, which is the most critical application of sliced distances. As detailed in F.8, setting $p > 1$ generally yields smoother gradients, making it more suitable for optimization tasks. Conversely, for $p = 1$, the weighting function emphasizes global discrepancies, and the transport cost follows the $L_1$ norm. These properties render the $p = 1$ setting robust to local outliers and effective for estimating global changes, which can be particularly beneficial for tasks involving noisy data.

### F.8    ANALYSIS OF ADVANTAGES OF HIGHER ORDER ($p > 1$)

TS-Sobolev is motivated by the need for a computationally efficient metric of order $p > 1$. While the standard Tree-Sliced Wasserstein distance lacks a tractable solution for orders $p > 1$, these higher orders is important for gradient-based learning, as $p$-Wasserstein metrics offer smoother gradients compared to the $p = 1$ case (Peyré et al., 2019). Beyond simply enabling tractability, our analysis reveals that TS-Sobolev ($p > 1$) introduces distinct advantages over standard $p$-Wasserstein. Overall, the better performance of TS-Sobolev stems from two complementary mechanisms: the improved optimization landscape inherent to the $L^p$ cost function and the preservation of fine-grained features introduced by the weighting function.

**Improved Optimization Landscape.**    The choice of $p$ dictates the convexity and smoothness of the underlying optimization objective. Let $\mathcal{L}(\Delta_e) := |\Delta_e|^p = |\mu(\gamma_e) - \nu(\gamma_e)|^p$, as in Equation (5), denote the unweighted transport loss associated with the mass discrepancy $\Delta_e$ on a given edge. For $p = 1$, the Tree-Sliced Wasserstein (TSW) distance behaves analogously to an $L^1$ loss. This formulation lacks strict convexity, which implies that optimization problems solving for measures $\mu$ or $\nu$ may admit non-unique minimizers, leading to potential instability (Santambrogio, 2015; Villani, 2003). Furthermore, the gradient magnitude for $p = 1$ remains constant ($|\nabla\mathcal{L}| \propto 1$, i.e., proportional to a constant) regardless of the proximity to the target, often leading to oscillations around the optimum unless the learning rate is carefully annealed. In contrast, for $p > 1$, the cost function becomes strictly convex, ensuring unique geodesics and well-conditioned gradient signals (Peyré et al., 2019). Crucially, the gradient magnitude scales with the transport cost, following $|\nabla\mathcal{L}| \propto |\Delta_e|^{p-1}$. This property ensures that gradients are large when distributions are distinct and vanish smoothly as $\Delta_e \to 0$, facilitating stable fine-tuning and convergence.

These better optimization characteristics directly translate into improved performance in the Gradient Flow experiments across both Euclidean and Spherical benchmarks in Section 4.

**Preserving Fine-Grained Structure.**    Beyond the optimization benefits inherent to standard $W_p$ metrics, TS-Sobolev employs a weighting mechanism that uniquely prioritizes the preservation of fine-grained features. As derived in Equation 4, the metric minimizes a cost weighted by the term $\hat{w}(x)^{1-p}$, where $\hat{w}(x) = 1 + \omega(\Lambda(x))$ represents the weight of the subtree rooted at $x$. For $p > 1$, this weighting factor decays as the subtree size $\omega(\Lambda(x))$ increases. Consequently, this mechanism downscale dominant gradients arising from the root and upper levels of the tree (where $\omega(\Lambda(x))$ is large), preventing global mass shifts from overwhelming the optimizer. Conversely, nodes deeper in the tree (near the leaves) possess smaller subtree weights, resulting in substantially larger values for $\hat{w}(x)^{1-p}$. This effectively concentrates the optimization signal on minimizing local discrepancies, ensuring the capture and preservation of fine-grained details.

In image generation, fine-grained details correspond to high-frequency features such as intricate textures and sharp edges. Consequently, this capacity to prioritize local feature details is a key factor driving the enhanced sample quality and sharpness observed in our large-scale diffusion training.

To further demonstrate the ability to capture fine-grained high-frequency signals, we conducted a controlled experiment using a 1D synthetic signal composed of distinct frequency modes. We defined a target probability density $p(x)$ on the domain $[0, 1]$ as a mixture of sine waves:

$$p(x) \propto 1 + 0.5\sin(2\pi \cdot k_{low} \cdot x) + 0.3\sin(2\pi \cdot k_{high} \cdot x) \tag{97}$$

where $k_{low} = 2$ represents low-frequency signals and $k_{high} = 20$ represents high-frequency signals. We initialized $N = 10000$ particles from a uniform distribution $\mathcal{U}[0, 1]$ and optimized their positions via gradient flow to minimize the distance to $p$, comparing TSW against TS-Sobolev$_2$. To quantify the capture of frequency modes, we computed the Discrete Fourier Transform (DFT), denoted as $\mathcal{F}$, of both the particle density $\hat{p}$ and the target $p$. We calculated the relative spectral error as the magnitude of the difference between their spectral components: low-frequency error $|\mathcal{F}(p)[k_{low}] - \mathcal{F}(\hat{p})[k_{low}]|$ and high-frequency error $|\mathcal{F}(p)[k_{high}] - \mathcal{F}(\hat{p})[k_{high}]|$.

The quantitative results are summarized in Table 12, and the error trajectories are visualized in Figure 8. In the low-frequency regime, both metrics perform similarly. At final, TSW achieves a low-frequency error of 34.68 compared to 33.94 for TS-Sobolev, indicating both effectively capture global structure. However, a significant disparity emerges in the high-frequency regime. While the high-frequency error for TSW plateaus, the error for TS-Sobolev$_2$ consistently decreases, reaching 14.16 compared to 20.15 for TSW. This clear trend demonstrates that TS-Sobolev$_2$ captures high-

Table 12: Low-Frequency and High-Frequency Error ($\downarrow$) over 10 runs.

| Iter | Low Frequency Error | | High Frequency Error | |
|---|---|---|---|---|
| | TSW | TS-Sobolev$_2$ | TSW | TS-Sobolev$_2$ |
| 250 | $48.18 \pm 1.45$ | $\mathbf{47.80 \pm 1.27}$ | $\mathbf{25.24 \pm 2.06}$ | $25.29 \pm 2.55$ |
| 500 | $44.55 \pm 2.12$ | $\mathbf{43.98 \pm 1.80}$ | $21.01 \pm 1.94$ | $\mathbf{20.45 \pm 2.18}$ |
| 750 | $39.96 \pm 2.46$ | $\mathbf{39.25 \pm 2.24}$ | $19.24 \pm 0.90$ | $\mathbf{16.05 \pm 1.50}$ |
| 1000 | $34.68 \pm 2.75$ | $\mathbf{33.94 \pm 2.55}$ | $20.15 \pm 1.39$ | $\mathbf{14.16 \pm 1.15}$ |

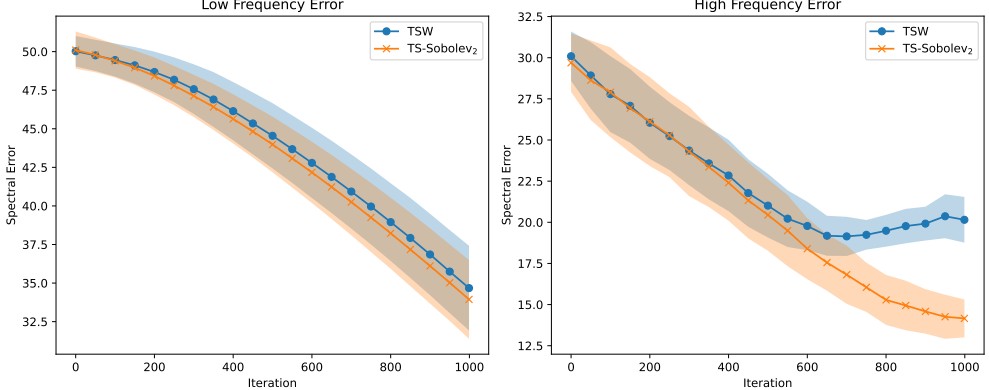

Figure 8: The evolution of error for low-frequency ($k = 2$) and high-frequency ($k = 20$) modes during gradient flow. While both methods reduce low-frequency error at a similar rate, TS-Sobolev (Orange) converges significantly better on the high-frequency component than TSW (Blue).

frequency signals significantly better than TSW, confirming our theoretical analysis regarding the preservation of fine-grained structure.

### F.9 HARDWARE SETTINGS

All experiments were conducted on a single NVIDIA A100 (40GB) GPU, with the exception of the denoising diffusion experiments, which were executed in parallel across two NVIDIA H100 GPUs.

## G BROADER IMPACTS

The ability to accurately and efficiently compute distances between complex probability distributions is a foundational challenge in many scientific and industrial fields. The TS-Sobolev framework presented in this work contributes a new tool for this task, with potential for broad societal impact. Applications span data-driven science, where more robust distributional comparison could refine medical image analysis for diagnostics, and generative AI, where it may enable the development of higher-fidelity models for creative content. A distinct advantage of our approach is its suitability for dynamic-support measures, a critical capability for real-time systems in domains like financial modeling, logistics, and environmental science. By advancing this fundamental computational primitive—the comparison of measures—our work can help foster progress across a range of applications reliant on sophisticated data analysis.

