# OpenReview forum: "Tree-sliced Sobolev IPM"
_ICLR.cc/2026/Conference — ICLR 2026 Poster_

### Official Review · Reviewer_yezB · 2025-10-27

**Soundness:** 3
**Presentation:** 2
**Contribution:** 3
**Rating:** 6
**Confidence:** 4

**Summary:**

In this work, the authors proposed a new type of tree-sliced metric to capture the $p$ -Wasserstein distance between two probability measures defined either in $\mathbb{R}^d$ or on a hypersphere. The main motivation is that recent advances in tree-sliced Wasserstein (TSW) [1,2] only provide closed-form expressions for $p = 1$, which is quite limited. On the other hand, the regularized Sobolev Integral Probability Metric -- Sobolev IPM -- (for any order $p \geq 1$) between two distributions on trees is known in closed form [3]. The idea is as follows: one randomly samples a tree structure (a slice) based on a procedure in [2], then computes the closed-form regularized Sobolev IPM (of order $p$) between two distributions within that tree structure. This process is repeated multiple times to obtain a Monte Carlo estimation of the proposed metric which is defined as the expectation of this MC estimator.

The authors showed that the proposed metric is indeed a valid metric (e.g., symmetric and satisfying the triangle inequality). When $p = 1$, it reduces to the tree-sliced Wasserstein metric, and for $p > 1$, it serves as a lower bound of TSW.

In the experiments, the proposed metric was applied in several contexts: gradient flows in either Euclidean space or on a hypersphere, as part of the objective function in training diffusion models, and in topic modeling. In most cases, the proposed metric outperforms other Wasserstein-based metrics and demonstrates improvements in downstream tasks.

**References**

[1] Tran, V. H., Pham, T., Tran, T., Le, T., & Nguyen, T. M. (2024). Tree-sliced Wasserstein distance on a system of lines. arXiv e-prints, arXiv-2406.

[2] Tran, H. V., Nguyen, K. N., Pham, T., Chu, T. T., Le, T., & Nguyen, T. M. (2025). Distance-based tree-sliced Wasserstein distance. arXiv preprint arXiv:2503.11050.

[3] Le, T., Nguyen, T., Hino, H., & Fukumizu, K. (2025). Scalable Sobolev IPM for Probability Measures on a Graph. arXiv preprint arXiv:2502.00737.

**Strengths:**

- The idea of combining tree-sliced Wasserstein with the Sobolev IPM to handle the general p-Wasserstein distance is novel and provides a practical way to approximate the $p$-Wasserstein distance in practice.

- The proposed approach introduces minimal computational overhead compared to existing Sliced or Tree-Sliced Wasserstein methods.

- The experiments are extensive and consistently demonstrate improved performance over previous works.

**Weaknesses:**

- The necessity of approximating the p-Wasserstein distance is not clearly motivated. It would be helpful to clarify what specific advantages the p-Wasserstein distance offers over the 1-Wasserstein distance, which the existing tree-sliced Wasserstein approach is already able to capture.

- Some experimental comparisons may favor the proposed method. For example, in the gradient flow experiment on the sphere (and possibly also in the Euclidean space), using the 2-Wasserstein distance as the evaluation criterion could naturally benefit the proposed method over Tree-Sliced Wasserstein [2], which is inherently based on the 1-Wasserstein distance. I wonder what the results look like if you used 1-Wasserstein distance as the criterion instead.

- The writing could be improved for clarity and structure. For instance, the definition of the regularized Sobolev IPM should be presented in the main text, as it is a central component of the proposed method. Additionally, the transition from “Tree Metric Spaces” to “Tree-Sliced Wasserstein Distance” is somewhat abrupt and may confuse readers. For example, the paragraph on “Tree Metric Spaces” might lead one to believe that the space consists solely of nodes, while in “Tree-Sliced Wasserstein Distance,” the entire tree structure (including edges) is considered. Improving the connection between these sections and providing a smoother introduction would enhance readability.

**Reference**

[2] Tran, H. V., Nguyen, K. N., Pham, T., Chu, T. T., Le, T., & Nguyen, T. M. (2025). *Distance-based tree-sliced Wasserstein distance.* arXiv preprint arXiv:2503.11050.

**Questions:**

- In 2D, when generating a system of lines, they inevitably intersect, allowing one to remove some intersection points to construct a tree. I wonder how this idea extends to higher dimensions ($d > 2$), where such lines generally do not intersect.

- Is the method applicable to more general Riemannian manifolds beyond the hypersphere? In principle, one could sample geodesics in those manifolds, so it would be interesting to see whether the approach can be generalized.

- As mentioned before, it would be helpful to clarify why we really need the p-Wasserstein distance, and in what ways the 1-Wasserstein distance falls short.

---

> ### Author Response · Authors · 2025-11-24
> **Reply [1/4]**
>
> We thank the Reviewer for their feedback. To ensure clarity and coherence, we have grouped related questions and weaknesses into unified responses below. For a summary of the changes made, please refer to our General Response.
>
> ---
>
> **W1. The necessity of approximating the p-Wasserstein distance is not clearly motivated. It would be helpful to clarify what specific advantages the p-Wasserstein distance offers over the 1-Wasserstein distance, which the existing tree-sliced Wasserstein approach is already able to capture.**
>
> **Q3. As mentioned before, it would be helpful to clarify why we really need the $p$-Wasserstein distance, and in what ways the 1-Wasserstein distance falls short.**
>
> **Answer W1 + Q3.**  We thank the Reviewer for the suggestion regarding the intuition behind the good performance of TS-Sobolev ($p>1$) and short-coming of $p=1$.
>
> TS-Sobolev is motivated by the need for a computationally efficient metric of order $p > 1$. While the standard Tree-Sliced Wasserstein (TSW) [1] distance lacks a tractable solution for orders $p > 1$, these higher orders is important for gradient-based learning, as $p$-Wasserstein metrics offer smoother gradients compared to the $p=1$ case [2]. Beyond simply enabling tractability, our analysis reveals that TS-Sobolev ($p > 1$) introduces distinct advantages over standard $p$-Wasserstein. Overall, the better performance of TS-Sobolev stems from two complementary mechanisms: the improved optimization landscape inherent to the $L^p$ cost function and the preservation of fine-grained features introduced by the weighting function.
>
> **Improved Optimization Landscape.** The choice of $p$ dictates the convexity and smoothness of the underlying optimization objective. Let $\mathcal{L}(\Delta_e) := |\Delta_e|^p = |\mu(\gamma_e) - \nu(\gamma_e)|^p$, as in Equation (5), denote the unweighted transport loss associated with the mass discrepancy $\Delta_e$ on a given edge. For $p=1$, the Tree-Sliced Wasserstein (TSW) distance behaves analogously to an $L^1$ loss. This formulation lacks strict convexity, which implies that optimization problems solving for measures $\mu$ or $\nu$ may admit non-unique minimizers, leading to potential instability [3, 4]. Furthermore, the gradient magnitude for $p=1$ remains constant ($|\nabla \mathcal{L}| \propto 1$, i.e., proportional to a constant) regardless of the proximity to the target, often leading to oscillations around the optimum unless the learning rate is carefully annealed. In contrast, for $p > 1$, the cost function becomes strictly convex, ensuring unique geodesics and well-conditioned gradient signals [2]. Crucially, the gradient magnitude scales with the transport cost, following $|\nabla \mathcal{L}| \propto |\Delta_e|^{p-1}$. This property ensures that gradients are large when distributions are distinct and vanish smoothly as $\Delta_e \to 0$, facilitating stable fine-tuning and convergence.
>
> These better optimization characteristics directly translate into improved performance in the Gradient Flow experiments across both Euclidean and Spherical benchmarks.
>
> **Preserving Fine-Grained Structure.** Beyond the optimization benefits inherent to standard $W_p$ metrics, TS-Sobolev employs a weighting mechanism that uniquely prioritizes the preservation of fine-grained features. As derived in the Equation (4), the metric minimizes a cost weighted by the term $\hat{w}(x)^{1-p}$, where $\hat{w}(x) = 1 + \omega(\Lambda(x))$ represents the weight of the subtree rooted at $x$. For $p > 1$, this weighting factor decays as the subtree size $\omega(\Lambda(x))$ increases. Consequently, this mechanism downscale dominant gradients arising from the root and upper levels of the tree (where $\omega(\Lambda(x))$ is large), preventing global mass shifts from overwhelming the optimizer. Conversely, nodes deeper in the tree (near the leaves) possess smaller subtree weights, resulting in substantially larger values for $\hat{w}(x)^{1-p}$. This effectively concentrates the optimization signal on minimizing local discrepancies, ensuring the capture and preservation of fine-grained details.
>
> In image generation, fine-grained details correspond to high-frequency features such as intricate textures and sharp edges. Consequently, this capacity to prioritize local feature details is a key factor driving the enhanced sample quality and sharpness observed in our large-scale diffusion training.

---

> ### Author Response · Authors · 2025-11-24
> **Reply [2/4]**
>
> **Answer W1+Q3 (cont).** To further demonstrate the ability to capture fine-grained high-frequency signals, we conducted a controlled experiment using a 1D synthetic signal composed of distinct frequency modes. We defined a target probability density $p(x)$ on the domain $[0, 1]$ as a mixture of sine waves:
>
> $$
> p(x) \propto 1 + 0.5\sin(2\pi \cdot k_{low} \cdot x) + 0.3\sin(2\pi \cdot k_{high} \cdot x)
> $$
>
> where $k_{low} = 2$ represents low-frequency signals and $k_{high} = 20$ represents high-frequency signals. We initialized $N=10000$ particles from a uniform distribution $\mathcal{U}[0, 1]$ and optimized their positions via gradient flow to minimize the distance to $p$, comparing TSW against TS-Sobolev$\_2$. To quantify the capture of frequency modes, we computed the Discrete Fourier Transform (DFT), denoted as $\mathcal{F}$, of both the particle density $\hat{p}$ and the target $p$. We calculated the relative spectral error as the magnitude of the difference between their spectral components: low-frequency error $|\mathcal{F}(p)[k_{low}] - \mathcal{F}(\hat{p})[k_{low}]|$ and high-frequency error $|\mathcal{F}(p)[k_{high}] - \mathcal{F}(\hat{p})[k_{high}]|$.
>
> The quantitative results are summarized in the table below, and the error trajectories are visualized in Figure 8 in the revised paper. In the low-frequency regime, both metrics perform similarly. At final, TSW achieves a low-frequency error of $34.68$ compared to $33.94$ for TS-Sobolev, indicating both effectively capture global structure. However, a significant disparity emerges in the high-frequency regime. While the high-frequency error for TSW plateaus, the error for TS-Sobolev$_2$ consistently decreases, reaching $14.16$ compared to $20.15$ for TSW. This clear trend demonstrates that TS-Sobolev$_2$ captures high-frequency signals significantly better than TSW, confirming our theoretical analysis regarding the preservation of fine-grained structure.
>
> **Table: Low-Frequency and High-Frequency Error ($\downarrow$) over 10 runs.**
> | Iter | Low Freq Error (TSW) | Low Freq Error (TS-Sobolev$_2$) | High Freq Error (TSW) | High Freq Error (TS-Sobolev$_2$) |
> | :--- | :--- | :--- | :--- | :--- |
> | 250 | $48.18 \pm 1.45$ | $\mathbf{47.80 \pm 1.27}$ | $\mathbf{25.24 \pm 2.06}$ | $25.29 \pm 2.55$ |
> | 500 | $44.55 \pm 2.12$ | $\mathbf{43.98 \pm 1.80}$ | $21.01 \pm 1.94$ | $\mathbf{20.45 \pm 2.18}$ |
> | 750 | $39.96 \pm 2.46$ | $\mathbf{39.25 \pm 2.24}$ | $19.24 \pm 0.90$ | $\mathbf{16.05 \pm 1.50}$ |
> | 1000 | $34.68 \pm 2.75$ | $\mathbf{33.94 \pm 2.55}$ | $20.15 \pm 1.39$ | $\mathbf{14.16 \pm 1.15}$ |
>
> *(Refer to Figure 8 in the revised paper for the error trajectories visualization.)*

---

> ### Author Response · Authors · 2025-11-24
> **Reply [3/4]**
>
> **W2. Some experimental comparisons may favor the proposed method. For example, in the gradient flow experiment on the sphere (and possibly also in the Euclidean space), using the 2-Wasserstein distance as the evaluation criterion could naturally benefit the proposed method over Tree-Sliced Wasserstein [2], which is inherently based on the 1-Wasserstein distance. I wonder what the results look like if you used 1-Wasserstein distance as the criterion instead.**
>
> **Answer W2.** We thank the Reviewer for this constructive suggestion. To verify our performance gains, we re-evaluated the gradient flow experiments using the **1-Wasserstein ($W_1$) distance**. As detailed in Table 1, the proposed method demonstrates significant improvements in the Euclidean setting under $W_1$ evaluation; notably, on the high-dimensional Gaussian 30d dataset, TS-Sobolev ($p=1.5$) achieves a final $W_1$ distance of **6.69**, substantially outperforming the strongest baseline, Db-TSW (16.1). Furthermore, in the spherical setting (Table 2), STS-Sobolev ($p=2$) achieves the lowest final log $W_1$ distance of **-2.158**, surpassing both STSW (-2.082) and the strongest sliced baseline, ARI-S3W (-2.136).
>
> **Table 1.** Average 1-Wasserstein distance ($W_1$) on Gaussian 30d (Euclidean)
>
> | Method | Iter 500 | Iter 1000 | Iter 1500 | Iter 2000 | Iter 2500 |
> | :--- | :--- | :--- | :--- | :--- | :--- |
> | SW | 23.5 | 23.2 | 22.9 | 22.6 | 22.3 |
> | SWGG | 22.6 | 22.7 | 22.7 | 22.7 | 22.7 |
> | LCVSW | 23.1 | 22.5 | 22.0 | 21.4 | 20.8 |
> | TSW-SL | 21.4 | 20.6 | 20.0 | 19.4 | 18.8 |
> | Db-TSW | 21.1 | 19.8 | 18.6 | 17.3 | 16.1 |
> | TS-Sobolev$_{1.2}$ | 21.0 | 19.5 | 18.0 | 16.4 | 14.9 |
> | TS-Sobolev$_{1.5}$ | 20.4 | 18.1 | 14.7 | 10.5 | **6.69** |
> | TS-Sobolev$_{2}$ | 22.1 | 21.2 | 20.1 | 19.2 | 18.6 |
>
> **Table 2.** Average Log 1-Wasserstein distance ($W_1$) on Mixture of vMFs (Spherical)
>
> | Method | Epoch 50 | Epoch 100 | Epoch 150 | Epoch 200 | Epoch 250 |
> | :--- | :--- | :--- | :--- | :--- | :--- |
> | SSW | -1.477 | -1.740 | -1.870 | -1.951 | -2.008 |
> | S3W | -1.213 | -1.373 | -1.451 | -1.486 | -1.509 |
> | RI-S3W (1) | -1.225 | -1.479 | -1.602 | -1.672 | -1.722 |
> | RI-S3W (5) | -1.423 | -1.662 | -1.782 | -1.863 | -1.913 |
> | ARI-S3W | -1.527 | -1.792 | -1.950 | -2.057 | -2.136 |
> | STSW | -1.530 | -1.812 | -1.969 | -2.046 | -2.082 |
> | STS-Sobolev$_{1.5}$ | -1.842 | -2.051 | -2.097 | -2.151 | -2.139 |
> | STS-Sobolev$_2$ | -1.814 | -2.054 | -2.120 | -2.150 | **-2.158** |
>
> ---
>
> **W3. The writing could be improved for clarity and structure. For instance, the definition of the regularized Sobolev IPM should be presented in the main text, as it is a central component of the proposed method. Additionally, the transition from “Tree Metric Spaces” to “Tree-Sliced Wasserstein Distance” is somewhat abrupt and may confuse readers. For example, the paragraph on “Tree Metric Spaces” might lead one to believe that the space consists solely of nodes, while in “Tree-Sliced Wasserstein Distance,” the entire tree structure (including edges) is considered. Improving the connection between these sections and providing a smoother introduction would enhance readability.**
>
> **Answer W3.** We sincerely thank the Reviewer for their constructive feedback on the paper’s clarity and structure. We have revised the manuscript to address these points:
> - We acknowledge that while the formulation was present in the original text (Equation 4), it was not sufficiently highlighted. We have revised the formatting and structure of the main text to make this definition stand out as a central component of our method.
> - We have rewritten the transition between "Tree Metric Spaces" (Section 2.1) and the "Tree-Sliced Framework" (Section 2.2) to ensure a smoother logical flow. We specifically clarified the relationship between the theoretical tree metric definitions and their application in the Tree-Sliced framework, explicitly noting the role of the continuous tree structure (including edges), to prevent confusion.

---

> ### Author Response · Authors · 2025-11-24
> **Reply [4/4]**
>
> **Q1. In 2D, when generating a system of lines, they inevitably intersect, allowing one to remove some intersection points to construct a tree. I wonder how this idea extends to higher dimensions $(d> 2)$, where such lines generally do not intersect.**
>
> **Answer Q1.** The tree in higher dimensions can be practically constructed by sampling an intersection point then sampling line that go through that intersection. There are two popular sampling process that both based on this idea, details about these are given in Appendix C.1 and visualized in Figure 3. We give high level overview two approaches:
>
> - Chain-structured: Sampling the first line $\theta_1$, selecting a point $x_1$ on the first line, sampling a line $\theta_2$ that goes through $x_1$ and repeat.
> - Concurrent-line: Sampling a common intersection $x$ of all lines (the root of tree), then sample all lines going through that intersection.
> We note that we adapt the concurrent-line tree structure.
>
> We would like to further clarify in our experiments that the tree construction is done with general $d \geq 2$, matching the dimension of data that TS-Sobolev is operating on. Tree systems with high dimensional $d$ is used throughout the paper (e.g. $d = 30$ for Gradient flow experiment with Gaussian 30d while d ~ 3000 in Diffusion Model training).
>
> ---
>
> **Q2. Is the method applicable to more general Riemannian manifolds beyond the hypersphere? In principle, one could sample geodesics in those manifolds, so it would be interesting to see whether the approach can be generalized.**
>
> **Answer Q2.** We thank the Reviewer for this interesting suggestion. Yes, in principle, our framework can be extended to general Riemannian manifolds, provided that the following components are appropriately adapted:
>
> - The tree systems must be defined within the manifold, for instance, by sampling and constructing unions of geodesics as the reviewer suggested.
> - The Radon-type transform requires two key elements: a splitting map and a projection function. These can be formulated using geodesic distances or other intrinsic manifold properties. Similar constructions have been explored in prior work on Sliced-Wasserstein variants for hyperbolic spaces and other manifolds [5, 6, 7].
> - Rigorous analysis is required to establish metric properties (such as injectivity) and to estimate computational complexity in these general settings.
>
> While feasible, establishing these theoretical foundations is non-trivial and lies beyond the scope of the current work. We will include a discussion regarding this potential generalization in the final version of the paper.
>
> ---
> We sincerely thank the Reviewer for their constructive feedback, which has helped improve our manuscript. We have revised the paper to address these points. We hope our response clarifies the concerns raised. We remain available to address any further questions during the discussion period.
>
> *References.*
>
> [1] Tran et al. Distance-based Tree-Sliced Wasserstein Distance. ICLR 2025.
>
> [2] Peyré et al. Computational optimal transport: With applications to data science. Foundations and Trends in Machine Learning 2019.
>
> [3] Santambrogio. Optimal Transport for Applied Mathematicians: Calculus of Variations, PDEs, and Modeling. Springer International Publishing 2015.
>
> [4] Villani. Topics in Optimal Transportation. American Mathematical Society 2003.
>
> [5] Bonet et al. Hyperbolic Sliced-Wasserstein via Geodesic and Horospherical Projections. arXiv:2211.10066.
>
> [6] Bonet et al. Sliced-Wasserstein Distances and Flows on Cartan-Hadamard Manifolds. JMLR 2025.
>
> [7] Bonet et al. Sliced-Wasserstein on Symmetric Positive Definite Matrices for M/EEG Signals. ICML 2023.

---

### Official Review · Reviewer_ZHaq · 2025-11-01

**Soundness:** 3
**Presentation:** 3
**Contribution:** 3
**Rating:** 6
**Confidence:** 4

**Summary:**

Tree-Sliced Wasserstein (TSW) practical use is constrained to the 1-Wasserstein because $W_p$ ($p>1$) for tree is costly. This paper proposes using regularized Sobolev IPM which has a computationally efficient closed-form solution on trees. THis recovers TSW exactly at $p = 1$ and is upper-bounded by TSW for $p > 1$, while maintaining the same $\mathcal{O} (L k n log n)$ complexity.

**Strengths:**

- A key achievement is generalizing to $p > 1$ without sacrificing computational efficiency. The closed-form expression for $\hat{\mathcal{S}}_p$ on discrete measures (Theorem B.6, Eq. 28) relies on edge coefficients $\beta_e$ that depend on $p$ but are efficiently pre-computable, adding negligible $\mathcal{O}(Lkn)$ overhead. Thus TS-Sobolev retains the dominant $O(Lkn \log n)$ complexity of TSW for \textit{any} $p \ge 1$. This claim is validated by the empirical runtime analysis (Appendix F.1, Figs 4-6), showing near-identical wall-clock times regardless of $p$.
- Good experimental results. For example,
    - Achieves faster convergence and lower final $W_2$ error compared to TSW ($p=1$) and SW variants in Euclidean settings (Table 1). Similarly outperforms spherical TSW (STSW) in spherical gradient flows (Table 4).
    - with DDGAN training gives sota FID scores on CIFAR-10, significantly improving upon TSW-based DDGANs with identical training time per epoch
    - Strong performance in spherical SSL (Table 3) and achieves the highest topic coherence ($C_V$) in both Euclidean and Spherical topic modeling benchmarks (Table 5).

**Weaknesses:**

- The main justification for using $\hat{\mathcal{S}}_p$ is its tractability. However, the paper offers limited intuition on the gain or loss by using this specific Sobolev-based IPM instead of $W_p$ on trees for $p>1$. $\hat{\mathcal{S}}_p$ relates to the weighted $L^{p'}$ norm of the critic function's derivative on the tree. How does minimizing this discrepancy differ fundamentally from minimizing transport cost ($d(x,y)^p$)? Does $\hat{\mathcal{S}}_p$ emphasize smoothness or local variations differently than $W_p$?

- The closed-form tractability of $\hat{\mathcal{S}}_p$ comes from a specific weighting function $\hat{w}(x) = 1 + \omega(\Lambda(x))$ inherent in the underlying Sobolev norm equivalence on trees (Theorem B.3). This weight depends on the measure (length) of the subtree $\Lambda(x)$ below point $x$ relative to the root (Eq. 22). It is not clear to the reviewer the geometric implications of this structural weighting. For example, does this make $\hat{\mathcal{S}}_p$ (and thus TS-Sobolev) inherently more sensitive to distributional differences occurring deeper in the tree projection (larger $\omega(\Lambda(x))$) versus closer to the root?

- The experiments consistently use the concurrent-line tree structure [cite: lines 1097-1102] and softmax splitting map, following configurations from recent TSW paper. While sensible for direct comparison, this limits the exploration of how TS-Sobolev performs under different structural assumptions. Appendix F.6 analyzes the $L$ vs. $k$ trade-off but doesn't compare tree topologies (e.g., concurrent vs chain or splitting maps within the TS-Sobolev context. The observed benefit of $p>1$ might interact with these choices.

**Questions:**

- Can the authors give more insight, perhaps related to the Sobolev IPM formulation involving derivatives, on why $p>1$ might lead to better optimization dynamics (e.g., smoother gradients, better conditioning) compared to $p=1$ (TSW), as suggested by the gradient flow results?
- Could the authors please comment on the practical effect of the structural weighting $\hat{w}(x) = 1 + \omega(\Lambda(x))$? Does this induce spatial sensitivity bias on the tree slices and how would it interact with the choice of $p$ for example?
- How robust are the benefits of $p>1$ expected to be under different tree generation schemes (e.g., chain structures) or alternative splitting maps?
- Appendix F.7 provides valuable empirical analysis showing $p \in [1.0, 2.0]$ is effective and higher $p$ can be unstable . Can the authors offer any further heuristic or theoretical guidance? For instance, does the optimal $p$ appear correlated with data dimensionality or the nature of the task (e.g., generation vs. optimization)? Why might $p=2$ perform particularly well across several diverse tasks (Tables 2, 4, 5)?

---

> ### Author Response · Authors · 2025-11-24
> **Reply [1/5]**
>
> We thank the Reviewer for their feedback. To ensure clarity and coherence, we have grouped related questions and weaknesses into unified responses below. For a summary of the changes made, please refer to our General Response.
>
> ---
>
> **W1. The main justification for using $\hat{\mathcal{S}}_p$ is its tractability. However, the paper offers limited intuition on the gain or loss by using this specific Sobolev-based IPM instead of $W_p$ on trees for $p > 1$. $\hat{\mathcal{S}}_p$ relates to the weighted $L^p$ norm of the critic function's derivative on the tree. How does minimizing this discrepancy differ fundamentally from minimizing transport cost $(d(x, y))^p$? Does $\hat{\mathcal{S}}_p$ emphasize smoothness or local variations differently than $W_p$?**
>
> **Answer W1.** We thank the Reviewer for this insightful question regarding the intuition of our proposed metric. To clarify the fundamental difference between minimizing the Sobolev discrepancy versus the transport cost, it is essential to view these metrics through the Integral Probability Metric (IPM) framework.
>
> An IPM measures the distance between distributions $\mu$ and $\nu$ by finding a "critic function" $f$ within a function class $\mathcal{F}$ that maximizes the discrepancy:
>
> \begin{equation}
> \gamma_{\mathcal{F}}(\mu, \nu) = \sup_{f \in \mathcal{F}} \left| \int f(x) \mu(dx) - \int f(y) \nu(dy) \right|.
> \end{equation}
>
> Standard Optimal Transport ($W_p$) is a special case of IPM where $\mathcal{F}$ is defined by constraining only the gradient of the critic. For instance, the 1-Wasserstein distance is obtained when $\mathcal{F}\_W := \{f : | f(x) - f(y) | \le d\_{\mathcal{T}}(x, y) \}$, where $d\_{\mathcal{T}}$ is the tree metric:
>
> $$
> W_1(\mu, \nu) = \sup_{f \in \mathcal{F}\_{W}} \vert \int_{\mathcal{T}} f(x) \mu(dx) - \int_{\mathcal{T}} f(y) \nu(dy)\vert= \int_{\mathcal{T}} |\mu(\Lambda(x)) - \nu(\Lambda(x))| \omega(dx).
> $$
>
> In essence, $W_1$ is equivalent to an IPM where the critic function is 1-Lipschitz (i.e., $|f'| \le 1$ everywhere). Thus, $W_1$ imposes a uniform gradient constraint on the critic function across the entire tree, focusing purely on the transport of mass.
>
> In contrast, the Sobolev IPM $(\mathcal{S_p})$ defines $\mathcal{F}$ by constraining both the critic's magnitude and its gradient via the Sobolev norm $\Vert f \Vert_{W^{1,p}} = (\Vert f\Vert_{L^p}^p + ||f'||_{L^p}^p)^{1/p}$. However, computing this standard Sobolev IPM is computationally intractable. The Regularized Sobolev IPM $(\hat{\mathcal{S}_p})$
>
> leverages the insight that on a rooted tree, the value of a function $f(x)$ is determined by the accumulation of its gradients along the path from the root. Consequently, the Sobolev norm constraint (which limits both value and gradient) is mathematically equivalent to a single weighted gradient constraint: $\Vert f' \Vert_{L^p_{\hat{w}}} \le 1$. The weight $\hat{w}$ implicitly controls the function's magnitude by penalizing gradients more heavily on edges that impact larger subtrees. This yields the following closed-form expression for $p \ge 1$:
>
> $$\hat{\mathcal{S}}\_p(\mu, \nu)^p = \int\_{\mathcal{T}} \hat{w}(x)^{1-p} |\mu(\Lambda(x)) - \nu(\Lambda(x))|^p \omega(dx),$$
>
> where $\hat{w}(x) = 1 + \omega(\Lambda(x))$ is a weight function.
>
> This structural weighting term, $\hat{w}(x)^{1-p}$, makes $\hat{\mathcal{S}}_p$ spatially adaptive. Unlike $W_1$, which applies a uniform transport cost, $\hat{\mathcal{S}}_p$ (for $p>1$) adjusts its sensitivity based on the tree structure. Near the root (large subtree volume), the weight enforces strict smoothness, making the metric robust to global mass shifts. Near the leaves (small subtree volume), the constraint relaxes, allowing the critic to change rapidly to capture fine-grained local variations. Consequently, $\hat{\mathcal{S}}_p$ imposes a structure-aware smoothness that prioritizes matching local details while being more forgiving of global displacements, as further analyzed in response to **W2 + Q2**.

---

> ### Author Response · Authors · 2025-11-24
> **Reply [2/5]**
>
> **W2. The closed-form tractability of $\hat{\mathcal{S}}_p$ comes from a specific weighting function $\hat{w}(x) = 1 + \omega(\Lambda(x))$ inherent in the underlying Sobolev norm equivalence on trees (Theorem B.3). This weight depends on the measure (length) of the subtree $\Lambda(x)$ below point $x$ relative to the root (Eq. 22). It is not clear to the reviewer the geometric implications of this structural weighting. For example, does this make $\hat{\mathcal{S}}_p$ (and thus TS-Sobolev) inherently more sensitive to distributional differences occurring deeper in the tree projection (larger $\omega(\Lambda(x))$) versus closer to the root?**
>
> **Q2. Could the authors please comment on the practical effect of the structural weighting $\hat{w}(x) = 1 + \omega(\Lambda(x))$? Does this induce spatial sensitivity bias on the tree slices and how would it interact with the choice of $p$, for example?**
>
> **Answer W2 + Q2.** We thank the Reviewer for highlighting this crucial aspect of our proposed metric. To clarify the practical effect of the structural weighting, we examine the closed-form expression for the Regularized Sobolev IPM:
>
> $$
> \hat{\mathcal{S}}\_p(\mu, \nu)^p = \int\_{\mathcal{T}} \hat{w}(x)^{1-p} |\mu(\Lambda(x)) - \nu(\Lambda(x))|^p \omega(dx).
> $$
>
> Here, the weight function is defined as $\hat{w}(x) = 1 + \omega(\Lambda(x))$, where $\Lambda(x)$ denotes the subtree rooted at $x$ and $\omega(\Lambda(x))$ represents its total length. For $p > 1$, the weighting term $\hat{w}(x)^{1-p}$ decreases as the subtree length increases. This introduces a specific spatial sensitivity bias:
>
> - Closer to the Root: Points near the root have a large subtree length $\omega(\Lambda(x))$. Consequently, the weight $\hat{w}(x)^{1-p}$ is small. This means the metric places less emphasis on transport costs in the upper levels of the tree, making it less sensitive to global shifts in the mass distribution.
> - Deeper in the Tree: Points deeper in the tree (near the leaves) have a small subtree length. Consequently, the weight $\hat{w}(x)^{1-p}$ is large. This places a greater emphasis on transport costs occurring at the leaves, making the metric more sensitive to local variations.
>
> The parameter $p$ controls this sensitivity bias. As $p$ increases, the weight assigned to edges with large subtree lengths decays, effectively concentrating the metric's sensitivity on the fine-grained details at the leaves. Practically, a loss that minimizes TS-Sobolev with high $p$ will prioritize the alignment of fine-grained local features while increasingly tolerant of large-scale displacements in the global structure.
>
> In image generation, fine-grained details correspond to high-frequency features such as intricate textures and sharp edges. Consequently, this capacity to prioritize local feature details is a key factor driving the enhanced sample quality and sharpness observed in our large-scale diffusion training.
>
> To further demonstrate the ability to capture fine-grained high-frequency signals, we conducted a controlled experiment using a 1D synthetic signal composed of distinct frequency modes. We defined a target probability density $p(x)$ on the domain $[0, 1]$ as a mixture of sine waves:
>
> $$
> p(x) \propto 1 + 0.5\sin(2\pi \cdot k_{low} \cdot x) + 0.3\sin(2\pi \cdot k_{high} \cdot x)
> $$
>
> where $k_{low} = 2$ represents low-frequency signals and $k_{high} = 20$ represents high-frequency signals. We initialized $N=10000$ particles from a uniform distribution $\mathcal{U}[0, 1]$ and optimized their positions via gradient flow to minimize the distance to $p$, comparing TSW against TS-Sobolev$_2$. To quantify the capture of frequency modes, we computed the Discrete Fourier Transform (DFT), denoted as $\mathcal{F}$, of both the particle density $\hat{p}$ and the target $p$. We calculated the relative spectral error as the magnitude of the difference between their spectral components: low-frequency error $|\mathcal{F}(p)[k\_{low}] - \mathcal{F}(\hat{p})[k\_{low}]|$ and high-frequency error $|\mathcal{F}(p)[k\_{high}] - \mathcal{F}(\hat{p})[k\_{high}]|$.
>
> The quantitative results are summarized in the table below, and the error trajectories are visualized in Figure 8 in the revised paper. In the low-frequency regime, both metrics perform similarly. At final iterations, TSW achieves a low-frequency error of $34.68$ compared to $33.94$ for TS-Sobolev, indicating both effectively capture global structure. However, a significant disparity emerges in the high-frequency regime. While the high-frequency error for TSW plateaus, the error for TS-Sobolev$_2$ consistently decreases, reaching $14.16$ compared to $20.15$ for TSW. This clear trend demonstrates that TS-Sobolev$_2$ captures high-frequency signals significantly better than TSW, confirming our theoretical analysis regarding the preservation of fine-grained structure.

---

> ### Author Response · Authors · 2025-11-24
> **Reply [3/5]**
>
> **Answer W2+Q2 (cont).**
>
> **Table: Low-Frequency and High-Frequency Error ($\downarrow$) over 10 runs.**
> | Iter | Low Freq Error (TSW) | Low Freq Error (TS-Sobolev$_2$) | High Freq Error (TSW) | High Freq Error (TS-Sobolev$_2$) |
> | :--- | :--- | :--- | :--- | :--- |
> | 250 | $48.18 \pm 1.45$ | $\mathbf{47.80 \pm 1.27}$ | $\mathbf{25.24 \pm 2.06}$ | $25.29 \pm 2.55$ |
> | 500 | $44.55 \pm 2.12$ | $\mathbf{43.98 \pm 1.80}$ | $21.01 \pm 1.94$ | $\mathbf{20.45 \pm 2.18}$ |
> | 750 | $39.96 \pm 2.46$ | $\mathbf{39.25 \pm 2.24}$ | $19.24 \pm 0.90$ | $\mathbf{16.05 \pm 1.50}$ |
> | 1000 | $34.68 \pm 2.75$ | $\mathbf{33.94 \pm 2.55}$ | $20.15 \pm 1.39$ | $\mathbf{14.16 \pm 1.15}$ |
>
> *(Refer to Figure 8 in the revised paper for the error trajectories visualization.)*
>
> **Q1. Can the authors give more insight, perhaps related to the Sobolev IPM formulation involving derivatives, on why $p > 1$ might lead to better optimization dynamics (e.g., smoother gradients, better conditioning) compared to $p = 1$ (TSW), as suggested by the gradient flow results?**
>
> **Answer Q1.** We thank the Reviewer for this question. The goal of achieving better optimization dynamics is the fundamental motivation for developing the tractable TS-Sobolev metric. While the standard Tree-Sliced Wasserstein (TSW) [1] distance lacks a tractable solution for orders $p > 1$, these higher orders is important for gradient-based learning, as $p$-Wasserstein metrics offer smoother gradients compared to the $p=1$ case [2]. Beyond simply enabling tractability, our analysis reveals that TS-Sobolev ($p > 1$) introduces distinct advantages over standard $p$-Wasserstein. Overall, the better performance of TS-Sobolev stems from two complementary mechanisms: the improved optimization landscape inherent to the $L^p$ cost function and the preservation of fine-grained features introduced by the weighting function.
>
> **Improved Optimization Landscape.** The choice of $p$ dictates the convexity and smoothness of the underlying optimization objective. Let $\mathcal{L}(\Delta_e) := |\Delta_e|^p = |\mu(\gamma_e) - \nu(\gamma_e)|^p$, as in Equation (5), denote the unweighted transport loss associated with the mass discrepancy $\Delta_e$ on a given edge. For $p=1$, the Tree-Sliced Wasserstein (TSW) distance behaves analogously to an $L^1$ loss. This formulation lacks strict convexity, which implies that optimization problems solving for measures $\mu$ or $\nu$ may admit non-unique minimizers, leading to potential instability [3, 4]. Furthermore, the gradient magnitude for $p=1$ remains constant ($|\nabla \mathcal{L}| \propto 1$, i.e., proportional to a constant) regardless of the proximity to the target, often leading to oscillations around the optimum unless the learning rate is carefully annealed. In contrast, for $p > 1$, the cost function becomes strictly convex, ensuring unique geodesics and well-conditioned gradient signals [2]. Crucially, the gradient magnitude scales with the transport cost, following $|\nabla \mathcal{L}| \propto |\Delta_e|^{p-1}$. This property ensures that gradients are large when distributions are distinct and vanish smoothly as $\Delta_e \to 0$, facilitating stable fine-tuning and convergence.
>
> These better optimization characteristics directly translate into improved performance in the Gradient Flow experiments across both Euclidean and Spherical benchmarks.
>
> **Preserving Fine-Grained Structure.** Beyond the optimization benefits inherent to standard $W_p$ metrics, TS-Sobolev employs a weighting mechanism that uniquely prioritizes the preservation of fine-grained features. As derived in the Equation (4), the metric minimizes a cost weighted by the term $\hat{w}(x)^{1-p}$, where $\hat{w}(x) = 1 + \omega(\Lambda(x))$ represents the weight of the subtree rooted at $x$. For $p > 1$, this weighting factor decays as the subtree size $\omega(\Lambda(x))$ increases. Consequently, this mechanism downscale dominant gradients arising from the root and upper levels of the tree (where $\omega(\Lambda(x))$ is large), preventing global mass shifts from overwhelming the optimizer. Conversely, nodes deeper in the tree (near the leaves) possess smaller subtree weights, resulting in substantially larger values for $\hat{w}(x)^{1-p}$. This effectively concentrates the optimization signal on minimizing local discrepancies, ensuring the capture and preservation of fine-grained details.
>
> Further details about the ability to capture fine-grained structure are in our response to **W2 + Q2**.

---

> ### Author Response · Authors · 2025-11-24
> **Reply [4/5]**
>
> **Q4. Appendix F.7 provides valuable empirical analysis showing $p \in [1.0, 2.0]$ is effective and higher $p$ can be unstable. Can the authors offer any further heuristic or theoretical guidance? For instance, does the optimal $p$ appear correlated with data dimensionality or the nature of the task (e.g., generation vs. optimization)? Why might $p = 2$ perform particularly well across several diverse tasks (Tables 2, 4, 5)?**
>
> **Answer Q4.** We appreciate the Reviewer's interest in the practical selection of the parameter $p$. Based on our theoretical framework and empirical observations, we can offer several heuristics regarding the instability of higher $p$ values and the specific effectiveness of $p=2$.
>
> Values of $p > 2$ introduce significant optimization challenges due to the extreme scaling of the structural weights. The weighting term $\hat{w}(x)^{1-p}$ contains a negative exponent that grows in magnitude as $p$ increases. This creates a loss landscape where weights can shift drastically based on the subtree volume: weights for edges near the root can vanish entirely (causing a loss of global structural guidance), while weights for edges with small volumes can grow disproportionately large (if the effective weight base is small), leading to exploding gradients. This numerical instability makes the optimization difficult to tune and prone to divergence.
>
> The specific success of $p=2$ likely stems from its unique geometric and computational balance. Geometrically, gradients are linear with respect to the error ($\Delta^{2-1} = \Delta$), providing the most consistent and stable signal across different scales of error. Computationally, $p=2$ is a special case where the edge coefficient $\beta_e$ takes a logarithmic form $\log(1 + \frac{w_e}{1+\omega(\gamma_e)})$, in contrast to the polynomial form for other values. This logarithmic scaling naturally compresses the dynamic range of the spatial weights. It prevents the bias against global structure from becoming too extreme, avoiding the vanishing weight problem, and allows $p=2$ to effectively prioritize fine-grained local details (leaves) while retaining sufficient sensitivity to global alignment (root).
>
> Finally, we provide intuition regarding the correlation between the optimal order $p$ and the nature of the task. In this paper, we prioritize gradient-based optimization, which is the most critical application of sliced distances. As detailed in our responses to W2 + Q2 and Q1, setting $p > 1$ generally yields smoother gradients, making it more suitable for optimization tasks. Conversely, for $p=1$, the weighting function emphasizes global discrepancies, and the transport cost follows the $L_1$ norm. These properties render the $p=1$ setting robust to local outliers and effective for estimating global changes, which can be particularly beneficial for tasks involving noisy data.

---

> ### Author Response · Authors · 2025-11-24
> **Reply [5/5]**
>
> **W3. The experiments consistently use the concurrent-line tree structure [cite: lines 1097–1102] and softmax splitting map, following configurations from recent TSW papers. While sensible for direct comparison, this limits the exploration of how TS-Sobolev performs under different structural assumptions. Appendix F.6 analyzes the $L$ vs. $k$ trade-off but does not compare tree topologies (e.g., concurrent vs. chain or splitting maps within the TS-Sobolev context). The observed benefit of $p > 1$ might interact with these choices.**
>
>
> **Q3. How robust are the benefits of $p > 1$ expected to be under different tree generation schemes (e.g., chain structures) or alternative splitting maps?**
>
> **Answer W3 + Q3.** We thank the Reviewer for this insightful comment. Theoretically, the benefits of the TS-Sobolev framework, specifically the flexibility to utilize $p > 1$, should be independent of the underlying tree structure or splitting map mechanism. These components primarily define the projection domain rather than the metric properties themselves. However, to empirically confirm this robustness, we conducted additional Gradient Flow experiments on the Gaussian 30d dataset. We evaluated all four combinations of **Chain** vs. **Concurrent** tree sampling and **Uniform** vs. **Distance-based** splitting maps. As shown in Table 1, we observe that TS-Sobolev$_{1.2}$ consistently outperforms TSW$_1$ across all four settings.
>
> **Table 1.** Average Wasserstein distance (multiplied by $10^{-1}$) between source and target distributions at iteration 2500. Results are averaged over 10 runs.
>
> | Tree Sampling | Splitting Map | TSW$_1$ | TS-Sobolev$_{1.2}$ | TS-Sobolev$_{1.5}$ | TS-Sobolev$_2$ |
> | :--- | :--- | :---: | :---: | :---: | :---: |
> | Chain | Uniform | 2.01 | **1.89** | 4.54 | 12.30 |
> | Chain | Distance | 1.56 | **1.49** | 1.88 | 5.30 |
> | Concurrent | Uniform | 1.93 | **1.83** | 3.37 | 11.10 |
> | Concurrent | Distance | 1.78 | **1.40** | 1.51 | 3.68 |
>
>
> ---
>
> We sincerely thank the Reviewer for their constructive feedback, which has helped improve our manuscript. We have revised the paper to address these points. We hope our response clarifies the concerns raised. We remain available to address any further questions during the discussion period.
>
> *References.*
>
> [1] Tran et al. Distance-based Tree-Sliced Wasserstein Distance. ICLR 2025.
>
> [2] Peyré et al. Computational optimal transport: With applications to data science. Foundations and Trends in Machine Learning 2019.
>
> [3] Santambrogio. Optimal Transport for Applied Mathematicians: Calculus of Variations, PDEs, and Modeling. Springer International Publishing 2015.
>
> [4] Villani. Topics in Optimal Transportation. American Mathematical Society 2003.

---

### Official Review · Reviewer_UFk1 · 2025-11-02

**Soundness:** 3
**Presentation:** 2
**Contribution:** 3
**Rating:** 6
**Confidence:** 3

**Summary:**

### **Summary**

This paper introduces the **Tree-Sliced Sobolev Integral Probability Metric (TS-Sobolev)** and its **spherical variant (STS-Sobolev)** — new metrics for comparing probability distributions that generalize the Tree-Sliced Wasserstein (TSW) distance to any order $p \ge 1$.

The main idea is to replace the 1-Wasserstein distance on tree metric spaces (used in TSW) with a **regularized Sobolev Integral Probability Metric (IPM)** that admits a **closed-form solution**. This substitution retains the computational efficiency of TSW (≈ $O(Lkn \log n)$) while enabling higher-order Sobolev metrics.

The authors establish key **theoretical guarantees**, including:
- Metricity and $E(d)$-invariance,
- A formal connection showing $\text{TS-Sobolev}_{p=1} = \text{TSW}$,
- Monte Carlo convergence rate $O(L^{-1/2})$.

The proposed methods are evaluated on a wide range of **Euclidean and spherical tasks**, including gradient flows, diffusion-based generative modeling (DDGAN on CIFAR-10), self-supervised learning on the sphere, and topic modeling on BBC and M10 datasets.

Across all benchmarks, TS-Sobolev and STS-Sobolev consistently **outperform Sliced and Tree-Sliced Wasserstein baselines** in accuracy or convergence rate, while maintaining similar runtime. The paper concludes that Sobolev-based slicing provides a scalable and flexible alternative to classical TSW for both Euclidean and non-Euclidean domains.

**Strengths:**

### **Strengths**

#### **Novel Theoretical Contribution**
- Introduces a closed-form **Sobolev IPM** on trees that generalizes **1-Wasserstein** while retaining efficiency.
- Provides formal proofs of **metricity**, **$E(d)$-invariance**, and **Monte Carlo convergence rate**.
- Connects **Sobolev IPM** and **tree-sliced methods** in a clean, unified framework.

#### **Scalability**
- Maintains the computational complexity $O(Lkn \log n)$ of existing TSW methods, enabling practical scalability to large datasets.
- Extensible to **spherical manifolds** via the **spherical Radon transform**.

#### **Strong Empirical Results**
- Consistently improves over both **SW** and **TSW** baselines across diverse domains.
- Demonstrates flexibility across geometries (**Euclidean** / **spherical**) and applications (gradient flows, SSL, generative modeling, topic modeling).

#### **Clarity of Exposition**
- Well-structured presentation: clear flow from background → formulation → theoretical properties → experiments.
- Figures and equations are informative, and notation aligns with prior **TSW literature**.

**Weaknesses:**

1. In the computation section, the authors claim a total complexity of $O(Lkn \log n + LKd n)$, but it seems that the tree construction cost is not explicitly included.
1.1 Moreover, the parameter $k$ (the number of lines per tree) controls a trade-off between computational efficiency and information preservation in $\mathbb{R}^d$. A detailed discussion of this trade-off would strengthen the paper. In particular, it would be useful to know how large $k$ can grow before the overall method becomes slower than classical OT solvers such as Sinkhorn, whose complexity is $O(n^2 / \epsilon)$.
1.2 Clarifying whether $k$ depends on $n$ would also help to assess the true scalability of the approach.

2. In the spherical setting, I suggest including additional sliced OT baselines for a more complete comparison — for example, “Spherical Sliced Optimal Transport” (arXiv:2411.06055) and “Stereographic Spherical Sliced Wasserstein Distances” (arXiv:2402.02345). These would provide a stronger empirical context and highlight the advantages of the proposed method under spherical geometry.

3. The Sobolev IPM involves gradient norms $|\nabla f|$, which are central to the metric’s definition. However, estimating $\nabla f$ on discrete samples and through non-smooth tree-sliced projections may introduce numerical instability or high-variance gradients. Prior work on Sobolev-based metrics (Mroueh et al., ICLR 2018; Deshpande et al., NeurIPS 2019; Korotin et al., ICML 2021) has highlighted that $\nabla f$ estimation can become unstable without smoothing or gradient regularization. The paper would benefit from analyzing or mitigating such potential instability in the proposed tree-sliced formulation.

**Questions:**

**Q1.**
Suppose the tree $\mathbb{T} \subset \mathbb{R}$, i.e., all nodes lie on a straight line. In this case, the tree-sliced construction should coincide with the original sliced OT construction. Under this 1D setting, can we claim that the regularized Sobolev IPM (Eq. (4)) or the Sobolev IPM (Eq. (3)) is equivalent to $W_p(\mu,\nu)$?
My understanding is that in 1D Eq. (4) reduces to
$$
\int_{\mathbb{R}} \big|\mu((-\infty,x]) - \nu((-\infty,x])\big|^p \, dx,
$$
which differs from the standard 1D OT definition
$$
W_p^p(\mu,\nu) = \int_0^1 |F_\mu^{-1}(t) - F_\nu^{-1}(t)|^p \, dt.
$$
If this understanding is correct, then the proposed Tree-Sliced Sobolev IPM (Def. 3.1) cannot recover the classical sliced OT formulation
$$
\int_{\mathbb{S}^{d-1}} W_p^p((P_\theta)_\# \mu, (P_\theta)_\# \nu) \, d\sigma(\theta),
$$
since $S_p^p(\mu,\nu)$ and $W_p^p(\mu,\nu)$ are not equivalent in 1D. Could the authors clarify this or specify under what conditions the two coincide?

**Q2.**
The paper motivates Sobolev IPM as more “frequency-aware” and capable of mitigating spectral bias, but the experiments mainly show performance improvements without spectral evidence. Could the authors include a more direct spectral analysis (e.g., using synthetic 1D or 2D signals with known low/high-frequency components) to verify that the proposed tree-sliced Sobolev formulation indeed captures or preserves high-frequency modes better than standard sliced OT?

---

> ### Author Response · Authors · 2025-11-24
> **Reply [1/5]**
>
> We thank the Reviewer for their feedback. To ensure clarity and coherence, we have grouped related questions and weaknesses into unified responses below. For a summary of the changes made, please refer to our General Response.
>
> ---
>
> **W1. In the computation section, the authors claim a total complexity of $O(Lkn\log(n)+LKdn)$, but it seems that the tree construction cost is not explicitly included. 1.1 Moreover, the parameter $k$ (the number of lines per tree) controls a trade-off between computational efficiency and information preservation in $\mathbb{R}^d$. A detailed discussion of this trade-off would strengthen the paper. In particular, it would be useful to know how large $k$ can grow before the overall method becomes slower than classical OT solvers such as Sinkhorn, whose complexity is $O(n^2/\epsilon)$. 1.2 Clarifying whether $k$ depends on $n$ would also help to assess the true scalability of the approach.**
>
> **Answer W1.** We thank the Reviewer for their insightful questions. Below, we address tree construction costs, comparisons with Sinkhorn, the independence of $k$, and the expressiveness trade-off.
>
> Regarding tree construction, we clarify that the complexity is $\mathcal{O}(Lkd)$, arising from sampling one root and $k$ lines in $\mathbb{R}^d$ per tree for $L$ trees. Being linear in $L, k, d$ and independent of $n$, this cost is negligible compared to the dominant $\mathcal{O}(Lkn \log n)$ projection and sorting steps. Empirically, **Table 1** confirms that tree sampling constitutes only $\approx 0.11\%$ of the total runtime ($n=256, L=10^5, k=10$). Note that the runtimes reported in our main paper already include this construction cost.
>
> **Table 1: Tree Construction Cost Analysis ($n=256, L=10^5, k=10$)**
>
> | Component | Runtime (ms) | Percentage of Total |
> | :--- | :--- | :--- |
> | Tree Sampling | $0.182 \pm 0.029$ | $0.11\%$ |
> | Total (Sampling + Compute) | $161.01 \pm 0.23$ | $100\%$ |
>
> Regarding the computational threshold relative to Sinkhorn, we note two factors. First, Sinkhorn scales quadratically $\mathcal{O}(n^2)$, whereas TS-Sobolev scales log-linearly $\mathcal{O}(n \log n)$, making TS-Sobolev inherently more scalable with respect to $n$. Second, empirical comparisons show TS-Sobolev remains faster on small datasets ($n=32$) even with inflated parameters (**Table 2**), and is roughly $7\times$ faster at moderate sizes ($n=256$) (**Table 3**). TS-Sobolev only approaches Sinkhorn's cost if $L$ and $k$ are raised to extreme values (e.g., $L=5 \times 10^5$), far exceeding standard usage ($L < 10^4, k < 10$). This efficiency is why Sliced and Tree-Sliced methods are preferred for real-world applications involving large-scale data (e.g., generative modeling), where quadratic OT solvers become computationally prohibitive.
>
> **Table 2: Runtime Comparison at $n=32$**
>
> | Method | Parameters | Runtime (ms) |
> | :--- | :--- | :--- |
> | **TS-Sobolev** | $L=10^5, k=10$ | $29.14 \pm 0.02$ |
> | **TS-Sobolev** | $L=10^5, k=50$ | $142.66 \pm 0.09$ |
> | **TS-Sobolev** | $L=5 \times 10^5, k=10$ | $144.13 \pm 0.48$ |
> | **Sinkhorn** |  | $157.80 \pm 0.60$ |
>
> **Table 3: Runtime Comparison at $n=256$**
>
> | Method | Parameters | Runtime (ms) |
> | :--- | :--- | :--- |
> | **TS-Sobolev** | $L=10^5, k=10$ | $161.01 \pm 0.23$ |
> | **Sinkhorn** |  | $1147.25 \pm 30.61$ |
>
> Regarding the scaling relationship, we confirm that $k$ (lines per tree) and $L$ (number of trees) are hyperparameters that do not depend on the sample size $n$. They are chosen freely based on the available computational budget.
>
> Finally, we thank the Reviewer for highlighting the trade-off between expressiveness and sampling complexity. Increasing $k$ captures complex geometry but expands the tree topology space exponentially, increasing Monte Carlo estimator variance. Consequently, larger $k$ requires a significant increase in $L$ to maintain convergence. **Appendix F.6** discusses this scaling behavior and the diminishing returns of increasing $k$ without sufficient $L$. In our experiments, we find $k < 10$ to provide a good trade-off between expressiveness and sampling complexity.

---

> ### Author Response · Authors · 2025-11-24
> **Reply [2/5]**
>
> **W2. In the spherical setting, I suggest including additional sliced OT baselines for a more complete comparison — for example, “Spherical Sliced Optimal Transport” (arXiv:2411.06055) and “Stereographic Spherical Sliced Wasserstein Distances” (arXiv:2402.02345). These would provide a stronger empirical context and highlight the advantages of the proposed method under spherical geometry.**
>
> **Answer W2.** We thank the Reviewer for the valuable suggestion. In response, we have conducted additional experiments in the spherical setting for the topic modeling and spherical gradient flow tasks, incorporating the two recommended baselines.
>
> The results for topic modeling are summarized in **Table 4**, and the results for spherical gradient flow are presented in **Table 5**. In both tasks, STS-Sobolev consistently outperforms the new spherical baselines, reinforcing the advantages of our proposed method.
>
> **Table 4: Average topic coherence $CV$ ($\uparrow$) on BBC and M10 over 10 runs.**
> | Method | BBC | M10 |
> | :--- | :--- | :--- |
> | SSW | $0.789\pm0.021$ | $0.446\pm0.012$ |
> | STSW | $0.795\pm0.021$ | $0.438\pm0.017$ |
> | S3W [1] | $0.785\pm0.019$ | $0.442\pm0.016$ |
> | LSSOT [2] | $0.793\pm0.014$ | $0.404\pm0.027$ |
> | **STS-Sobolev$_{2}$ (Ours)** | $\mathbf{0.804} \pm \mathbf{0.008}$ | $\mathbf{0.462} \pm \mathbf{0.020}$ |
>
> **Table 5: Log of the Wasserstein distance ($\downarrow$) between source and target distributions over 10 runs on a mixture of 12 vMFs (Spherical Gradient Flow).**
> | Methods | Epoch 50 | Epoch 100 | Epoch 150 | Epoch 200 | Epoch 250 |
> | :--- | :--- | :--- | :--- | :--- | :--- |
> | SSW | $-2.439 \pm 0.053$ | $-2.787 \pm 0.040$ | $-2.909 \pm 0.041$ | $-2.979 \pm 0.037$ | $-3.014 \pm 0.034$ |
> | S3W [1] | $-2.022 \pm 0.036$ | $-2.211 \pm 0.045$ | $-2.284 \pm 0.056$ | $-2.290 \pm 0.054$ | $-2.289 \pm 0.064$ |
> | LSSOT [2] | $-2.078 \pm 0.030$ | $-2.444 \pm 0.023$ | $-2.546 \pm 0.023$ | $-2.582 \pm 0.023$ | $-2.598 \pm 0.021$ |
> | STSW | $-2.693 \pm 0.030$ | $-3.171 \pm 0.041$ | $-3.376 \pm 0.031$ | $-3.488 \pm 0.049$ | $-3.549 \pm 0.072$ |
> | **STS-Sobolev$_{1.5}$** | $\mathbf{-3.099 \pm 0.032}$ | $-3.324 \pm 0.050$ | $-3.427 \pm 0.055$ | $-3.499 \pm 0.064$ | $-3.540 \pm 0.078$ |
> | **STS-Sobolev$_{2}$** | $-3.081 \pm 0.026$ | $\mathbf{-3.376 \pm 0.058}$ | $\mathbf{-3.513 \pm 0.094}$ | $\mathbf{-3.578 \pm 0.108}$ | $\mathbf{-3.616 \pm 0.123}$ |

---

> ### Author Response · Authors · 2025-11-24
> **Reply [3/5]**
>
> **W3. The Sobolev IPM involves gradient norms $|\nabla f|$, which are central to the metric’s definition. However, estimating $\nabla f$ on discrete samples and through non-smooth tree-sliced projections may introduce numerical instability or high-variance gradients. Prior work on Sobolev-based metrics (Mroueh et al., ICLR 2018; Deshpande et al., NeurIPS 2019; Korotin et al., ICML 2021) has highlighted that $\nabla f$ estimation can become unstable without smoothing or gradient regularization. The paper would benefit from analyzing or mitigating such potential instability in the proposed tree-sliced formulation.**
>
> **Answer W3.** We appreciate the Reviewer's concern regarding numerical instability. However, it is important to clarify that our method avoids the sources of instability found in the cited works.
>
> The instability described in cited works is from the need to solve inner optimization problems to approximate the metric. For instance, Sobolev GAN [3] and neural OT solvers [4] rely on parameterizing the critic with neural networks, creating a min-max optimization landscape. Similarly, Max-Sliced Wasserstein [5] requires solving an optimization problem to identify the worst-case projection direction. In contrast, our TS-Sobolev formulation does not perform any inner optimization. Instead, it leverages the Regularized Sobolev IPM, which admits a precise closed-form analytical expression on trees. Just like the standard Sliced Wasserstein or Tree-Sliced Wasserstein (TSW), our metric is defined as an expectation over random tree systems, which we approximate via Monte Carlo sampling. By replacing inner optimization or projection search with a closed-form expectation, we ensure that the metric evaluation is numerically stable. In our evaluations across various applications, we found the optimization using TS-Sobolev to be stable without requiring the additional regularization.
>
> Furthermore, the proposed formulation provides inherent stability for downstream tasks through favorable gradient dynamics. The metric minimizes terms of the form $L(\Delta_e) = |\Delta_e|^p$, where $\Delta_e = \mu(\gamma_e) - \nu(\gamma_e)$ as in Equation 5. For the $p=1$ case (TSW), the objective resembles an $L^1$ loss, which lacks strict convexity and produces constant-magnitude gradients that can lead to oscillation around the optimum. In contrast, for $p > 1$, the cost function becomes strictly convex with respect to the transported mass variables. This guarantees unique minimizers and ensures that gradient magnitudes scale proportionally with the error ($\propto |\Delta_e|^{p-1}$), providing large gradients for initial alignment and naturally decaying gradients for stable fine-tuning. While standard $p$-Wasserstein distances ($W_p$) theoretically offer these same optimization benefits over $W_1$, they are computationally intractable to compute on trees for $p > 1$ as they lack a closed-form solution. This limitation has previously restricted tree-sliced methods to the $p=1$ case. Our TS-Sobolev formulation bridges this gap, offering a scalable tree-sliced metric that unlocks the better optimization dynamics of $p>1$ while maintaining the computational efficiency of $W_1$.
>
> Appendix F.8 of the revised paper provides a more comprehensive treatment of TS-Sobolev's optimization advantages.
>
> **Q1. Suppose the tree $\mathbb{T} \subset \mathbb{R}$, i.e., all nodes lie on a straight line. In this case, the tree-sliced construction should coincide with the original sliced OT construction. Under this 1D setting, can we claim that the regularized Sobolev IPM (Eq. (4)) or the Sobolev IPM (Eq. (3)) is equivalent to $W_p(\mu, \nu)$?  ... Could the authors clarify this or specify under what conditions the two coincide?**
>
> **Answer Q1.** We agree with the reviewer that for $p > 1$, the proposed metric is not equivalent to the standard $p$-Wasserstein distance ($W_p$). This is evident from the definition of the Regularized Sobolev IPM in Equation (4):
> $$\hat{\mathcal{S}}\_{p}(\mu,\nu)^{p} =\int\_{\mathcal{T}} \hat{w}(x)^{1-p} \vert\mu(\Lambda(x))-\nu(\Lambda(x))\vert^{p} \omega(dx).$$
>
> For $p > 1$, this formulation differs from the standard $W_p$ definition. However, strictly for the case of $p=1$, the exponent $1-p$ becomes $0$, yielding $\hat{w}(x)^0 = 1$. Consequently, Equation (4) simplifies to:
>
> $$\hat{\mathcal{S}}\_{1}(\mu,\nu) = \int\_{\mathcal{T}}\vert\mu(\Lambda(x))-\nu(\Lambda(x))\vert\omega(dx),$$
>
> which exactly recovers the closed-form solution for the 1-Wasserstein distance ($W_1$) on a tree (Equation (2)).
>
> Therefore, if $p=1$, TS-Sobolev reduces to the Tree-Sliced Wasserstein (TSW) distance. Furthermore, as established in prior work, if the tree consists of a single line ($k=1$), TSW reduces to the Sliced 1-Wasserstein (SW$_1$) distance.
>
> Thus, our method recovers the classical SW distance (SW$_{1}$) if and only if $p=1$ and $k=1$.

---

> ### Author Response · Authors · 2025-11-24
> **Reply [4/5]**
>
> **Q2. The paper motivates Sobolev IPM as more “frequency-aware” and capable of mitigating spectral bias, but the experiments mainly show performance improvements without spectral evidence. Could the authors include a more direct spectral analysis (e.g., using synthetic 1D or 2D signals with known low/high-frequency components) to verify that the proposed tree-sliced Sobolev formulation indeed captures or preserves high-frequency modes better than standard sliced OT?**
>
> **Answer Q2.** We thank the Reviewer for their insightful comments. We acknowledge that the original manuscript did not fully exlain this advantage, and we appreciate the Reviewer prompting a deeper investigation into this motivation. In response, we provide the following analysis and synthetic experiments to demonstrate the method's capacity for preserving fine-grained structure (high-frequency signals)."
>
>
> TS-Sobolev employs a weighting mechanism that prioritizes the preservation of fine-grained features. As derived in the Equation (4), the metric minimizes a cost weighted by the term $\hat{w}(x)^{1-p}$, where $\hat{w}(x) = 1 + \omega(\Lambda(x))$ represents the weight of the subtree rooted at $x$. For $p > 1$, this weighting factor decays as the subtree size $\omega(\Lambda(x))$ increases. Consequently, this mechanism downscale dominant gradients arising from the root and upper levels of the tree (where $\omega(\Lambda(x))$ is large), preventing global mass shifts from overwhelming the optimizer. Conversely, nodes deeper in the tree (near the leaves) possess smaller subtree weights, resulting in substantially larger values for $\hat{w}(x)^{1-p}$. This effectively concentrates the optimization signal on minimizing local discrepancies, ensuring the capture and preservation of fine-grained details.
>
> In image generation, fine-grained details correspond to high-frequency features such as intricate textures and sharp edges. Consequently, this capacity to prioritize local feature details is a key factor driving the enhanced sample quality and sharpness observed in our large-scale diffusion training.
>
> To further demonstrate the ability to capture fine-grained high-frequency signals, we conducted a controlled experiment using a 1D synthetic signal composed of distinct frequency modes. We defined a target probability density $p(x)$ on the domain $[0, 1]$ as a mixture of sine waves:
>
> $$
> p(x) \propto 1 + 0.5\sin(2\pi \cdot k_{low} \cdot x) + 0.3\sin(2\pi \cdot k_{high} \cdot x)
> $$
>
> where $k_{low} = 2$ represents low-frequency signals and $k_{high} = 20$ represents high-frequency signals. We initialized $N=10000$ particles from a uniform distribution $\mathcal{U}[0, 1]$ and optimized their positions via gradient flow to minimize the distance to $p$, comparing TSW against TS-Sobolev$_2$. To quantify the capture of frequency modes, we computed the Discrete Fourier Transform (DFT), denoted as $\mathcal{F}$, of both the particle density $\hat{p}$ and the target $p$. We calculated the relative spectral error as the magnitude of the difference between their spectral components: low-frequency error $\vert\mathcal{F}(p)[k\_{low}] - \mathcal{F}(\hat{p})[k\_{low}]\vert$ and high-frequency error $\vert\mathcal{F}(p)[k\_{high}] - \mathcal{F}(\hat{p})[k\_{high}]\vert$.
>
> The quantitative results are summarized in the table below, and the error trajectories are visualized in Figure 8 in the revised paper. In the low-frequency regime, both metrics perform similarly. At final, TSW achieves a low-frequency error of $34.68$ compared to $33.94$ for TS-Sobolev, indicating both effectively capture global structure. However, a significant disparity emerges in the high-frequency regime. While the high-frequency error for TSW plateaus, the error for TS-Sobolev$_2$ consistently decreases, reaching $14.16$ compared to $20.15$ for TSW. This clear trend demonstrates that TS-Sobolev$_2$ captures high-frequency signals significantly better than TSW, confirming our theoretical analysis regarding the preservation of fine-grained structure.
>
> **Table: Low-Frequency and High-Frequency Error ($\downarrow$) over 10 runs.**
> | Iter | Low Freq Error (TSW) | Low Freq Error (TS-Sobolev$_2$) | High Freq Error (TSW) | High Freq Error (TS-Sobolev$_2$) |
> | :--- | :--- | :--- | :--- | :--- |
> | 250 | $48.18 \pm 1.45$ | $\mathbf{47.80 \pm 1.27}$ | $\mathbf{25.24 \pm 2.06}$ | $25.29 \pm 2.55$ |
> | 500 | $44.55 \pm 2.12$ | $\mathbf{43.98 \pm 1.80}$ | $21.01 \pm 1.94$ | $\mathbf{20.45 \pm 2.18}$ |
> | 750 | $39.96 \pm 2.46$ | $\mathbf{39.25 \pm 2.24}$ | $19.24 \pm 0.90$ | $\mathbf{16.05 \pm 1.50}$ |
> | 1000 | $34.68 \pm 2.75$ | $\mathbf{33.94 \pm 2.55}$ | $20.15 \pm 1.39$ | $\mathbf{14.16 \pm 1.15}$ |
>
> *(Refer to Figure 8 in the revised paper for the error trajectories visualization.)*

---

> ### Author Response · Authors · 2025-11-24
> **Reply [5/5]**
>
> We sincerely thank the reviewer for their constructive feedback, which has helped improve our manuscript. We have revised the paper to address these points. We hope our response clarifies the concerns raised. We remain available to address any further questions during the discussion period.
>
> *References.*
>
> [1] Tran et al. *Stereographic Spherical Sliced Wasserstein Distances*. ICML 2024.
>
> [2] Liu et al. *Linear Spherical Sliced Optimal Transport: A Fast Metric for Comparing Spherical Data*. ICLR 2025.
>
> [3] Mroueh et al. *Sobolev GAN.* ICLR 2018.
>
> [4] Korotin et al. *Wasserstein-2 Generative Networks.* ICLR 2021.
>
> [5] Deshpande et al. *Max-Sliced Wasserstein Distance and its use for GANs.* CVPR 2019.

---

### Author Response · Authors · 2025-11-24
**General Response to Reviewers and Area Chairs**

Dear Reviewers and Chairs,

We sincerely thank you for your thoughtful reviews and valuable feedback. We also appreciate the significant effort the Chairs have invested in coordinating this process to ensure a fair and constructive discussion.

We have uploaded our revised manuscript and rebuttal responses, in which we highlight the changes in red. We are encouraged that Reviewers UFk1 and yezB recognized TS-Sobolev as a novel, theoretically valid, and practical generalization of the Tree-Sliced Wasserstein distance. We also appreciate that all Reviewers highlighted the computational efficiency of our method and the comprehensiveness of our empirical results.

To address the concerns raised, we have made the following key improvements:

* **Intuition:** We have posted a comment on "Advantages of Higher Order Metrics" to clarify why TS-Sobolev outperforms TSW. Specifically, we detailed the better optimization dynamics and the preservation of fine-grained features.
* **New Experiments:** We have added a synthetic experiment to explicitly demonstrate theoretical advantages (Reviewer UFk1), ablation studies across different Tree-Sliced settings (Reviewer ZHaq), and results using additional evaluation metrics (Reviewer yezB).
* **Clarifications:** We have improved the explanation of the Tree Construction process and the writing quality (Reviewer yezB), clarified computational complexity (Reviewer UFk1), and explicitly discussed the advantages of the $p=2$ case (Reviewer ZHaq).

A detailed list of changes can be found in the "Summary of Revisions" comment.

We hope that these revisions and our individual responses adequately address your concerns. Please let us know if you have any remaining questions; we are more than happy to provide further clarification during the discussion period.

Best regards,

The Authors

---

### Author Response · Authors · 2025-11-24
**Advantages of Higher Order Metrics [1/2]**

We address the common concern raised by the Reviewers regarding the intuition behind the good performance of TS-Sobolev ($p>1$) as follows.

TS-Sobolev is motivated by the need for a computationally efficient metric of order $p > 1$. While the standard Tree-Sliced Wasserstein (TSW) [1] distance lacks a tractable solution for orders $p > 1$, these higher orders are important for gradient-based learning, as $p$-Wasserstein metrics with $p>1$ offer smoother gradients compared to the $(p=1)$ case [2]. Beyond simply enabling tractability, our analysis reveals that TS-Sobolev ($p > 1$) introduces distinct advantages over standard $p$-Wasserstein. Overall, the better performance of TS-Sobolev stems from two complementary mechanisms: the improved optimization landscape inherent to the $L^p$ cost function and the preservation of fine-grained features introduced by the weighting function.

**Improved Optimization Landscape.** The choice of $p$ dictates the convexity and smoothness of the underlying optimization objective. Let $\mathcal{L}(\Delta_e) := |\Delta_e|^p = |\mu(\gamma_e) - \nu(\gamma_e)|^p$, as in Equation (5), denote the unweighted transport loss associated with the mass discrepancy $\Delta_e$ on a given edge. For $p=1$, the Tree-Sliced Wasserstein (TSW) distance behaves analogously to an $L^1$ loss. This formulation lacks strict convexity, which implies that optimization problems solving for measures $\mu$ or $\nu$ may admit non-unique minimizers, leading to potential instability [3, 4]. Furthermore, the gradient magnitude for $(p=1)$ remains constant ($|\nabla \mathcal{L}| \propto 1$, i.e., proportional to a constant) regardless of the proximity to the target, often leading to oscillations around the optimum unless the learning rate is carefully annealed. In contrast, for $p > 1$, the cost function becomes strictly convex, ensuring unique geodesics and well-conditioned gradient signals [2]. Crucially, the gradient magnitude scales with the transport cost, following $|\nabla \mathcal{L}| \propto |\Delta_e|^{p-1}$. This property ensures that gradients are large when distributions are distinct and vanish smoothly as $\Delta_e \to 0$, facilitating stable fine-tuning and convergence.

These better optimization characteristics directly translate into improved performance in the Gradient Flow experiments across both Euclidean and Spherical benchmarks.

---

> ### Author Response · Authors · 2025-11-24
> **Advantages of Higher Order Metrics [2/2]**
>
> **Preserving Fine-Grained Structure.** Beyond the optimization benefits inherent to standard $W_p$ metrics, TS-Sobolev employs a weighting mechanism that uniquely prioritizes the preservation of fine-grained features. As derived in Equation (4), the metric minimizes a cost weighted by the term $\hat{w}(x)^{1-p}$, where $\hat{w}(x) = 1 + \omega(\Lambda(x))$ represents the weight of the subtree rooted at $x$. For $p > 1$, this weighting factor decays as the subtree size $\omega(\Lambda(x))$ increases. Consequently, this mechanism downscale dominant gradients arising from the root and upper levels of the tree (where $\omega(\Lambda(x))$ is large), preventing global mass shifts from overwhelming the optimizer. Conversely, nodes deeper in the tree (near the leaves) possess smaller subtree weights, resulting in substantially larger values for $\hat{w}(x)^{1-p}$. This effectively concentrates the optimization signal on minimizing local discrepancies, ensuring the capture and preservation of fine-grained details.
>
> In image generation, fine-grained details correspond to high-frequency features such as intricate textures and sharp edges. Consequently, this capacity to prioritize local feature details is a key factor driving the enhanced sample quality and sharpness observed in our large-scale diffusion training.
>
> To further demonstrate the ability to capture fine-grained high-frequency signals, we conducted a controlled experiment using a 1D synthetic signal composed of distinct frequency modes. We defined a target probability density $p(x)$ on the domain $[0, 1]$ as a mixture of sine waves:
>
> $$
> p(x) \propto 1 + 0.5\sin(2\pi \cdot k_{low} \cdot x) + 0.3\sin(2\pi \cdot k_{high} \cdot x)
> $$
>
> where $k_{low} = 2$ represents low-frequency signals and $k_{high} = 20$ represents high-frequency signals. We initialized $N=10000$ particles from a uniform distribution $\mathcal{U}[0, 1]$ and optimized their positions via gradient flow to minimize the distance to $p$, comparing TSW against TS-Sobolev$_2$ . To quantify the capture of frequency modes, we computed the Discrete Fourier Transform (DFT), denoted as $\mathcal{F}$, of both the particle density $\hat{p}$ and the target $p$. We calculated the relative spectral error as the magnitude of the difference between their spectral components:  low-frequency error $\vert\mathcal{F}(p)[k\_{low}] - \mathcal{F}(\hat{p})[k\_{low}]\vert$ and high-frequency error $\vert \mathcal{F}(p)[k\_{high}] - \mathcal{F}(\hat{p})[k\_{high}]\vert $.
>
> The quantitative results are summarized in the table below, and the error trajectories are visualized in Figure 8 in the revised paper. In the low-frequency regime, both metrics perform similarly. At final iterations, TSW achieves a low-frequency error of $34.68$ compared to $33.94$ for TS-Sobolev, indicating both effectively capture global structure. However, a significant disparity emerges in the high-frequency regime. While the high-frequency error for TSW plateaus, the error for TS-Sobolev$_2$ consistently decreases, reaching $14.16$ compared to $20.15$ for TSW. This clear trend demonstrates that TS-Sobolev$_2$ captures high-frequency signals significantly better than TSW, confirming our theoretical analysis regarding the preservation of fine-grained structure.
>
> **Table: Low-Frequency and High-Frequency Error ($\downarrow$) over 10 runs.**
> | Iter | Low Freq Error (TSW) | Low Freq Error (TS-Sobolev$_2$) | High Freq Error (TSW) | High Freq Error (TS-Sobolev$_2$) |
> | :--- | :--- | :--- | :--- | :--- |
> | 250 | $48.18 \pm 1.45$ | $\mathbf{47.80 \pm 1.27}$ | $\mathbf{25.24 \pm 2.06}$ | $25.29 \pm 2.55$ |
> | 500 | $44.55 \pm 2.12$ | $\mathbf{43.98 \pm 1.80}$ | $21.01 \pm 1.94$ | $\mathbf{20.45 \pm 2.18}$ |
> | 750 | $39.96 \pm 2.46$ | $\mathbf{39.25 \pm 2.24}$ | $19.24 \pm 0.90$ | $\mathbf{16.05 \pm 1.50}$ |
> | 1000 | $34.68 \pm 2.75$ | $\mathbf{33.94 \pm 2.55}$ | $20.15 \pm 1.39$ | $\mathbf{14.16 \pm 1.15}$ |
>
> *(Refer to Figure 8 in the revised paper for the error trajectories visualization.)*
>
> ---
> *References*
>
> [1] Tran et al. *Distance-based Tree-Sliced Wasserstein Distance.* ICLR 2025.
>
> [2] Peyré et al. *Computational optimal transport: With applications to data science.* Foundations and Trends in Machine Learning 2019.
>
> [3] Santambrogio. *Optimal Transport for Applied Mathematicians: Calculus of Variations, PDEs, and Modeling.* Springer International Publishing 2015.
>
> [4] Villani. *Topics in Optimal Transportation.* American Mathematical Society 2003.

---

### Author Response · Authors · 2025-11-24
**Summary of Revisions**

Following the Reviewers' insightful comments, we have made significant improvements to the manuscript. Beyond correcting typographical errors, we have implemented the following substantive changes:

* We added Appendix F.8 to provide a detailed theoretical explanation of the advantages of TS-Sobolev, specifically regarding its improved optimization landscape and ability to preserve fine-grained features.
* To empirically validate the theoretical claims, we conducted new synthetic experiments demonstrating TS-Sobolev's capability in capturing high-frequency, fine-grained signals in Appendix F.8.
* We expanded Appendix F.7 to discuss the specific benefits of using $p=2$ and provided guidelines on when to prefer Tree-Sliced Wasserstein [1] versus TS-Sobolev based on task requirements.
* We revised both the Abstract and Introduction to more clearly articulate the motivation behind developing TS-Sobolev and to explicitly highlight its advantages over existing metrics.
* We improved the transition and logical connection between Section 2.1 and Section 2.2 to better explain how the Tree-Sliced Framework adapts Sobolev IPM for machine learning data.
* We updated the formatting and structure to visually highlight the Regularized Sobolev IPM formulation within the main text.
* We added the computational complexity of the Tree Sampling process to Table 6 in Appendix F.1.
* We incorporated S3W [2] and LSSOT [3] as additional baselines for the spherical settings, revising the result analysis in Section 4.2 to show that TS-Sobolev continues to outperform all baselines.
* We re-evaluated the Gradient Flow experiments (both Euclidean and Spherical) using the 1-Wasserstein metric to verify the performance gains of TS-Sobolev, summarizing these results in Appendix F.2.
* We conducted ablation studies on tree sampling configurations (comparing tree structures and splitting maps) to verify that the benefits of TS-Sobolev translate effectively to other settings in Appendix F.2.

---
*References*

[1] Tran et al. *Distance-based Tree-Sliced Wasserstein Distance.* ICLR 2025.

[2] Tran et al. *Stereographic Spherical Sliced Wasserstein Distances*. ICML 2024.

[3] Liu et al. *Linear Spherical Sliced Optimal Transport: A Fast Metric for Comparing Spherical Data*. ICLR 2025.

---

### Author Response · Authors · 2025-12-02
**Briefing [4/4]: Additional Results from the Rebuttal Phase**

Dear New AC,

During the rebuttal period, we conducted significant additional work to validate our claims. These results have been added to the revised paper and the discussion forum.

### 1. Analyzing Advantages of Higher Order Metrics (Appendix F.8)
We provided a comprehensive analysis showing that $p>1$ offers a strictly convex optimization landscape with better gradient scaling compared to $p=1$, alongside a weighting mechanism that prioritizes fine-grained features.

Since high-frequency components correspond to fine-grained details (e.g., edges in images), we designed a synthetic experiment to validate TS-Sobolev's capability to capture them. While TSW ($p=1$) and TS-Sobolev performed similarly on low frequencies, TS-Sobolev ($p=2$) significantly reduced high-frequency error compared to TSW.

We highly encourage the AC to read our detailed response in **Appendix F.8** for the intuition behind these findings and the experiment to support them.

### 2. New Spherical Baselines (Section 4.2)
We added comparisons against S3W and LSSOT as requested by Reviewer UFk1. STS-Sobolev outperforms both new baselines across Topic Modeling and Spherical Gradient Flow tasks.

**Table 1**: Average topic coherence $CV$ ($\uparrow$) on BBC and M10.
| Method | BBC | M10 |
| :--- | :--- | :--- |
| S3W | $0.785\pm0.019$ | $0.442\pm0.016$ |
| LSSOT | $0.793\pm0.014$ | $0.404\pm0.027$ |
| **STS-Sobolev$_{2}$** | $\mathbf{0.804} \pm \mathbf{0.008}$ | $\mathbf{0.462} \pm \mathbf{0.020}$ |

**Table 2**: Log $W_2$ ($\downarrow$) on Mixture of 12 vMFs (Spherical Gradient Flow).
| Method | Epoch 250 |
| :--- | :--- |
| S3W | $-2.289 \pm 0.064$ |
| LSSOT | $-2.598 \pm 0.021$ |
| STSW | $-3.549 \pm 0.072$ |
| **STS-Sobolev$_{2}$** | $\mathbf{-3.616 \pm 0.123}$ |

### 3. Ablation of Tree Generation Schemes (Appendix F.2)
We tested TS-Sobolev under different tree generation schemes ("Chain" vs. "Concurrent") and splitting maps ("Uniform" vs. "Distance-based") to address Reviewer ZHaq's comments. TS-Sobolev consistently outperforms TSW, showing that the benefits of the TS-Sobolev are transferable across all four tree-generation schemes.

**Table 3.** Average $W_2$ distance (multiplied by $10^{-1}$) on Gaussian 30d at iteration 2500.

| Tree Sampling | Splitting Map | TSW$_1$ | TS-Sobolev$_{1.2}$ |
| :--- | :--- | :---: | :---: |
| Chain | Uniform | 2.01 | **1.89** |
| Chain | Distance | 1.56 | **1.49** |
| Concurrent | Uniform | 1.93 | **1.83** |
| Concurrent | Distance | 1.78 | **1.40** |

### 4. $W_1$ Evaluation Metric (Appendix F.2)
To address potential bias in evaluation (Reviewer yezB), we re-evaluated Euclidean Gradient flows using the $W_1$ metric. TS-Sobolev maintains better performance even when evaluated on the $W_1$ metric.

**Table 4.** Average $W_1$ distance on Gaussian 30d (Euclidean) at Iter 2500.
| Method | TSW-SL | Db-TSW | TS-Sobolev$_{1.5}$ |
| :--- | :--- | :--- | :--- |
| $W_1$ Dist | 18.8 | 16.1 | **6.69** |

**Table 5.** Average Log $W_1$ on Mixture of vMFs (Spherical) at Epoch 250.
| Method | ARI-S3W | STSW | STS-Sobolev$_2$ |
| :--- | :--- | :--- | :--- |
| Log $W_1$ | -2.136 | -2.082 | **-2.158** |

---

### Author Response · Authors · 2025-12-02
**Briefing [3/4] Response to Reviewer Concerns**

Dear New AC,

The reviewers were generally positive about the novelty, validity, scalability, and empirical results of TS-Sobolev but raised constructive questions regarding intuition and baselines. We address the primary concerns raised in our rebuttal as follows:

### 1. Computational Complexity and Tree Construction (Reviewer UFk1)
* Concern: Is the tree construction cost included? How large must parameters (like $k$) be for the method to become slower than Sinkhorn?
* Reply: We clarified that tree construction is $\mathcal{O}(Lkd)$ and negligible. We demonstrated that TS-Sobolev scales linearly with $\mathcal{O}(Lk)$ and log-linearly with $N$ ($\mathcal{O}(N \log N)$), remaining significantly faster than Sinkhorn ($\mathcal{O}(N^2)$) unless parameters like $L$ are raised to extreme values (e.g., $L=5 \times 10^5$) that far exceed standard usage.

### 2. Numerical Instability of Gradients (Reviewer UFk1)
* Concern: Potential instability in gradient estimation often found in neural-based Sobolev metrics.
* Reply: We clarified that, unlike prior Sobolev-based methods (e.g., Sobolev GAN) that require solving inner min-max optimization problems, TS-Sobolev uses a closed-form analytical expression (Equation 4). This eliminates the source of instability.

### 3. Missing Spherical Baselines (Reviewer UFk1)
* Concern: Request for comparison against recent methods like "Spherical Sliced Optimal Transport" (S3W) [1] and "Linear Spherical Sliced Optimal Transport" (LSSOT) [2].
* Reply: We implemented these baselines and added comparisons in the Spherical Gradient Flow and Topic Modeling tasks. STS-Sobolev continued to outperform both S3W and LSSOT.

### 4. Intuition for Better Performance of $p>1$ (Reviewers UFk1, ZHaq, yezB)
* Concern: Why does TS-Sobolev ($p>1$) perform better than TSW ($p=1$)?
* Reply: We provided a detailed theoretical and empirical analysis showing two key advantages:
    *  Optimization Landscape: $p>1$ yields a strictly convex cost function with gradients that scale with error magnitude (unlike the constant gradients of $L^1$), leading to more stable convergence.
    *  Fine-Grained Features: The Sobolev weighting function $\hat{w}(x)^{1-p}$ naturally down-weights global shifts (near the tree root) and up-weights local variations (near leaves), allowing the metric to prioritize fine-grained details.

### 5. Fundamental Difference between $W_p$ and $\hat{\mathcal{S}}_p$ (Reviewer ZHaq)
* Concern: What is the fundamental difference between minimizing the Regularized Sobolev IPM ($\hat{\mathcal{S}}_p$) versus the standard $p$-Wasserstein distance ($W_p$)?
* Reply: We clarified that the difference stems from the weighting function $\hat{w}(x)^{1-p}$ inherent to $\hat{\mathcal{S}}_p$, which does not appear in $W_p$.

### 6. Ablation of Tree Generation Schemes (Reviewer ZHaq)
* Concern: Request to ablate different tree generation schemes (e.g., Chain vs. Concurrent) and splitting maps to verify the transferability of the method's benefits.
* Reply: We conducted an ablation study comparing Chain vs. Concurrent tree structures and Uniform vs. Distance-based splitting maps. TS-Sobolev consistently outperformed TSW across all tree generation schemes.

### 7. Evaluation Metrics (Reviewer yezB)
* Concern: Does using $W_2$ as an evaluation metric unfairly favor our method (since we optimize for $p>1$)?
* Reply: We re-ran gradient flow experiments using $W_1$ distance as the evaluation metric. TS-Sobolev ($p>1$) still outperformed TSW ($p=1$) and other baselines.

### 8. Presentation and Writing Clarity (Reviewer yezB)
* Concern: Suggestions to improve the flow between background sections and better highlight the core equations.
* Reply: We rewrote the transition between "Tree Metric Spaces" and the "Tree-Sliced Framework" to explicitly clarify the continuous tree structure. We also updated the text formatting to visually emphasize the definition of the Regularized Sobolev IPM.
---

*References.*

[1] Tran et al. *Stereographic Spherical Sliced Wasserstein Distances*. ICML 2024.

[2] Liu et al. *Linear Spherical Sliced Optimal Transport: A Fast Metric for Comparing Spherical Data*. ICLR 2025.

---

### Author Response · Authors · 2025-12-02
**Briefing [2/4] Summary of Key Contributions**

Dear New AC,

To assist with your assessment, we summarize the core contributions and results of our submission, *"Tree-Sliced Sobolev IPM"*.

### 1. Core Contribution and Novelty
* **Problem:** Existing Tree-Sliced Wasserstein (TSW) [1] methods are computationally limited to order $p=1$. Standard $p$-Wasserstein ($p>1$) on trees is computationally intractable. This restriction is a significant bottleneck, as higher-order metrics (p > 1) are preferred in gradient-based learning for more favorable optimization landscapes.
* **Method:** We introduce TS-Sobolev (and its spherical counterpart STS-Sobolev). We leverage the *Regularized Sobolev Integral Probability Metric (IPM)*, which admits a closed-form solution on trees for any $p \ge 1$.
* **Significance:** This provides a scalable tree-sliced metric for $p>1$ with the same computational complexity ($\mathcal{O}(Lkn \log n)$) as standard TSW. It serves as a drop-in replacement for TSW with added optimization benefits.

### 2. Key Theoretical Results
* **Metric Properties:** We prove TS-Sobolev is a valid metric on Euclidean space and satisfies $E(d)$-invariance (Theorem 3.2).
* **Relation to TSW:** We prove that for $p=1$, TS-Sobolev exactly recovers TSW. For $p>1$, it is upper-bounded by TSW (Theorem 3.3).
* **Spherical Extension:** We extend the framework to the hypersphere, defining STS-Sobolev. We prove it is a valid $O(d+1)$-invariant metric (Theorem E.2) and generalizes Spherical TSW (STSW), recovering it exactly at $p=1$ (Theorem E.3).


### 3. Experimental Results (Euclidean and Spherical settings)
* **Gradient Flows:** TS-Sobolev ($p=1.2, 1.5, 2$) achieves faster convergence and lower Wasserstein error than TSW and Sliced-Wasserstein (SW) baselines on 8 Gaussians and 30D Gaussians (Table 1).
* **Generative Modeling:** We achieved SOTA FID scores on CIFAR-10 unconditional generation compared to other Sliced/Tree-Sliced methods, with no increase in training time (Table 2).
* **Spherical Self-Supervised Learning:** STS-Sobolev outperforms Spherical Tree-Sliced Wasserstein (STSW) in representation learning on CIFAR-10, achieving improved accuracy (Table 3).
* **Topic Modeling:** We achieve the highest Topic Coherence ($C_V$) on BBC and M10 datasets in both Euclidean and Spherical settings compared to LDA, ProdLDA, and other OT baselines (Table 5).
* **Runtime Efficiency:** Empirical analysis confirms that TS-Sobolev's runtime is nearly identical to TSW with $p=1$ and remains consistent across different orders of $p$, demonstrating that TS-Sobolev incur negligible computational overhead (Figure 6).

---
*References.*

[1] Tran et al. Distance-based Tree-Sliced Wasserstein Distance. ICLR 2025.

---

### Author Response · Authors · 2025-12-02
**Briefing [1/4] Note of Appreciation to the New Area Chair**

Dear new AC,

We are sincerely grateful to you for stepping in to manage our submission at this critical stage. Your support is highly appreciated.

To assist with your evaluation, we have compiled three concise summaries in the messages below:

*Briefing [2/4]: Summary of Key Contributions*

*Briefing [3/4]: Response to Reviewer Concerns*

*Briefing [4/4]: Additional Results from the Rebuttal Phase*

As the discussion period concluded earlier than expected, we were unable to get further feedback from the other reviewers.

We kindly request that you carefully review our paper, the rebuttal, and additional results. We ask that you consider how we have resolved all raised concerns. We are confident that the reviewers would have been satisfied with our rebuttals and would have considered improving their scores had the discussion continued.

We trust that your thorough oversight will ensure a fair and accurate assessment of our work by the AC, SAC, and PC.

Best regards,

The Authors

---

### Meta-Review · Area_Chair_oroc · 2026-01-06

**Summary:**

In general, the original reviews were rather positive about the originality of the proposed method, and its good performances in the various experimental settings. No critical issue was raised in the original  reviewers. Among other, most of the reviewers asked clarifications about why the case p >1 was of interest and performed better, for which Authors added a substantial analysis in the revised version. Other comments (regarding missing baselines or lack of clarity in the explanations) were also addressed  successfully by the authors.  Overall, I believe that this is a solid piece of work which I recommend for acceptance.

**Reviewer Concerns:**

Inituitions and experiments for why considering p >1 were added to the paper
Missing baselines on Spherical Ot were added.
Explanations on the cost of contructing trees were discussed.
Ablations studies (notably wrt. Tree Generation Schemes) were added

**Reviewer Scores:**

Given my perception of the answers, some reviewers might have raised their socre from 6 to 7 given the responses of the Authors.

---

### Decision · Program_Chairs · 2026-01-26

Accept (Poster)